# Developing a sequential cropping capability in the JULESvn5.2 land–surface model

Camilla Mathison[1,2], Andrew J Challinor[2], Chetan Deva[2], Pete Falloon[1], Sébastien Garrigues[3,4], Sophie Moulin[3,4], Karina Williams[1,5], and Andy Wiltshire[1,5]

[1]Met Office Hadley Centre, FitzRoy Road, Exeter, EX1 3PB, UK
[2]School of Earth and Environment, Institute for Climate and Atmospheric Science, University of Leeds, Leeds, LS2 9AT, UK
[3]EMMAH (UMR1114), INRAE, Avignon, France
[4]Université d'Avignon et des Pays de Vaucluse, UMR1114 – EMMAH, 84000 Avignon, France
[5]School of, University of Exeter, Exeter, EX

**Correspondence:** Camilla Mathison (camilla.mathison@metoffice.gov.uk)

**Abstract.**

Sequential cropping (also known as multiple or double cropping) is common in tropical regions, where the crop seasons are largely dictated by the main wet season. The Asian summer monsoon (ASM) provides the water resources for crops grown for the whole year, thereby influencing crop production outside the ASM period. Land surface models (LSMs) typically simulate a single crop per year in a field or location. However, in order to understand how sequential cropping influences demand for resources, we simulate all the crops grown within a year in a field or location in a seamless way. In this paper we implement sequential cropping in a branch of the Joint UK Land Environment Simulator (JULES) and demonstrate its use at Avignon, a site that uses a form of the sequential cropping system. Avignon provides over 15-years of continuous flux observations which we use to evaluate JULES with sequential cropping. We apply the same method to a regional and 4-single-gridbox simulations for the North Indian states of Uttar Pradesh and Bihar to simulate the regular rice–wheat rotation, where there is a variation in growing conditions. The inclusion of a secondary crop in JULES using the sequential cropping method presented does not change the crop growth or development of the primary crop. During the secondary crop growing period, the carbon and energy fluxes for an irregular (Avignon) and regular crop rotation (India single gridboxes) are modified; they are largely unchanged for the primary crop growing period. In a regular crop rotation, the inclusion of a secondary crop using this sequential cropping method affects the available soil moisture in the top 1.0 m throughout the year, with larger fluctuations in sequential crops compared with single crop simulations even outside the secondary crop growing period. JULES simulates sequential cropping at Avignon, the four India locations and the regional run; representing both crops within one growing season in each of the crop rotations presented. This development is a step forward in the ability of JULES to simulate crops in tropical regions, where this cropping system is already prevalent. It also provides the opportunity to assess the potential for other regions to implement sequential cropping as an adaptation to climate change.

# 1 Introduction

Climate change is likely to impact all aspects of crop production affecting plant growth, development and crop yield (Hatfield and Prueger, 2015) as well as cropping area and cropping intensity (Iizumi and Ramankutty, 2015). The impact of climate change on agriculture has been the focus of several large collaborative projects such as the Agricultural Model Intercomparison and Improvement Project (AgMIP; Rivington and Koo, 2010; Rosenzweig et al., 2013, 2014) and the Inter-Sectoral Impact Model Intercomparison Project (ISIMIP; Warszawski et al., 2013, 2014). These projects have highlighted the likelihood of competition between crops grown for food and those grown for bio-energy in order to mitigate climate change (Frieler et al., 2015). Petrie et al. (2017) discuss how the use of sequential cropping systems may have made it possible for populations in some areas to adapt to large changes in monsoon rainfall between 2200–2100 BC. These ancient agricultural practices are common today across most tropical countries but may also be a useful adaptation, especially where traditionally mono-crop systems are currently used, in order to meet a future rising demand for food (Hudson, 2009) or the demand for bio-fuels. This sort of adaptation is already happening in some locations. Mueller et al. (2015) show that longer growing seasons in the extratropics have made the cultivation of multiple crops in a year at northern latitudes more viable. Warmer spring temperatures in the Brahmaputra catchment have allowed earlier planting of a winter crop, leaving time for a second crop (Zhang et al., 2013).

The South Asia economy is highly dependent on the agricultural industry and other industries also with a high demand for water (Mathison et al., 2015). The most important source of water for this part of the world is the Asian Summer Monsoon (ASM), which typically occurs between June and September (Goswami and Xavier, 2005); this phenomenon provides most of the water resource for any given year. The South Asia crop calendar is defined by the ASM, which has an important influence on the productivity across the whole year (Mathison et al., 2018), thereby affecting crop production outside the Monsoon period.

Intercropping or sequential cropping allow farmers to make the most efficient use of limited resources and space in order to maximize yield potential and lower the risk of complete crop failure. These techniques also influence ground cover, soil erosion and chemical properties, albedo and pest infestation (Waha et al., 2013). Intercropping is the simultaneous cultivation of multiple crop species in a single field (Cong et al., 2015) while sequential cropping (also called multiple or double cropping) involves growing two or more crops on the same field in a given year (Liu et al., 2013; Waha et al., 2013). We use the term sequential cropping from here on to avoid confusion with other cropping systems. Sequential cropping systems are common in Brazil where the soybean–maize or soybean–cotton rotations are used (Pires et al., 2016) and for South Asia where the rice–wheat systems are the most extensive, dominating in many Indian states (Mahajan and Gupta, 2009), across the Indo-Gangetic Plain (IGP) (Erenstein and Laxmi, 2008) and Pakistan (Erenstein et al., 2008). States such as Punjab, Haryana, Bihar, Uttar Pradesh and Madhya Pradesh (Mahajan and Gupta, 2009) account for approximately 75 % of national food grain production for India. Rice-rice rotations are the second most prevalent crop rotation to rice-wheat rotations, these are typically found in the north eastern regions of India and Bangladesh (Sharma and Sharma, 2015) with some regions cultivating as many as three rice crops per year.

## 1.1 Modelling sequential cropping

The modelling of crop rotations is a regular feature of soil carbon simulations (Bhattacharyya et al., 2007). Bhattacharyya et al. (2007) found that the rice–wheat rotation, common across the IGP, has helped maintain carbon stocks. However, in recent years, the yields of rice and wheat have plateaued, leading farmers to diversify and include other additional crops in the rotation, potentially depleting carbon stocks. The modelling of crop rotations has also been represented in the field of agricultural economics with work regarding sequential cropping being mainly to understand influences on decision-making; therefore focusing on short timescales and at the farm management level (Dury et al., 2012; Caldwell and Hansen, 1993).

Many dynamic global vegetation models (DGVMs), used to study the effects of climate change, simulate a single crop in a field per year, both for individual sites and gridded simulations. This may be due in part to some global observation datasets such as Sacks et al. (2010) reporting only one growing period per year at a given location for most crops (Waha et al., 2012). Where different crop calendars are available for different regions e.g. MIRCA2000 (Portmann et al., 2010), rice and wheat are divided equally between the kharif (i.e. sown during the monsoon and harvested during the autumn) and rabi seasons (i.e. the drier winter/spring growing season), when in reality wheat is only grown during the rabi season (Biemans et al., 2016).

The Lund–Potsdam–Jena managed Land model (LPJml- Bondeau et al., 2007) is one of the few models that is able to simulate sequential cropping. Sharma and Sharma (2015) use LPJml to simulate monoculture systems such as the rice–rice system grown in Bangladesh, while Waha et al. (2013) extend LPJml to consider sequential cropping in Africa for two different crops on the same field within a year. Waha et al. (2013) specify different growing periods for each crop in the rotation, where the growing period is calculated from the sum of the daily temperatures above a crop specific temperature threshold. Waha et al. (2013) use the Waha et al. (2012) method to specify the onset of the main rainy season as the start of the growing season, where growing season is defined as the period of time in which temperature and moisture conditions are suitable for crop growth. The growing period of the first crop in the rotation begins on the first wet day of the growing season, with the second crop assumed to start immediately after harvest of the first crop. Waha et al. (2013) find that when considering the impact of climate change, the type of cropping system is important because yields differ between crops and cropping systems. Biemans et al. (2016) also use a version of LPJml refined for South Asia, to estimate water demand and crop production for South Asia. Biemans et al. (2016) combine the output from two separate simulations, each with different kharif and rabi land-use maps and zonal sowing and harvest dates based on observed monsoon patterns. Biemans et al. (2016) find that accounting for multiple different crops being grown on the same area at different times of the year improves the simulations of demand for water for irrigation, particularly the timing of the demand. Waha et al. (2013) and Biemans et al. (2016) simulate more than one crop growing on the same area using very different methods, both have highlighted the importance of representing this type of cropping system. It would be beneficial for more land-surface models to develop the capability to simulate different cropping systems and link crop production with irrigation both to improve the representation of the land surface in coupled models and to improve climate impacts assessments.

The JULES model is the land-surface scheme used by the UK Met Office for both weather and climate applications. It is also a community model and can be used in standalone mode; which is how it is used in the work presented here. The

parametrisation of crops in JULES (JULES-crop) is described in Osborne et al. (2015) and Williams et al. (2017). JULES-crop is a dual-purpose crop model intended for use both within standalone JULES, enabling a focus on food production and water availability applications, as well as being the land-surface scheme within climate and earth system models. JULES-crop has been used in standalone mode in recent studies such as Williams and Falloon (2015) and Williams et al. (2017). The aim is that these studies and this one, will lead to using JULES in these larger models to allow the feed-backs from regions with extensive croplands and irrigation systems, like South Asia, to have an effect on the atmosphere e.g. via Methane emissions from rice paddies or evaporation from irrigated fields (Betts, 2005).

## 1.2 JULES-crop: rationale for sequential cropping in JULES-Crop

JULES-crop is typically run as a single crop model, represented by the red curve in Fig. 1, where a primary crop is simulated but no second crop is possible and the land is left fallow with a minimum surface cover. In many regions, sequential cropping is the main cropping system used, with several crops cultivated one after another. JULES-crop has been developed for implementation in Earth System and Climate models for application in adaptation and mitigation studies. Only being able to simulate one crop per year is therefore limiting application in many parts of the world. In the changes to JULES described in this paper, new controls are implemented to allow the current JULES-crop code to be run more than once in a year at a particular location, so that sequential cropping systems can be represented in JULES. Sequential cropping is available from version 5.7 of JULES, this option is represented by the black curve in Fig. 1.

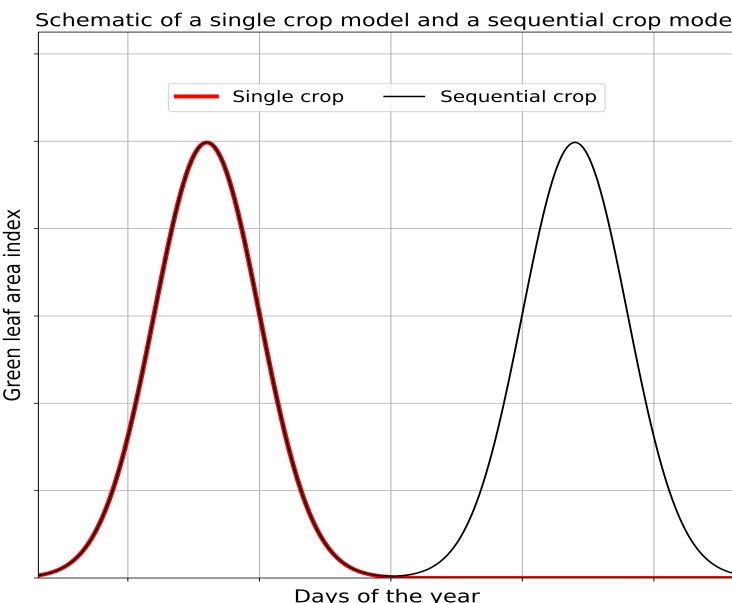

**Figure 1.** A schematic describing the single crop model (red curve) that is part of the standard JULES-crop and the new option for including sequential cropping (black curve). This schematic represents a generic crop at a single location.

This is part of a project to develop simulations for South Asia to understand the impacts of climate change on both agriculture and water sectors (Mathison et al., 2015, 2018) using existing RCM projections (Kumar et al., 2013; Mathison et al., 2013). This will improve understanding of the impacts of climate change and how they affect each other. Sequential cropping provides clear added benefits for the following reasons:

– providing a more realistic representation of the observed surface land cover.

– allowing the continuous simulation of a location where different crops are grown within the same area, thereby simulating water resource demand from crops.

– allowing the climate to affect both the water and crops, while simultaneously allowing interactions between water and crops throughout the year makes it possible to simulate the integrated impacts of climate change on these two sectors.

– providing the opportunity to investigate the impact of adopting sequential cropping for regions where it is not currently used.

A site in Avignon, France (Garrigues et al., 2015, 2018) and two states in India are simulated to illustrate and evaluate the method implemented in the JULES standalone model at version 5.2 for simulating crop rotations; representing both irregular rotations (as at Avignon) and the regular sequential cropping systems used in India. The method is summarized by Fig. 2 and
described in Sect. 3. The objectives of this study are to test the following hypotheses with regard to the implementation of the presented sequential cropping method in JULES:

1. **Null hypothesis**: the inclusion of a secondary crop on the same field does not change the growth and development of the primary crop in an irregular sequential cropping rotation with long fallow periods.
   **Alternative hypothesis**: the inclusion of a secondary crop on the same field modifies the growth and development of the
   primary crop in an irregular sequential cropping rotation with long fallow periods.

2. **Null hypothesis**: the inclusion of a secondary crop on the same field does not change the energy and carbon fluxes in an irregular sequential cropping rotation with long fallow periods.
   **Alternative hypothesis**: the inclusion of a secondary crop on the same field modifies the energy and carbon fluxes in an irregular sequential cropping rotation with long fallow periods.

3. **Null hypothesis**: in a regular rotation without long fallow periods, the inclusion of a secondary crop on the same field does not change the crop development of the primary crop or the gridbox energy and carbon fluxes and soil conditions.
   **Alternative hypothesis**: in a regular rotation without long fallow periods, the inclusion of a secondary crop on the same field modifies the crop development of the primary crop, the gridbox energy and carbon fluxes and soil conditions.

Hypotheses 1 will be assessed by comparison of the observed Leaf Area Index (LAI), canopy height and total above ground
biomass at Avignon with single crop and sequential crop simulations. Hypotheses 2 will be assessed by comparison of observed fluxes, Gross Primary Productivity (GPP), latent heat ($LE$) and sensible heat ($H$) at Avignon with single crop and sequential

crop simulations. For Hypotheses 3 we compare JULES yields with observed yields for single and sequential crop simulations and analyse the same variables as for Hypotheses 1 and 2 for four locations across the North Indian states of Uttar Pradesh and Bihar. We also assess if the implementation of sequential crops affects the soil moisture in a regular sequential crop system, which does not have long periods of bare soil. For a regular sequential crop system without long fallow periods, changes in soil
moisture are more likely to be due to the effects of sequential cropping and are less likely to be affected by evaporation from bare soil.

The Avignon site represents the irregular cropping rotation, chosen because it has been observed and documented over several years (2001 to 2014), growing a range of crops throughout this period. No equivalent site to Avignon has been found for South Asia. The continuous measurements of surface fluxes provided by the Avignon dataset are a unique resource for
evaluating land surface models (LSMs) and for testing and implementing more irregular crop rotations in LSMs. Garrigues et al. (2015) use this dataset to evaluate LSM simulations of evapotranspiration using the interactions between soil, biosphere, and atmosphere scheme (ISBA) LSM (Noilhan and Planton, 1989) specifically, the version from Calvet et al. (1998); ISBA-A-gs. We focus on a two-crop-rotation between 2005 and 2012.

The North Indian states of Uttar Pradesh and Bihar represent the regular cropping rotation, chosen because the rice-wheat
rotation is the dominant cropping system there, with these states being key producers of these crops. The method is used in a series of single gridbox simulations across these two states to allow for comparison with Avignon and in a regional simulation to demonstrate that this method can be applied at larger scales.

The paper is structured as follows, Section 2 describes the JULES model and the method for implementing the sequential cropping system in JULES is outlined in Sect. 3. The simulations are described in Sect. 4, the observations used in Sect. 5, the
results in Sect. 6 and discussion in Sect. 7. Conclusions are provided in Sect. 8.

## 2   Model description

JULES is a process-based model that simulates the fluxes of carbon, water, energy and momentum between the land-surface and the atmosphere. JULES represents both vegetation (including natural vegetation and crops) and non-vegetation surface types including; urban areas, bare soil, lakes, and ice. With the exception of the ice tile all these tiles can co-exist within a
gridbox so that a fraction of the surface within each gridbox is allocated between surface types. For the ice tile a grid box must be either completely covered in ice or not (Shannon et al., 2018). JULES treats each vegetation type as a separate tile within a gridbox, with each one represented individually with its own set of parameters and properties, such that each tile has a separate energy balance. The model and the equations it is based on are described in detail in Best et al. (2011) and Clark et al. (2011). Prognostics such as leaf area index (LAI) and canopy height are therefore available for each tile. The forcing air temperature,
humidity and windspeed are prescribed for the gridbox as a whole for a given height. Below the surface the soil type is also uniform across each gridbox (where the number of soil tiles is set to one). We use JULES-crop (Osborne et al., 2015; Williams et al., 2017) to simulate the crops in this study. The main aim of JULES-crop is to improve the simulation of land-atmosphere interactions where crops are a major feature of the land-surface (Osborne et al., 2015).

Photosynthesis in JULES-crop uses the same parameters and code as the natural Plant Functional Types (PFTs). There are two temperature parameters: $T_{\text{low}}$ and $T_{\text{upp}}$; these define the upper and lower temperature parameters for leaf biochemistry and photosynthesis within JULES (Clark et al., 2011) and are used to calculate the maximum rate of carboxylation of Rubisco (unstressed by water availability and ozone effects - $V_{\text{cmax}}$, with units of $\text{mol CO}_2 \text{ m}^{-2} \text{ s}^{-1}$) as defined in Clark et al. (2011) and reproduced here in Eq. 1. Equation 1 is the $V_{\text{cmax}}$ at any desired temperature.

$$V_{\text{cmax}} = \frac{n_{eff} n_l(0) \, f_{\text{T}}(T_{\text{c}})}{[1 + e^{0.3(T_{\text{c}} - T_{\text{upp}})}][1 + e^{0.3(T_{\text{low}} - T_{\text{c}})}]} \tag{1}$$

$$f_{\text{T}}(T_{\text{c}}) = Q_{10_{leaf}}^{0.1(T_c - 25)} \tag{2}$$

where $f_{\text{T}}$ is the standard $Q_{10}$ temperature dependence (given in Eq. 2) and $T_{\text{c}}$ is the canopy temperature. $n_{eff}$ represents the scale factor in the $V_{cmax}$ calculation (in units of $\text{mol CO}_2 \text{ m}^{-2} \text{ s}^{-1} \text{ kgC(kgN)}^{-1}$) and $n_l(0)$ the top leaf nitrogen concentration (in units of $\text{kgN (kgC)}^{-1}$). More details regarding the calculation of $V_{\text{cmax}}$ are provided in Clark et al. (2011) and Williams et al. (2017). $V_{\text{cmax}}$ is an important component in two limiting factors for photosynthesis; the Rubisco-limited rate and the rate of transport of photosynthetic products; Equation 1 shows the relationship between $V_{\text{cmax}}$ and temperature. GPP is used to describe the total productivity of a plant; this defines the gross carbon assimilation in a given time. Net Primary Productivity (NPP) is GPP minus plant respiration; NPP is used in the crop partitioning code and subsequently in the calculation of the yield in JULES. The nitrogen cycle in JULES cannot yet be used with the crop model, so in this study the same assumption is made as in Williams et al. (2017), that crops are not nitrogen limited.

The effective temperature (see Eq. 3) is the function that the model uses to relate air or leaf temperature to the cardinal temperatures that define a plant's development; these are the base temperature ($T_b$), maximum temperature ($T_m$) and optimum temperature ($T_o$) and are specific for each crop. Different models define their effective temperature function in different ways, for example Fig. 1 of Wang et al. (2017) provides a number of different possible definitions. The JULES definition described by Eq. 3 is most similar to type 4 given in Wang et al. (2017). Type 4 increases gradually towards the optimum temperature with a steeper decline from the optimum to the maximum. Other functions have no decline or a flatter top which can have different effects on the development of the crop. In JULES the cardinal temperatures and the 1.5m tile (i.e. air) temperature ($T$) are used to calculate the thermal time, i.e. the accumulated effective temperature ($T_{eff}$) to which a crop is exposed (Osborne et al., 2015). Table 3 summarizes the settings for these temperatures used in this analysis. The crop model integrates an effective

temperature over time as the crop develops through these stages, with the carbon partitioned according to the Development Index (DVI).

$$
T_{eff} = \begin{cases}
0 & \text{for} \quad T < T_b \\
T - T_b & \text{for} \quad T_b \leq T \leq T_o \\
(T_o - T_b)\left(1 - \dfrac{T - T_o}{T_m - T_o}\right) & \text{for} \quad T_o < T < T_m \\
0 & \text{for} \quad T \geq T_m
\end{cases}
\tag{3}
$$

The DVI is a function of the thermal time since emergence, therefore DVI=-1 is sowing, 0 is emergence and 1 is flowering. Maturity and therefore harvest occurs at a DVI of 2 (Osborne et al., 2015) under standard growth conditions but may be harvested earlier in other situations in the model (Williams et al., 2017). In reality the maturity date and the harvest dates are not usually the same date. The integrated effective temperature in each development stage is referred to as the thermal time of that development stage (Eq. 3 and Osborne et al. (2015); Mathison et al. (2018)).

Crop development can also be affected by the length of the day. However, in these simulations, as in (Osborne et al., 2015), this effect is not included. The thermal time is then used to calculate the rate of crop development or rate of increase of the Development Index, described by Eq. 4.

$$
\frac{dDVI}{dt} = \begin{cases}
\dfrac{T_{\text{eff}}}{TT_{\text{emr}}} & \text{for} \quad -1 \leq DVI < 0 \\
\left(\dfrac{T_{\text{eff}}}{TT_{\text{veg}}}\right) & \text{for} \quad 0 \leq DVI < 1 \\
\dfrac{T_{\text{eff}}}{TT_{\text{rep}}} & \text{for} \quad 1 \leq DVI < 2
\end{cases}
\tag{4}
$$

where $TT_{\text{emr}}$ is the thermal time between sowing and emergence, $TT_{\text{veg}}$ and $TT_{\text{rep}}$ are the thermal time between emergence and flowering and between flowering and maturity respectively. These are calculated either using a temperature climatology from the driving data and sowing dates from observations or using the method presented in Mathison et al. (2018) to create a reliable sowing and harvest dataset. The advantage of using the Mathison et al. (2018) method is that there is no missing data, which is often the case when using observed data. Whichever source of sowing and harvest dates are used, the aim is for the crop to reach maturity, on average by the harvest date. The sowing and harvest dates used in the simulations in this analysis are described in Sect. 4.

In order to simulate the characteristics of a typical sequential cropping location using JULES we have implemented modifications to both JULES-crop and the irrigation code. To simulate crops in sequence on the same gridbox, each crop must be completed cleanly so the second one can be sown accordingly. The specification of a latest harvest date (latestharvestdate) forces the harvest of the first crop regardless of whether it has reached maturity or not. The latestharvestdate is a safeguard built into the model, usually set to a date well after the expected harvest date. It is expected that when working properly, the first crop would be harvested well before this latestharvestdate and this safeguard should not be needed. If this safeguard is needed, the

user is alerted that the harvest has been triggered because the crop has not matured. The user therefore knows when the model is not working correctly and has some initial information, aiding the investigation into the nature of any problem. Although its use has been tested prior to implementation, the latestharvestdate was not needed in the simulations demonstrating this method here. The latestharvestdate safeguard is preferable to the simulation of a crop growing for an unrealistically long time and overlapping the next growing season. This is essential for the implementation of sequential cropping at a global or regional scale, where the model is forced to grow crops that are potentially unsuitable for a particular gridbox. This is more likely for global simulations, which typically simulate a restricted set of crop types and varieties. These modifications are controlled using the l_croprotate switch (see table 1). Therefore l_croprotate ensures the following:

– All crops are initialized at the start of a simulation so that they can be used later when they are needed within the crop rotation being modelled.

– If JULES is simulating a crop rotation, the user must supply a latestharvestdate so that the first crop is harvested before the second crop is sown (a latestharvestdate can also be specified without using l_croprotate).

The current JULES default for irrigation allows individual tiles to be specified (when frac_irrig_all_tiles is set to false) but the irrigation is applied as an average across a gridbox and therefore actually occurs across tiles. The flag set_irrfrac_on_irrtiles restricts the irrigation to the tiles specified by irrigtiles only (see table 1). This new functionality is needed because many locations that include crop rotations include crops that both do and do not require irrigation.

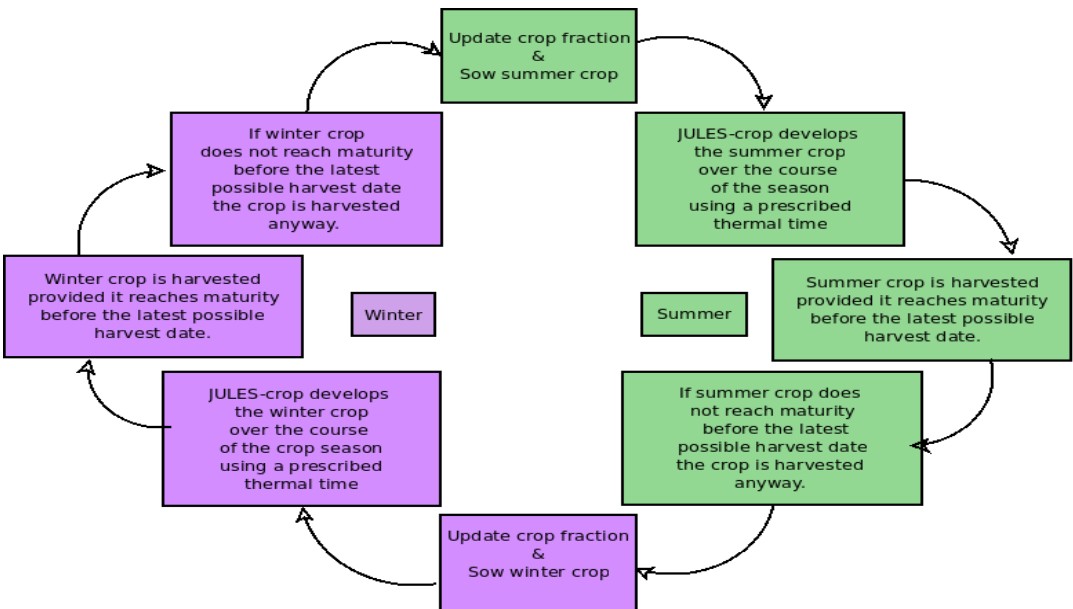

**Figure 2.** A flow chart showing the sequence followed to carry out the crop rotation in JULES. The first step (top green box) in the sequence is to update the first crop fraction, this occurs as or just before the first crop is sown.

## 3   Method for sequential cropping in JULES

The sequential cropping method implemented into JULES as part of this study is illustrated by the flow chart in Fig. 2 and described here using the Avignon site simulation. The Avignon site is a point run which is assumed to be entirely used to grow sorghum (from spring – late summer) and winter wheat (from winter – early summer). JULES updates the fraction of the site that is allocated to sorghum (winter wheat) just before the sowing date so that the appropriate crop occupies the whole of the site. The fraction of the site that is sorghum (winter wheat) is prescribed in the Avignon case using observed sowing and harvest dates. Once the fraction is updated the crop is sown, it then develops between the stages of: sowing and emergence, emergence and flowering and flowering and maturity.

It is recommended for sequential cropping to prescribe a latest possible harvest date for those instances where the crop does not develop quickly enough and therefore does not reach maturity before the next crop in the rotation is due to be sown (Sect. 2). In this study the latestharvestdate is set but never actually required for any of the simulations, which is the ideal scenario. The flow chart shown in Fig. 2 is equally applicable to the India simulations. Rice is therefore represented by the summer crop (green boxes) and wheat is represented by the winter crop (purple boxes). This method could be extended to include as many crops as occurs in a rotation at a particular location.

## 4   Model simulations

The description of the simulations is divided into two sections. Section 4.1 presents how the method is applied to a well observed site in order to describe and demonstrate how the cropping method works and evaluate it against observations at this location. The cropping system at the Avignon site is representative of a sequential cropping system, with sorghum planted during the summer months, followed by a winter wheat crop straight after. However, this site also represents a more irregular cropping pattern during some years, with a long fallow spell after the wheat crop and sorghum sometimes not sown until the following year. Section 4.2 applies the method to locations in Northern India where a more traditional sequential cropping system is commonly used, with a regular rotation between rice during the wetter kharif season and wheat during the drier rabi season. The parameter settings and switches used in JULES for the simulations in this study are provided in tables 1, 2 and 3. The Avignon and India simulations use the same settings wherever possible; these are provided in Table 1 (see Avignon settings and India settings columns).

The plant functional type (PFT) parameter settings are also broadly the same between simulations, with the majority of these from Osborne et al. (2015) and therefore based on natural grasses. The crops are different between the two sets of simulations with winter wheat and sorghum at the Avignon site and spring wheat and rice at the India locations. The PFT parameters used in this study that govern $V_{cmax}$: including the lower ($T_{low}$) and upper ($T_{upp}$) temperatures for photosynthesis, $n_{eff}$ and $n_l(0)$ are tuned to the maximum leaf assimilation expression from Penning de Vries et al. (1989) for each crop (see Table 2). These values are consistent with the wider literature (Hu et al., 2014; Sinclair et al., 2000; Olsovska et al., 2016; Xue, 2015; Makino, 2003; Ogbaga, 2014). The parameters, $\mu_{rl}$ and $\mu_{sl}$ are the ratios of root to leaf and stem to leaf nitrogen concentrations respectively; these are tuned to those given in Penning de Vries et al. (1989) to lower the plant maintenance respiration, which

was high in some of the initial simulations. The crop parameters are mainly from Osborne et al. (2015), with maize parameters used for sorghum (see Sect 4.1) except for the cardinal temperatures (see Table 3) which are from Nicklin (2012).

The calculation of the soil moisture availability factor (Beta, $\beta$, see Table 2) is different between the Avignon and India simulations. $\beta$ is based on the top 1.4m of soil, it is zero below the wilting soil moisture and one above a critical soil moisture, this is shown in Fig. 1 of Williams et al. (2018). In the Avignon simulations we assume a rectangular root distribution and the total depth of the rootzone $d_r$ to be 1.5 m, equivalent to the observed average maximum root depth over all of the years at the Avignon site. $\beta$ is then calculated using this maximum root depth together with the average properties of the soil. The India single gridbox simulations assume an exponential root distribution with an e-folding depth $d_r$ of 0.5 m because we do not have an observed root depth for these locations. In all simulations in this study, we adjust the parameters that affect the use of water by the plant so that the plants experience less water stress (this parameter is P0 and is set to 0.5 (Allen et al., 1998), see table 2). This is because water stress is not the main focus of this analysis, but the representation of soil moisture stress on vegetation is a known issue in JULES; this is the subject of a large international collaborative effort (Williams et al., 2018; Harper et al., in preparation). The individual simulations are described in more detail in Sect. 4.1 and Sect. 4.2 for the Avignon and India simulations respectively. The purpose of including Avignon is because it provides a wealth of observations for evaluating land surface models, where there is no equivalent site for South Asia. Observations of these fluxes show if the model is correctly representing the fluxes and coverage of the land surface. The purpose of including a simulation that does not use the crop model but approximates crops using grasses is to show how the model performs with the correct LAI and height, i.e. it is a clean test of the representation of leaf photosynthesis, stomatal conductance, water stress and leaf-to-canopy scaling within the model (these parts of the code are shared by both natural vegetation and crops).

## 4.1 Avignon site simulation

The Avignon "remote sensing and flux site" of the National Research Institute for Agriculture, Food and Environment (INRAE) described in Garrigues et al. (2015, 2018), provides a well studied location (France; 43°55'00.4"N, 4°52'41.0"E ) with several years of crop rotation data. We focus on the period with a rotation of just two crops: winter wheat and sorghum between 2005 and 2012. The aim of simulating the crops at this site is to demonstrate the new sequential cropping functionality in JULES and show how the implementation of sequential cropping affects the JULES crops simulated. JULES already contains parameterizations for wheat and maize. The wheat in JULES is the spring variety which is similar to the winter wheat crop that is grown at Avignon. Spring wheat does not require a vernalization period, which is a process usually needed for winter wheat varieties to achieve optimum yields (Griffiths et al., 1985; Robertson et al., 1996; Mathison et al., 2018). Vernalization is not explicitly implemented in JULES; therefore spring and winter wheat can be simulated interchangeably. The maize crop is a C4 crop that is similar to sorghum. Therefore we use these existing parameterizations rather than develop new ones. There are two varieties of sorghum grown at Avignon, the variety grown in 2009 is a fodder crop with a much shorter growing period and a larger LAI than the variety grown in 2007 and 2011. Therefore, the 2009 sorghum crop is planted much later in the year compared to the other two sorghum seasons (2007 and 2011) but harvested at a similar time.

The Avignon JULES simulations are driven using the meteorological site observations outlined in Section 5.1 and Garrigues et al. (2015, 2018) using a half hourly timestep. Irrigation is only applied to the summer crops, this is the sorghum crop at Avignon. The observed irrigation amounts are added to the precipitation driving data at the exact day and time they were applied to the crops (Garrigues et al., 2015, 2018). The irrigation and other settings governing irrigation are therefore not switched on in JULES for the Avignon site simulations (See Table 1, column 'Avignon settings'). We include simulations for the Avignon site where the crops are represented by grasses (Avi-grass) for comparison with the simulations that use the JULES-crop model. In the Avi-grass simulations the LAI and the canopy height are prescribed from observations in order to capture the growing seasons correctly without the crop model and the PFT parameters are adjusted to be the same as the crops. These Avi-grass simulations use the same photosynthesis and respiration calculation as JULES-crop, but this is not allowed to influence LAI as they do in the crop model. This allows the evaluation of the photosynthesis and respiration parts of the model, together with the water and energy fluxes, when the observed LAI and canopy height is used. In the Avi-grass simulations JULES is not modelling the crops as grasses but fixing some parts of the crops (LAI and canopy height) straight to observations. We also run two simulations that use the crop model; a single crop (Avi-single) and a sequential crop simulation (Avi-sequential). In both Avi-single and Avi-sequential simulations, the LAI and the canopy height are calculated by the model. The JULES total above ground biomass is calculated from the sum of the stem, leaf and harvest carbon pools for each crop. Observed sowing and harvest dates from Garrigues et al. (2015) are used to calculate the thermal time requirements for each crop represented in the simulations, these are provided in Table 4. During the periods between each crop, the ground is mostly bare (Garrigues et al., 2018). The only difference between the Avi-sequential and Avi-single simulations is that Avi-single only simulates wheat, therefore no sowing dates are provided for sorghum.

## 4.2 India simulations

The India simulations focus on the north Indian states of Uttar Pradesh and Bihar. These states are key producers of rice and wheat in South Asia and use a regular rice-wheat rotation that is prevalent in this part of India (Mahajan and Gupta, 2009). We include single gridbox simulations and a regional simulation. The single gridbox simulations are a selection of four locations, selected from across these two states in order to gain understanding of the model response to the variation in conditions across the two states. For each gridbox both single crop (referred to as India-single) and sequential crop (referred to as India-sequential) simulations are run. India-single and India-sequential are set up in the same way, with the only difference being that sowing dates are provided for just one crop. For consistency with the rest of the simulations only wheat is simulated in India-single. The single gridbox simulations enable a similar analysis to that described for Avignon (Sect. 4.1), while the regional simulation (this is only a sequential crop run) is a demonstration of the sequential cropping method being used at larger scales. For the regional simulation we assume that wheat and rice are grown in every gridbox across the two states and the crops are not limited by nutrient availability. The sequential cropping system in this region involves growing rice during the wet monsoon months and an irrigated wheat crop during the dry winter. In these simulations (both single gridbox and regional), wheat is only irrigated during its growing period and without applying limits due to water availability (this is referred to as

unlimited irrigation). The wheat varieties grown in India are spring wheat, which is the standard variety represented by JULES (see Sect. 4.1).

The locations of the selected gridboxes are shown on a map of the surface altitude for South Asia in Fig. 3a. The driving data used for these four simulations is from an RCM simulation run for South Asia for the period 1991–2007 as described below. Figure 3 (b, c and d) show a close-up view of the locations selected. Map (b) in Fig. 3 shows the average total monsoon precipitation for the 1991-2007 period, while (c) and (d) show the average minimum and maximum temperatures respectively to illustrate that these four gridboxes are representative of the climate of the wider Uttar Pradesh/Bihar region.

In both the single gridbox and regional simulations, JULES is run using a 3-hourly timestep using driving data from ERA-interim (Dee et al., 2011; Simmons et al., 2007) downscaled to 25 km using the HadRM3 regional climate model (RCM- Jones et al., 2004). This RCM simulation is one of an ensemble of simulations produced for the EU-HighNoon FP7 project for the whole of the Indian subcontinent (25 N, 79 E–32 N, 88 E) for the period 1991-2007. The HighNoon simulations are described in detail in previous publications such as Kumar et al. (2013) and Mathison et al. (2013, 2015). HadRM3 provides more regional detail to the global data with lateral atmospheric boundary conditions updated 3-hourly and interpolated to a 150 s timestep. These simulations include a detailed representation of the land surface in the form of version 2.2 of the Met Office Surface Exchange Scheme (MOSESv2.2; Essery et al., 2001). JULES has been developed from the MOSESv2.2 land surface scheme and therefore the treatment of different surface types is consistent between the RCM and JULES (Essery et al., 2001; Mathison et al., 2015). In the India single gridbox simulations the sowing dates are prescribed using climatologies calculated from the observed dataset, Bodh et al. (2015), from the government of India, Ministry of Agriculture and Farmers welfare. Thermal times are calculated using these climatological sowing and harvest dates from Bodh et al. (2015) and a thermal climatology from the model simulation as described in Osborne et al. (2015), the values used in the simulations here are provided in Table 5. In the regional simulation the thermal time requirements are estimated from the sowing and harvest dates provided by the Mathison et al. (2018) method to avoid problems with missing observed data. The settings used for the India simulations are provided in Table 1 (column 'India settings'). Plots of the regional ancillaries for each of rice and wheat are provided in Appendix C.

## 5 Observations

### 5.1 Avignon observations

The length and detail of the observation record at the Avignon site means it is an ideal site to demonstrate the method being implemented in JULES for simulating sequential cropping. High resolution meteorological data, important for the practicalities of running the JULES model is available on a half hourly basis; this includes air temperature, humidity, windspeed and atmospheric pressure at a height of 2m above the surface. Cumulative rainfall, radiation measurements and sensible ($H$) and latent heat ($LE$) fluxes are also available, with the latter flux measurements enabling the evaluation of the JULES fluxes. Cumulative evapotranspiration ($ET$) are derived from the half hourly $LE$ measurements. The observations for evaluating the model include soil measurements of soil moisture along with plant measurements including canopy height (measured every 10 days), above

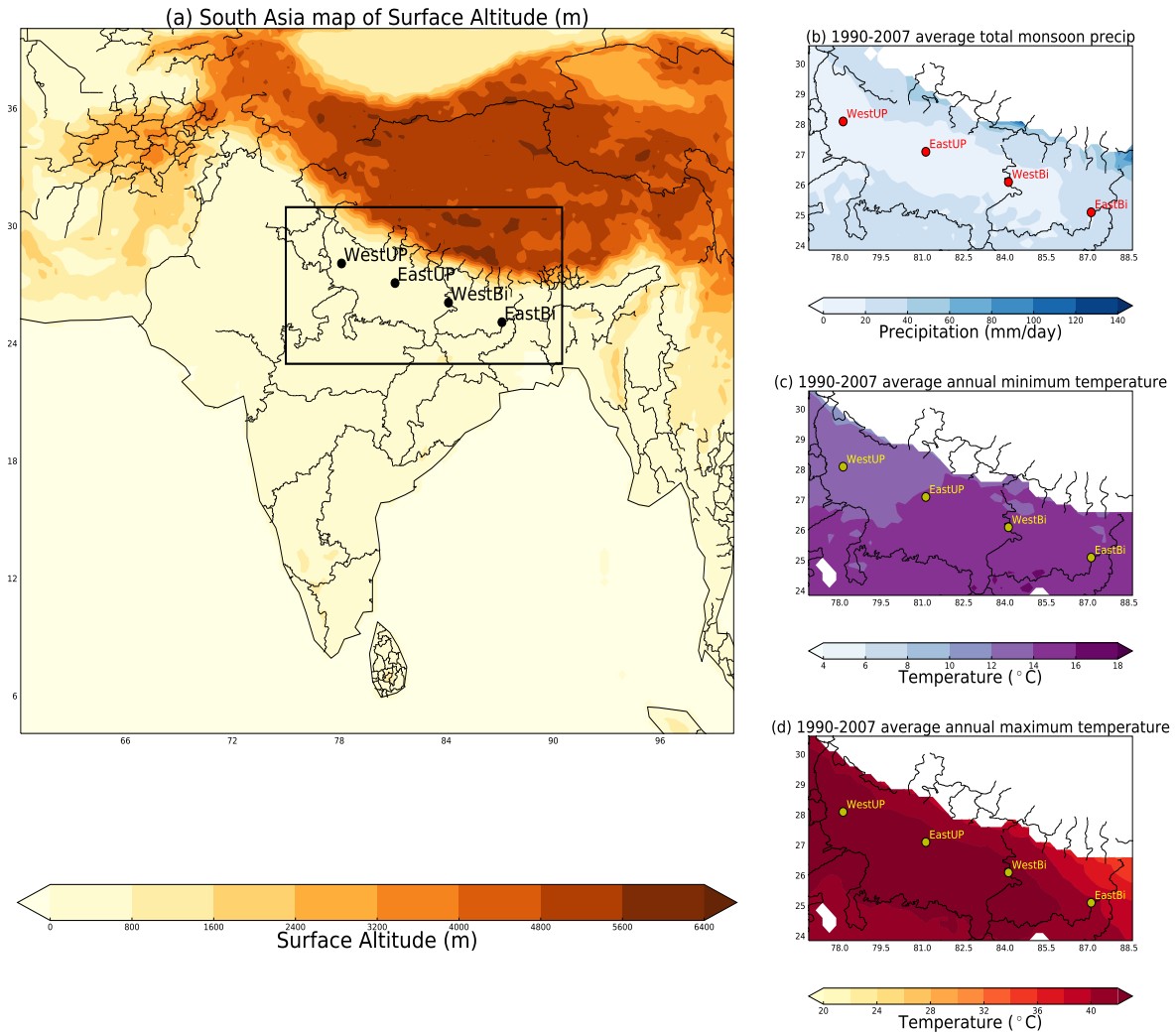

**Figure 3.** A map showing the location of the single gridbox simulations in the wider context of India on a map of the surface altitude (a) from the regional climate model that is used in the JULES simulations. The same locations are shown in three smaller maps (b,c,d) that zoom in on the two states of Uttar Pradesh and Bihar. Map (b) shows the total monsoon precipitation, map (c) shows the minimum temperature, and map (d) the maximum temperature averaged for the period 1991-2007.

ground dry weight biomass (taken at four field locations) and LAI; biomass and LAI are destructive measurements repeated up to six times per crop cycle (Garrigues et al., 2015). More information is documented in Garrigues et al. (2015) regarding the site and the observations available.

## 5.2 India observations

District level area and production data from the International Crops Research Institute for the Semi-Arid Tropics (ICRISAT, 2015) are used to calculate district level yields. These are then gridded at the resolution of the Era-Interim data (0.25°) to ensure that the scale of simulated and observed yields matched. We also show average crop yield observations for three, 5-year periods (Ray et al., 2012a) between 1993 and 2007 (1993–1997, 1997–2003, 2003–2007). Data from Ray et al. (2012a) is made available via Ray et al. (2012b). Ray et al. (2012b) are based on previous publications (Monfreda et al., 2008; Ramankutty et al., 2008). All the observations used include the period of the single gridbox simulations which are from 1991–2007. We show both of these datasets to highlight that there is a range in the estimates of yield for this region.

## 6 Results

### 6.1 Avignon site results

Avignon is characterized by a Mediterranean climate with a mean annual temperature of 287.15°K (14°C) and most rainfall falling in autumn (with an annual average of 687 mm). The Avignon timeseries of temperature (with a 10-day smoothing applied) is shown in Figure 4a and precipitation (10-day totals, which include actual irrigation amounts) in Fig. 4b (Garrigues et al., 2015). Figure 4 shows the fairly regular distribution of rainfall throughout the year (b) and the relative consistency of the annual temperature range for Avignon ( 26°), with only a brief cold snap in early 2012 having a much lower minimum. The results from the Avignon simulations are separated into sections; Sect. 6.1.1 and Sect. 6.1.3 examine the effect of sequential cropping, focusing on each of the first (crop growth and development) and second (carbon and energy fluxes) hypotheses respectively. We evaluate JULES against observations for crop growth and development in Sect. 6.1.2 and fluxes in Sect. 6.1.4.

### 6.1.1 The effect of including a secondary crop on the same field at Avignon on the growth and development of the primary crop

Figure 5 shows the timeseries of total above ground biomass (a), LAI (b) and canopy height (c) for Avi-sequential and Avi-single. Avi-grass are not shown as these follow the observed canopy height and LAI exactly as these values are prescribed in the simulations without crops. Figure 5 shows that the Avi-sequential crops are developing throughout the crop seasons with maxima of biomass, LAI and canopy height occurring at approximately the correct time for both crops. Therefore, the lack of vernalization in the model does not affect the simulation of winter wheat at Avignon. The total above ground biomass from JULES is plotted as a time series (dashed lines) for comparison with observations, which are provided as a single timeseries with the crop type confirmed from the timing of the observations. These are plotted alongside the model represented by purple asterisks (Fig. 5a). Figure 5 shows that there are only small difference between Avi-single (dotted line) and Avi-sequential (dashed), particularly for total above ground biomass (Fig. 5a) and canopy height (Fig. 5c). The LAI in JULES is typically more sensitive to changes in conditions than the biomass or canopy height, but even these differences between Avi-single and Avi-sequential LAI are small. The observations show that there was a sorghum crop during the summer immediately before the

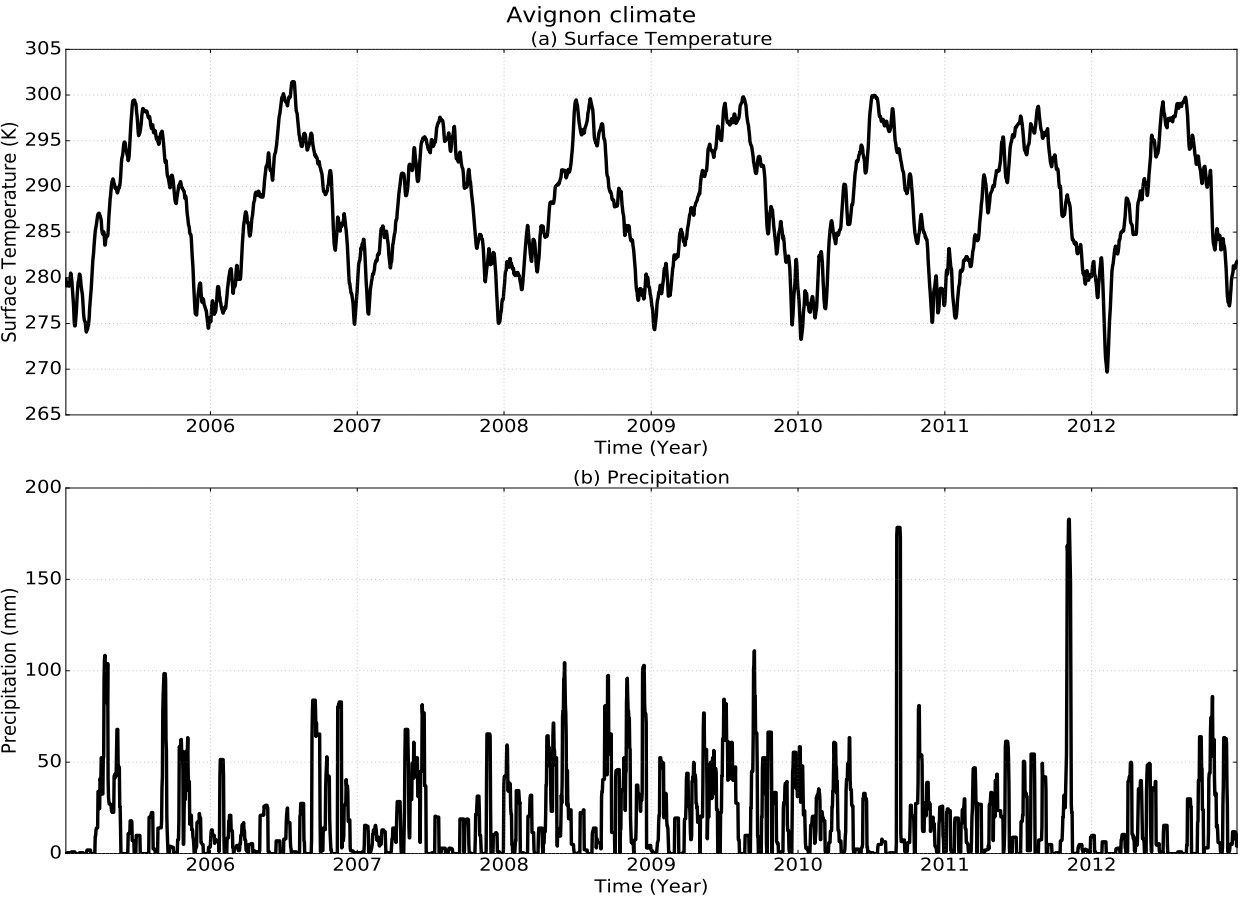

**Figure 4.** Timeseries of temperature (a) and precipitation which includes the observed irrigation amounts added at the exact day and time they were applied to the crops (b) at Avignon for the time period analysed (2005-2012)

2008 and 2012 wheat crop, during these years the LAI of Avi-single is slightly larger than the Avi-sequential and observations. In Avi-single, this sorghum crop is not present, which could affect the condition of the soils in the model at the time the wheat is sown. Overall, Fig. 5 shows that a similar wheat crop is simulated in both the single and sequential crop simulations, which is expected because both use the same parameterization to calculate crop development. Including a secondary crop on the same field in Avi-sequential does not really affect the primary crop growth and development, which means that Avi-single and Avi-sequential simulate a very similar primary (wheat) crop.

### 6.1.2 Model evaluation of JULES crop growth and development at Avignon

The shorter observed growing season for the fodder variety of sorghum grown in 2009 is shown by the red solid line in Fig. 5 b and c. No sorghum is simulated in Avi-single, so the red dotted line is zero and not visible on the plots of biomass (a), LAI

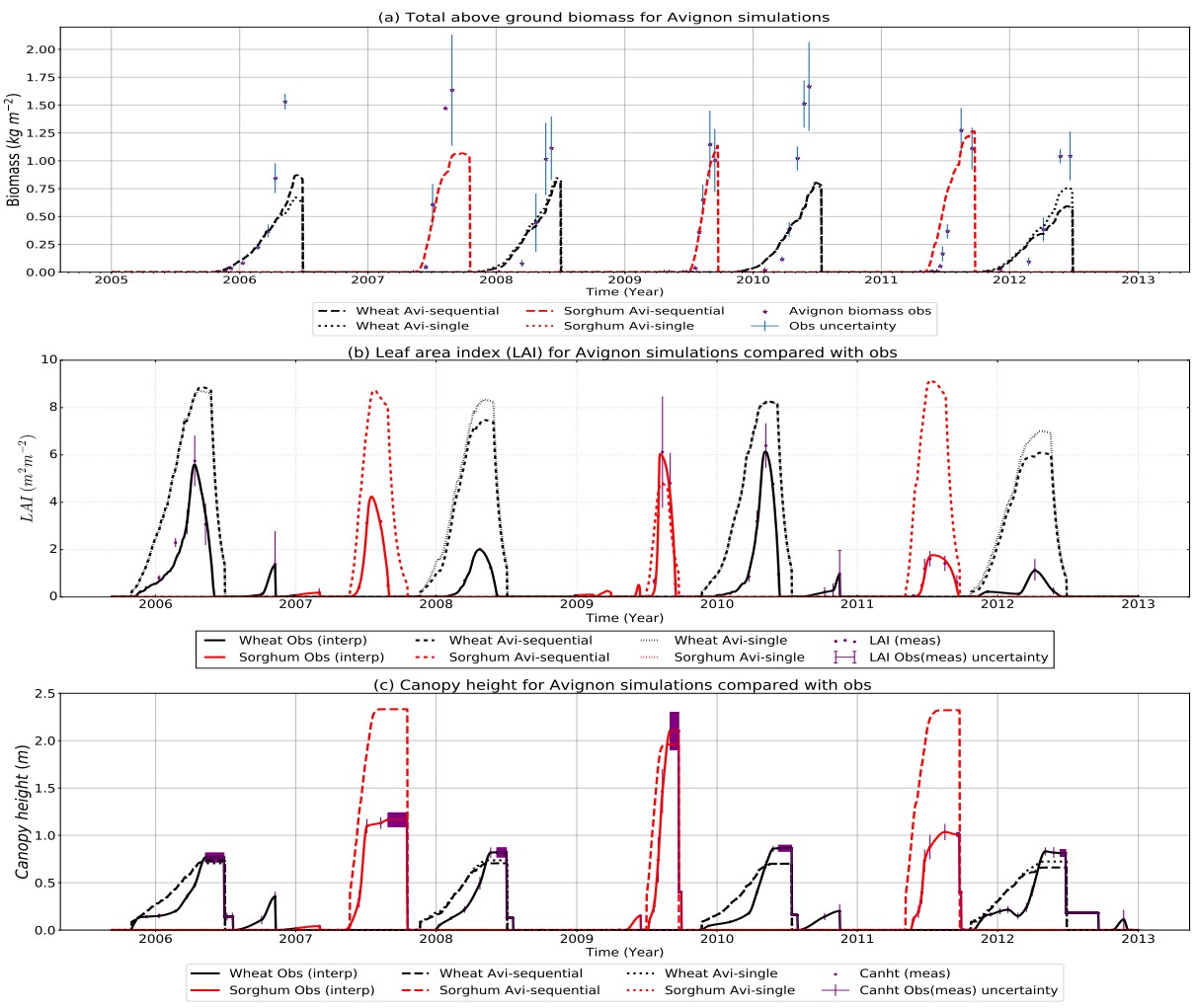

**Figure 5.** Timeseries of total above ground biomass (a), leaf area index (LAI) (b) and canopy height (c) for the Avignon site for wheat (black) and sorghum (red) for observations (solid lines) and simulations using the observed sowing and harvest dates: Avi-sequential (dashed) and the wheat only Avi-single (dotted) simulations for the period between 2005 and 2013 using observed sowing and harvest dates. Simulations with prescribed LAI and canopy height are not shown here as these follow the observed LAI and canopy height. The red dotted sorghum line in the Avi-single simulation is not visible because this is zero for the wheat only simulation. Observed above ground biomass in plot (a) shown by purple asterisks. The standard deviation of the measurements is shown to represent the uncertainty in the observations

(b) and canopy height (c) in Fig. 5. Avi-sequential simulates the 2009 sorghum season well, in terms of biomass (Fig. 5a), LAI (Fig. 5b) and canopy height (Fig. 5c); with relatively small differences between the simulations and observations maximum values (LAI: 1 $m^2$ $m^{-2}$ and canopy height: 0.1 m). In the 2007 sorghum season Avi-sequential overestimates the maximum LAI and canopy height by approximately two times the observations (see Fig. 5b and c) and underestimates the total biomass

(see Fig. 5a) by about 30 %. For the 2011 season the Avi-sequential sorghum biomass equals the magnitude of the observations; however, the maximum LAI is overestimated by four times in the model (similar to 2007) and the maximum canopy height is approximately two times the observed maximum. The canopy height is very close to observations for wheat in all four seasons; however, the wheat LAI is overestimated and the biomass is underestimated in all years. The two wheat seasons of 2006 and

2010 are closer to the LAI observations than 2008 and 2012, but the underestimation of the biomass is greater for these seasons. The increase in biomass for both crops through the start of the season follows the observations quite closely but in most years, especially for wheat, JULES-crop (using either single or sequential crops) does not accumulate enough biomass later in the crop season to reach the observed maxima.

### 6.1.3 The effect of including a secondary crop on the same field at Avignon on the energy and carbon fluxes

The largest difference between the GPP in Avi-sequential and Avi-single is during the sorghum growing period shown in Fig. 6a, where the GPP is zero for Avi-single because sorghum is not being simulated. The $H$ and $LE$ flux timeseries are shown in Fig. 6b and Fig. 6c respectively. All three simulations shown, Avi-sequential (blue line), Avi-single (cyan line) and Avi-grass (red line), largely follow each other closely, except during the sorghum growing period, which is not represented using the single crop version of JULES. During the sorghum growing period, Avi-single has a lower $LE$ and higher $H$ than

Avi-sequential and Avi-grass, which is due to the differences in land-cover represented. In Avi-grass and Avi-sequential, it is possible to include representation of the two observed crops and therefore represent the actual land-cover at the Avignon site, this is not possible in Avi-single. The inclusion of this secondary crop on the same field in Avi-sequential modifies the energy and carbon fluxes for the part of the year the secondary crop is being represented but the primary crop fluxes remain similar to those in Avi-single.

### 6.1.4 Model evaluation of JULES energy and carbon fluxes at Avignon

The peaks in productivity shown in the LAI in Fig. 5b are consistent with the two years (2006 and 2007) of GPP observations, shown by the black line in Fig. 6a. The 2006 wheat crop is represented in the GPP of all three simulations, although it is underestimated in all of them (Fig. 6a). The GPP in Avi-single is lower than both Avi-sequential and Avi-grass during the second half of the wheat growing period. The decline in GPP at the end of the 2006 wheat season is quite close to

the observations for the three simulations, with Avi-grass (red line) being slightly early and both the crop simulations; Avi-sequential (blue line) and Avi-single (cyan line) being slightly late. For the sorghum growing period, the magnitude and timing of the maximum GPP for Avi-sequential (blue line) are a good fit to observations. However, the increase in GPP begins slightly too early for Avi-sequential and slightly late for Avi-grass. The Avi-grass simulations slightly underestimate the maximum GPP during the sorghum season and it occurs a little later than observed (Fig. 6a). The decline in GPP at the end of the sorghum

season occurs at the same time as the observations for both Avi-grass and Avi-sequential. These results are quantified in Fig. A.1 with both Avi-grass (Fig. A.1a) and Avi-sequential (Fig. A.1b) showing a strong linear correlation, with r values greater than 0.8. The values for Avi-single (Fig. A.1c) are lower with an r value of approximately 0.5, this is because the observations

contain seasons with both sorghum and wheat, which is not possible in Avi-single. The statistics discussed here are summarised in the GPP row of Table 6).

A comparison between the simulated $H$ and $LE$ and observations are shown in Fig. A.2 and Fig. A.3 respectively. The RMSE and bias values for $H$ and $LE$ are given in Table 6, these are generally comparable to those from Table 5 in Garrigues et al. (2015), which are $LE$: rmse of 52.4 $Wm^{-2}$, bias of -11.8 $Wm^{-2}$, and $H$: rmse of 56.2 $Wm^{-2}$, bias of 17.6 $Wm^{-2}$.

The linear correlations for $H$ (shown in Fig. A.2) are strong for all three simulations with r values above 0.7, (Avi-grass in Fig. A.2a, Avi-sequential in Fig. A.2b and Avi-single in Fig. A.2c). The linear correlations for $LE$ are more variable between the three simulations with Avi-grass (see Fig. A.3a) having the strongest correlation (r value of 0.81), Avi-single (see Fig. A.3c) having the weakest correlation (r value of 0.64) and Avi-sequential (see Fig. A.3b) in between the two (r value of 0.73). The $H$ values for all three simulations and the $LE$ values for Avi-grass and Avi-sequential are comparable, but lower than those from Table 5 in Garrigues et al. (2015), which provides correlation values of 0.8 for $LE$ and 0.85 for $H$. The annual cycle of $LE$ and $H$ are shown in Fig. A.4, a and b respectively. Generally the seasonal cycles of $H$ and $LE$ are captured well in JULES (see Figure A.4 and the timeseries in Fig. 6, plot b and c). The annual cycle for $LE$ is close to observations in the first half of the year, but too high in the second half for Avi-grass and Avi-sequential. Avi-single is much too low, which explains its lower r value. Overall $H$ is closer to observations for all three simulations, however, the annual cycles show that both Avi-grass and Avi-single are a little too high and similarly Avi-sequential a little too low; explaining why the r values for this variable were much closer to each other.

## 6.2 India results: single gridbox simulations

The four India locations (gridboxes) selected for analysis in this study are shown on a map of South Asia in Fig. 3 (plot a) with smaller inset plots (b, c and d) focusing on the sequential cropping region being considered across the states of Uttar Pradesh and Bihar. Figure 7 shows the differences in the timeseries of the average precipitation (a), temperatures (b), and vapour pressure deficit (VPD) (c) at each of these four gridboxes with the different crop seasons emphasized by the different colour shading (yellow for wheat and pink for rice) on each of the plots. The temperatures rarely reach the low temperatures of the $t_{base}$ cardinal temperatures set in the model shown for rice (green) or wheat (orange) on Fig. 7 (b); however the high temperatures do exceed the maximum cardinal temperatures for these crops, especially those set for wheat. In general EastBi is cooler than the other locations in more of the years, with the two locations in Uttar Pradesh often being the warmest. The precipitation at each location is variable (see Fig. 7 plot a) with variation in the distribution of precipitation through the monsoon period which could be important for crop yields. Challinor et al. (2004), for example, found that in two seasons with similar rainfall totals, the distribution of the rainfall during the growing season strongly affected groundnut crop yield. There is also a clear seasonal cycle in the vapour pressure deficit (VPD), increasing toward the end of the wheat season and decreasing into the rice season. EastBi generally has the lowest VPD, with WestUP and EastUP usually the highest throughout the timeseries shown (see Fig. 7). These plots suggest that there is a gradual change in conditions from west to east across Uttar Pradesh and Bihar with increasing humidity and rainfall and decreasing maximum temperatures from west to east. The India single gridbox results are focused on the third hypothesis and for consistency with the Avignon results, are presented in a similar way. Section

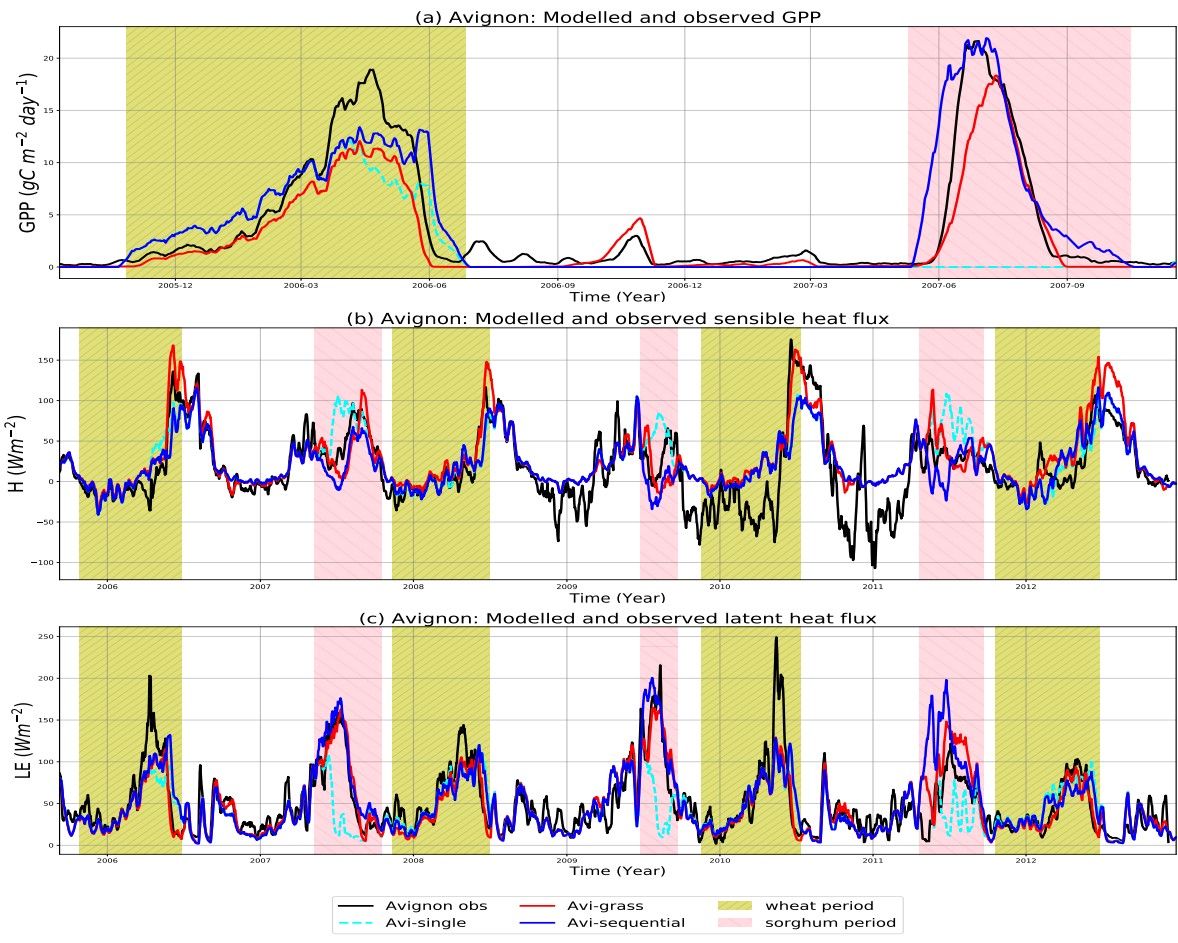

**Figure 6.** Timeseries of GPP (a), $H$ (b) and $LE$ (c) for the Avignon site compared with observations (black lines). For $H$ (b) and $LE$ (c), the whole period from 2005-2012 is shown, while GPP (a) shows the period 2005-2007 due to availability of observations. In the GPP plot only one complete winter wheat (yellow) and one complete sorghum season (pink) are highlighted. The following model simulations are also shown: Avi-grass (red), Avi-sequential (blue) and a wheat only, Avi-single (cyan). In each plot a 10-day smoothing has been applied to the daily data.

6.2.1 examines the effect of including a secondary crop in JULES on the primary crop growth and development, however, there are fewer observations available for these India locations, so JULES yields are compared with ICRISAT (2015) yields (see 6.2.2). The remaining sections examine the effect of the inclusion of a secondary crop on the primary crop for carbon fluxes in

Sect. 6.2.3 and energy fluxes in Sect. 6.2.4. Section 6.2.5 looks more closely at the effect of a secondary crop on soil moisture fields for this regular crop rotation.

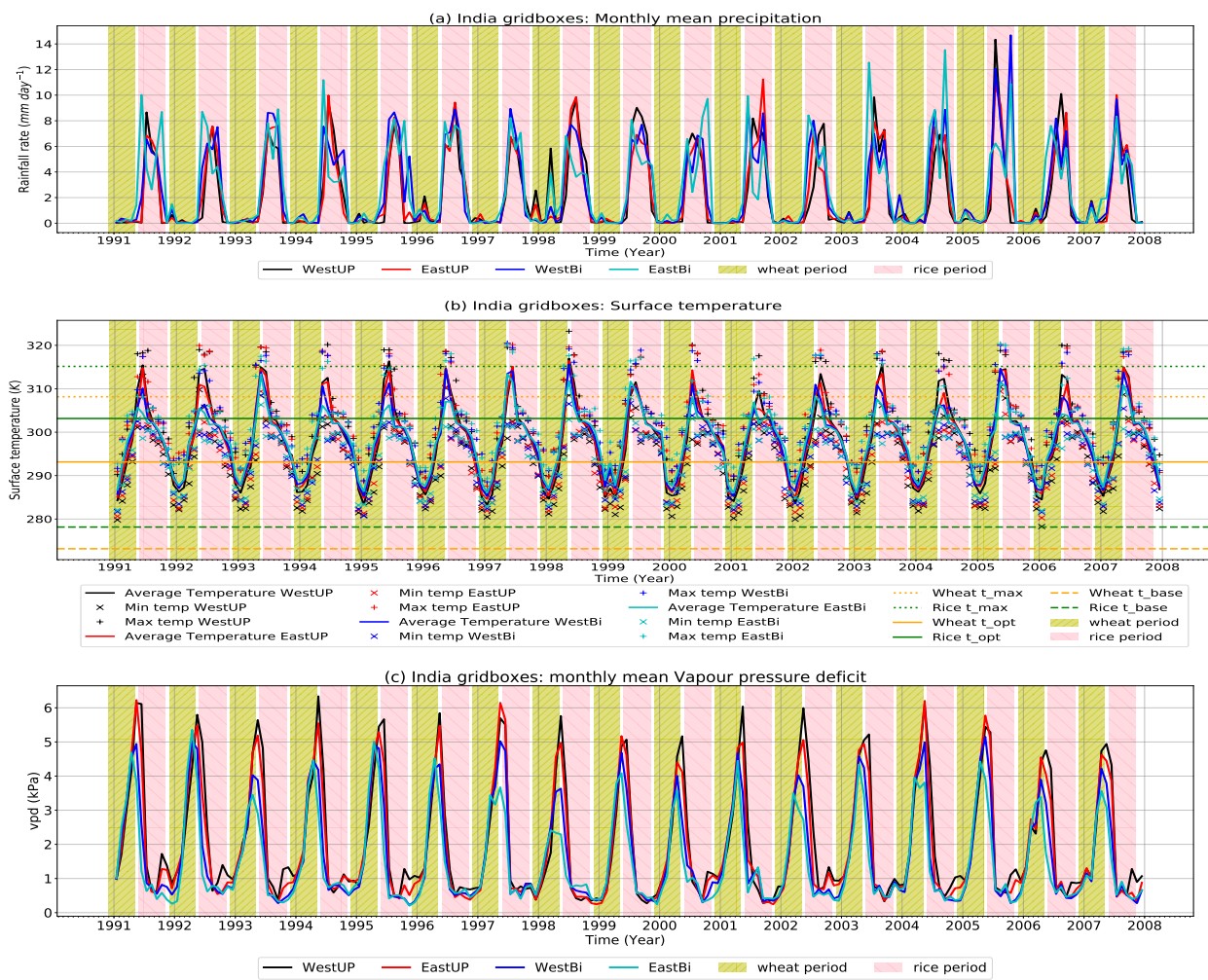

**Figure 7.** Timeseries of monthly precipitation (a), temperature (b), and vapour pressure deficit (c) at each of the India locations shown by the solid lines (WestUP-black, EastUP-red, WestBi-blue and EastBi-cyan). Plot (b) also shows the minimum ('x') and maximum ('+') temperatures for each of the locations for each month together with the JULES cardinal temperatures (horizontal lines) for rice (green) and wheat (orange): Max temperatures (dotted line), optimum temperatures (solid line) and base temperatures (dashed line).

### 6.2.1 The effect of including a secondary crop on the same field on the growth and development of the primary crop for the India locations

India-sequential simulations at these India locations produce both a rice and wheat crop yield (see Fig. B.1, with red representing rice and black representing wheat). JULES is therefore growing both wheat and rice at each of these locations within one growing season and is therefore simulating the sequential cropping rotation. The wheat crop in India-sequential is very similar to the wheat crop in India-single, Fig. 8 shows this in terms of yield, while Fig. 9 and Fig. B.3 show the LAI and the canopy height respectively. In each of the Figures showing LAI, canopy height and yield, it is only occasionally possible to distinguish between the India-single and India-sequential simulations, highlighting that the crop growth and development of the primary crop is not really changed by the implementation of a secondary crop at the same location in JULES.

### 6.2.2 Model evaluation of JULES crop growth and development for the India locations

It is useful to assess if the LAI and canopy height are plausible in these India gridbox simulations. This is important, especially where the results are to be applied to analysis of future water resource requirement, where an overestimation (underestimation) of size or leaf area for a crop could skew the results towards a higher (lower) resource requirement. In these simulations the canopy height (see Fig. B.3) for both rice and wheat at each location is between 0.5 and 0.7 m (see Fig. B.3) which is an expected value for a typical crop, as described in (Penning de Vries et al., 1989). Figure 9 shows the LAI for each of the four locations, indicating that the wheat LAI from JULES is between 5 and 7 $\mathrm{m^2m^{-2}}$ across the locations; this is also an expected value for a crop according to Penning de Vries et al. (1989). Rice LAI is lower (between 2 and 4 $\mathrm{m^2m^{-2}}$) with the lowest values for WestUP, slightly increasing from west to east locations. For WestUP particularly, rice (red solid line) has a small LAI (see Fig. 9) but it generates a yield (red asterisks Fig. B.1) that falls within the range of the observations for each year. However, wheat (black solid line) generates a LAI that is closer to expected values but a smaller yield compared with observations (see Fig. B.1, black asterisks).

Figure 8 shows the observed yields from ICRISAT (2015) compared against the India gridbox yields at each of the India locations, both India-single and India-sequential simulations are shown on this figure. Figure 8 shows how the yields change at each of the locations from west to east. The observed yields, particularly for wheat are larger to the west reducing to the east. In both simulations, JULES underestimates the western yields but tends to overestimate the eastern yields. This is confirmed in the timeseries of the harvest pool (solid lines) for each crop shown in Fig. B.1. Figure B.1 shows the India-sequential yield (asterisks), the average dataset from Ray et al. (2012a) (filled triangles) and the ICRISAT (2015) yield (filled circles) (as on Fig. 8). India-single is not shown in Fig. B.1 because it is indistinguishable from the wheat crop simulated in India-sequential, therefore the following yield biases are calculated using India-sequential. The inclusion of both observation datasets highlights the spread between yield estimates for this region. At WestUP, (asterisks on Fig. B.1, Fig. 8 black circles) the average bias between India-sequential and observations across both datasets is -0.13 $\mathrm{kg\ m^{-2}}$ for wheat and -0.064 $\mathrm{kg\ m^{-2}}$ for rice (Fig. 8 red circles). The average bias across both observation datasets is much smaller for the other locations with rice and wheat yields within the range of the observations for most years for both EastUP and WestBi (average bias across both crops at these

locations ranges from -0.07 to 0.02 kg m$^{-2}$). During the second half of the simulation the wheat yield is underestimated by India-sequential more often at EastUP but this is just the occasional year for WestBi and does not occur at all for EastBi. For EastBi the rice yields are often toward the top of the range provided by the two observed datasets but still within the range of the observations (see Fig. B.1); this gives on average a positive bias of 0.06 for rice and 0.02 for wheat.

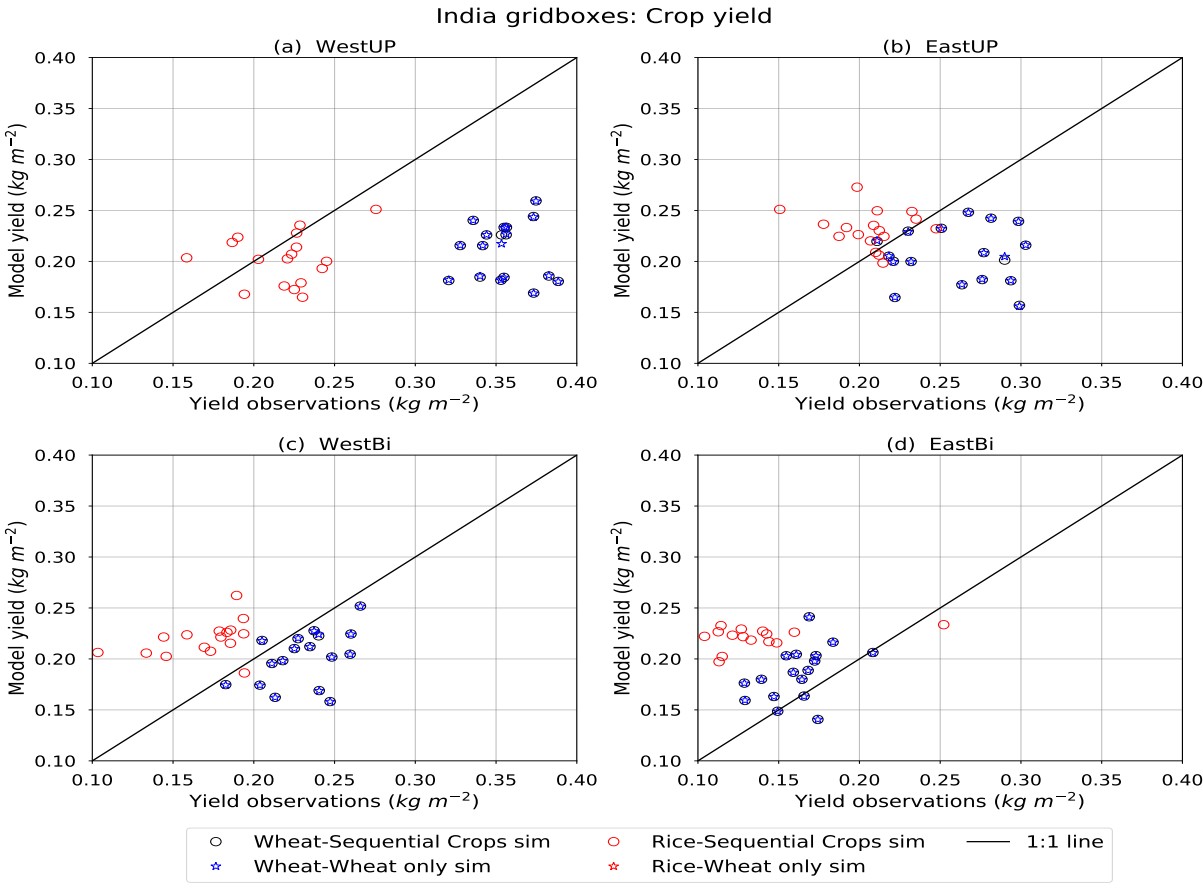

**Figure 8.** Scatter plot comparing the observed rice and wheat yields (ICRISAT, 2015) against JULES simulations at each of the India locations shown in Fig. 3, India-sequential shown by circles (rice in red and wheat in black). India-single shown by asterisks (rice shown in red and wheat shown in blue).

### 6.2.3 The effect of including a secondary crop on the same field on fluxes of carbon for the India locations

The timeseries and annual cycle of NPP (Fig. B.4c and Fig. 10a) and GPP (Fig. B.4d and Fig. 10c) for India-sequential show that each year there is a first peak for wheat (yellow stripes in the timeseries) and a secondary peak during the rice season (pink stripes in the timeseries). For wheat, which is the primary crop (grown during the last 25 days of one year and the first 140 days of the following year), the annual climatologies show that these carbon fluxes are very similar in both India-sequential (NPP -

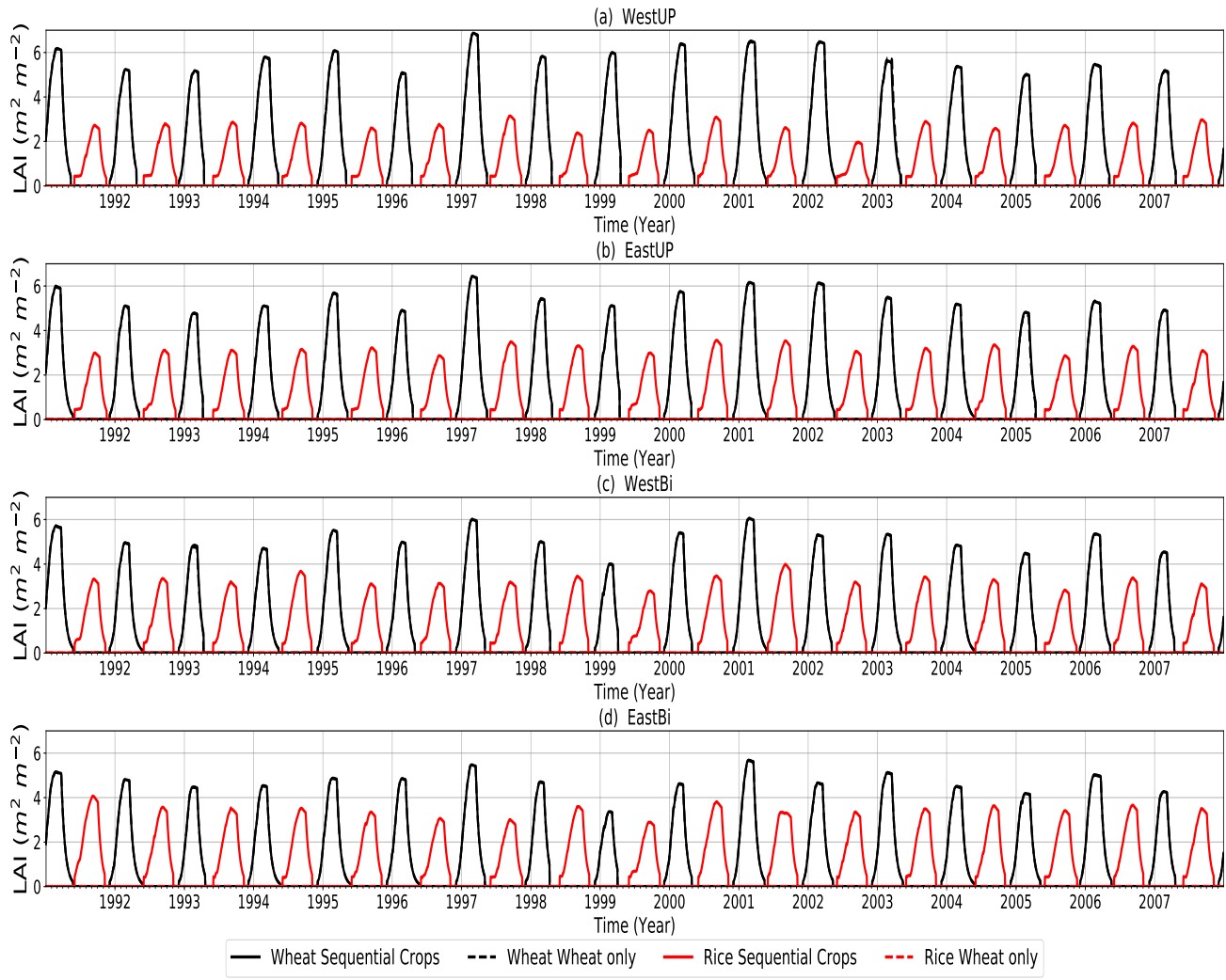

**Figure 9.** Timeseries of the LAI of rice (red) and wheat (black) at each of the India locations shown in Fig. 3. India-sequential is shown by solid lines and India-single is shown by dashed lines although they are indistinguishable from each other.

Fig. 10a and GPP - Fig. 10c) and India-single (NPP - Fig. 10b and GPP - Fig. 10d). However, the rice crop (between day 141 and day 330 of a typical year) is not represented in the India-single simulation, therefore the NPP and GPP go to zero during the rice season (Fig. 10b and d respectively). The similarity between the carbon fluxes during the wheat seasons in these two simulations is highlighted by the NPP (a) and GPP (b) shown for each of the India locations from west to east in Figures B.6, B.7, B.8 and B.9, which specifically compare India-sequential with India-single.

The annual climatologies show that wheat NPP begins to decline at around day 41 of the year in both India-sequential (Fig. 10a) and India-single (Fig. 10b), this is quite early in the season and may have a direct impact on the yield. This could be related to the way the carbon is partitioned to different parts of the plant or due to the ratio of thermal time of vegetation to reproduction in JULES. A short timeseries of India-sequential showing how carbon is partitioned to the different parts of the plant for wheat (black) and rice (red) are shown in Fig. B.10. The relationship between NPP and the yield is discussed further in Sect. 7.2.

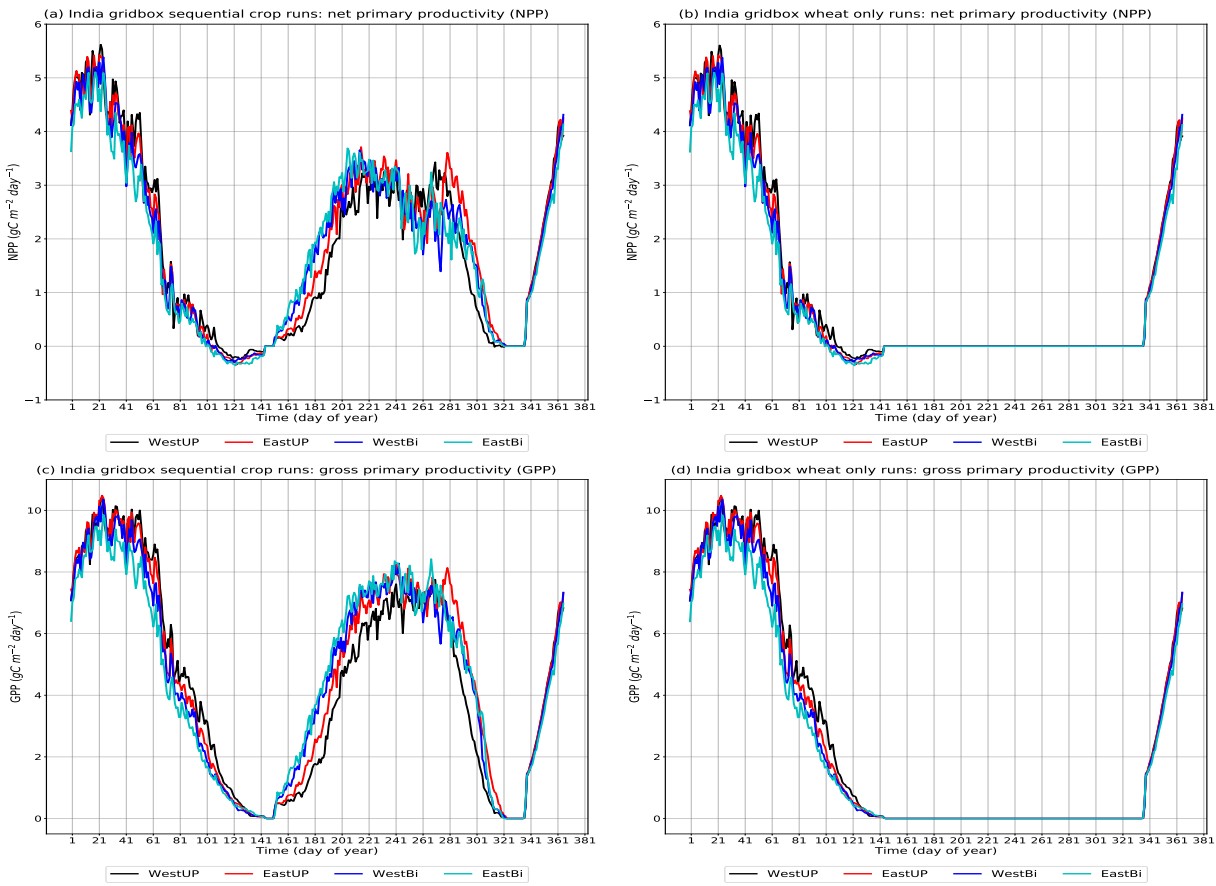

**Figure 10.** Annual climatologies (in day of year) of NPP for India-sequential (a) and India-single (b), and of GPP for India-sequential (c) and India-single (d). Each of the India locations shown in Fig. 3 is represented by a solid line of a different colour: WestUP - black, EastUP - red, WestBi - blue and EastBi - cyan.

### 6.2.4 The effect of including a secondary crop on the same field on energy fluxes for the India locations

Similar to the carbon fluxes, the energy fluxes ($LE$ and $H$) for each of the locations have a regular pattern each year (India-sequential fluxes are shown in Fig. B.4a and Fig. B.4b respectively). The annual climatologies for all the locations on one axes

are shown for India-sequential (Fig. 11a and Fig. 11c) and India-single (Fig. 11b and Fig. 11d) side-by-side. India-sequential and India-single are compared at each location in Figs. B.6, B.7, B.8 and B.9 for $H$ (c) and $LE$ (d) respectively; the energy fluxes for the two simulations at each location are largely indistinguishable from each other during the wheat season, common to both simulations. The main differences in the energy fluxes occur during the rice season, where India-single has a lower $LE$

5   (Fig. 11c and Fig. 11d) and higher $H$ (Fig. 11a and Fig. 11b) than India-sequential.

We investigate the changes in $LE$ further, using the evapotranspiration (g) and non-evapotranspiration moisture fluxes (h) for each location (see Figs. B.6, B.7, B.8 and B.9). The sequential crop method affects the individual components of the moisture flux, resulting in a larger component from the evapotranspiration flux and a lower component from non-evapotranspiration fluxes than India-single.

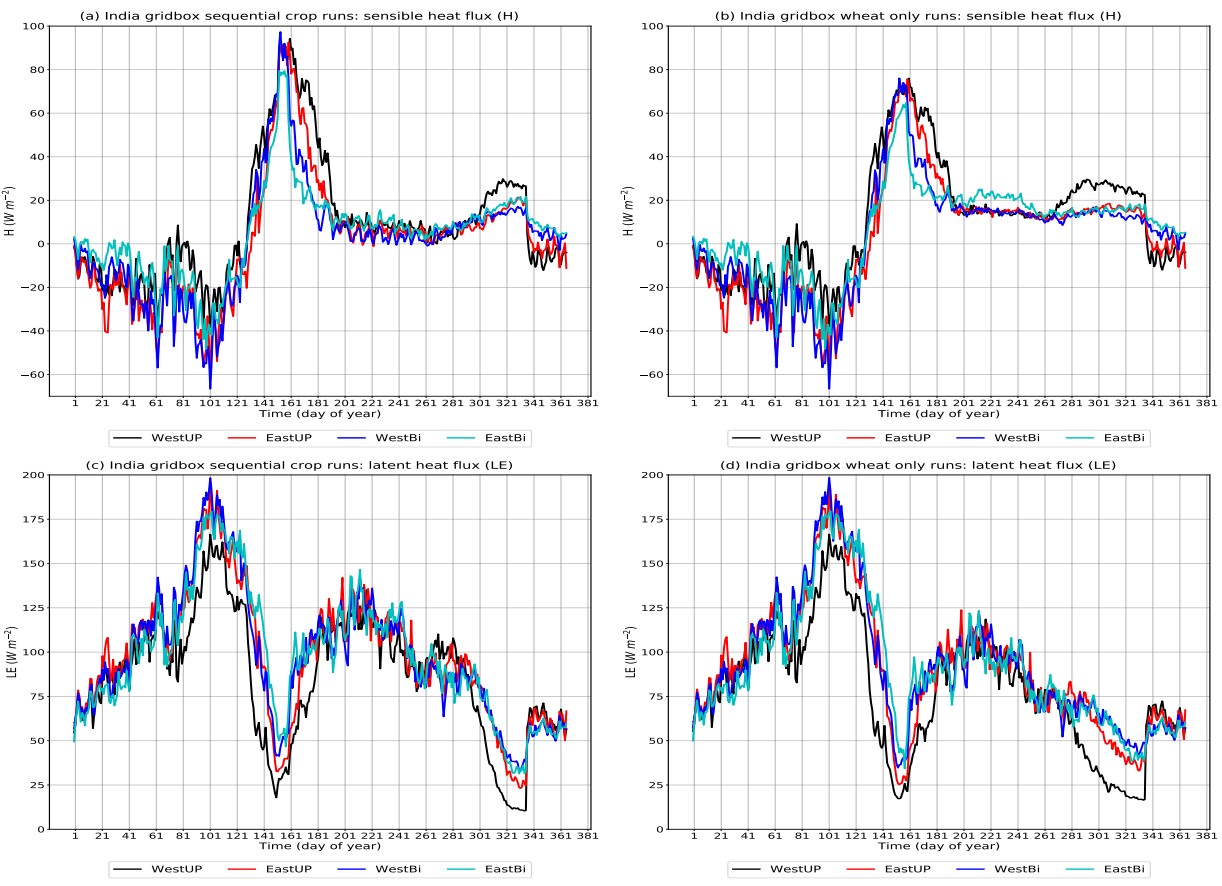

**Figure 11.** Annual climatologies (in day of year) of $LE$ for India-sequential (a) and India-single (b), and of $H$ for India-sequential (c) and India-single (d). Each of the India location locations shown in Fig. 3 is represented by a solid line of a different colour: WestUP - black, EastUP - red, WestBi - blue and EastBi - cyan.

### 6.2.5 The effect of including a secondary crop on the same field on soil moisture for the India locations

The India-sequential timeseries for $\beta$ (see Sect. 4) and the available soil moisture in the top 1.0 m in Fig. B.5 a and b respectively, show that most of the years are quite similar. There are occasional years which differ, for example, there are two short periods in 1998 and 2001, where EastBi has the lowest available soil moisture (See Fig. B.5b), these periods correspond with a lower monsoon rainfall at this location (see Fig. 7). We show the annual climatologies of these soil moisture fields for India-sequential and India-single side-by-side in Fig. 12. Each location is shown separately in Figs. B.6, B.7, B.8 and B.9 comparing only the sequential and single crop simulations for $\beta$ (e) and the available soil moisture in the top 1.0 m of soil (f).

The effect of sequential crops on the soil moisture availability factor ($\beta$) is similar to the carbon and heat fluxes, with India-sequential (Fig. 12a) and India-single (Fig. 12c) being consistent with each other for the wheat season. $\beta$ is mainly affected during the secondary growing period (rice) with India-sequential (Fig. 12a) showing a larger decrease in soil moisture availability over a shorter part of the year than India-single (Fig. 12b). Unlike $\beta$ and the fluxes of carbon and heat, the available soil moisture in the top 1.0 m is affected by sequential cropping throughout the year, with much larger fluctuations in India-sequential (Fig.12c) than in India-single (Fig.12d) even outside the rice season.

Between locations these moisture fields (Fig. 12) show that WestUP has the lowest available soil moisture and therefore $\beta$ value, suggesting this location is likely to be the most water stressed. It is also the western locations that are more affected by the implementation of sequential crops. WestBi on the other hand is least affected by the implementation of sequential crops, often with the highest $\beta$ and the most consistent available soil moisture in the top 1.0 m across the year of the four locations. This is consistent with the temperature and precipitation timeseries shown in Fig. 7, where the locations to the east are wetter and cooler than those to the west. This means there is more available soil moisture in the top 1.0 m for the eastern locations compared with the western locations.

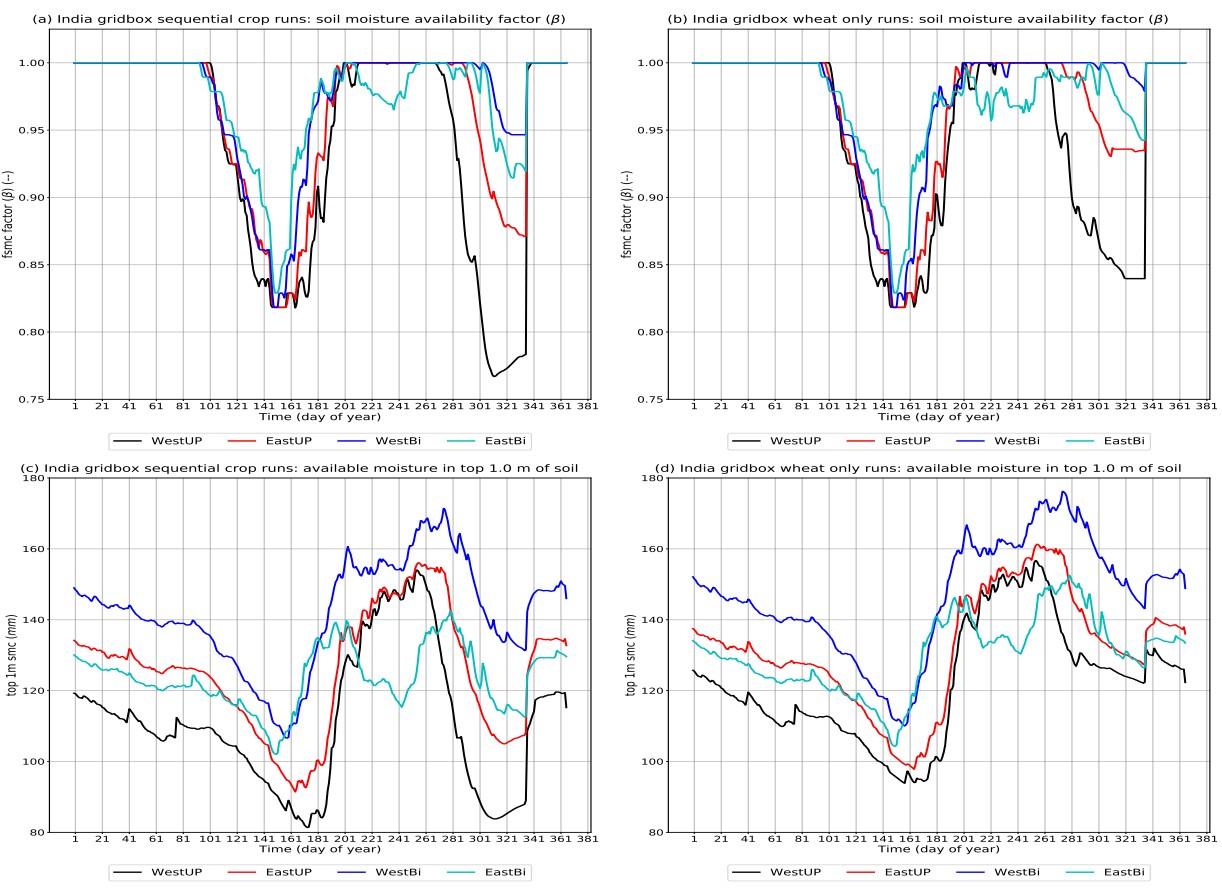

**Figure 12.** Annual climatology of moisture fluxes (in day of year), including; the gridbox soil moisture availability factor (beta) for India-sequential (a) and India-single (b); the gridbox available moisture in the top 1.0 m of soil for India-sequential (c) and India-single (d). Each of the India locations shown in Fig. 3 is represented by a solid line of a different colour: WestUP - black, EastUP - red, WestBi - blue and EastBi - cyan.

## 6.3 India regional results

The sequential cropping method is applied to a regional simulation as a demonstration that this is now possible. We show a small number of relevant results from this simulation, but these results are not intended as a comprehensive evaluation. Figures C.3 and C.4 show the average of the maximum annual LAI and canopy height for each crop for the regional simulation of the
rice–wheat rotation across Uttar Pradesh and Bihar between 1991-2007. As with the single gridbox simulations, the canopy heights are quite large for both rice and wheat, while the LAI is smaller, particularly for rice and to the west of the region. Comparisons with observations for this regional simulation show if the regional JULES yields produced using sequential crops are plausible. However, these comparisons should not be over-interpreted because the combined stresses that will be implicit in the observations, such as nutrient and water stress are not accounted for in this simulation. We also assume that rice and wheat
are grown in rotation everywhere, which will be an overestimation. The yield observations for the region are shown in Fig. C.5 for rice (a) and wheat (c); yields for both crops decrease from west to east. In general, the spatial distribution of simulated rice and wheat yields are quite close to the observations, although the rice yields in JULES appear to increase slightly from west to east.

The timeseries of the seasonal yields are shown in Fig. C.6 for rice (a, c) and wheat (b, d) area averaged for each state of
Uttar Pradesh (a, b) and Bihar (c, d). These show that there is considerable annual variability in the observations and the model yields. These observed yields have not been detrended, so improvements to land management practices such as irrigation or fertilization could account for increases in observed yields at the start of the timeseries. Averaging only for the Uttar Pradesh state area, the rice model yields (Fig. C.6a) are consistently lower than observed but the wheat model yields (Fig. C.6b) are much closer to observed until toward the end of the simulation; from 2000 to 2006 the model yields decline only recovering
as the simulation finishes. However, for the Bihar state area, the rice model yields are consistently higher than the observations (Fig. C.6c) in all but one year (1999) and wheat yields are on a par with observations for most years (Fig. C.6d).

## 7    Discussion

In Section 6 we present point simulations for Avignon (Sect. 6.1) and both single gridbox simulations (Sect. 6.2) and a regional simulation (Sect. 6.3) for India. The sequential crop simulations show that JULES is able simulate these different crops,
reproducing the crops in rotation at the expected times of the year for several successive years. In this section we discuss these results in more detail and what they mean for future applications of the method presented. For clarity we use the hypotheses to focus the discussion within the Sections on Avignon ( in Sect. 7.1) and India (in Sect. 7.2).

### 7.1    Avignon discussion

The Avignon simulations focus on a period between 2005 and 2013 where two crops were grown, it approximates winter
wheat using spring wheat and a c4 crop based on maize to represent sorghum. During this period two varieties of sorghum were grown, with a shorter season variety grown in 2009 compared with the other two years (2007 and 2011).

### 7.1.1 The effect of including a secondary crop on the same field at Avignon on the growth and development of the primary crop

There are three simulations presented; Avi-sequential, Avi-single and Avi-grass. The wheat season is represented in all simulations. The Avi-grass simulation has prescribed LAI and canopy height, which means that the crop development and growth is held to observations and is therefore not being controlled by the crop model. In the two crop simulations (Avi-sequential and Avi-single), the LAI and canopy height are calculated by the model, producing a similar wheat crop in each simulation in terms of LAI, canopy height and total above ground biomass. There are small differences in the LAI of the single crop simulation, but only for wheat seasons which occur immediately after a sorghum season that is not being represented. The very small differences between wheat in the two crop simulations is expected because the same crop code is used in each one, this illustrates that crop growth and development are unaffected by the inclusion of sequential cropping in this irregular sequential cropping system. Although Avi-sequential does not perfectly reproduce the observations at Avignon, it does capture the different seasons and crops at this site.

In the Avi-sequential simulations, the 2009 sorghum is the best year in terms of model performance, with a good approximation of the LAI, canopy height and biomass. The performance of JULES compared with observations using these existing spring wheat and maize parameterizations suggests that improvements are possible by developing winter wheat and sorghum type crop parameterizations in JULES. Garrigues et al. (2015) highlight that 2006 and 2008 have atypical rainfall during the wheat season, with 2006 being very dry (256 mm of rain during the wheat season) and 2008 being very wet (500 mm during the wheat season). Therefore, in 2008 Avignon received 73 percent of its annual average (see Sect. 6.1) during the wheat season alone; these differing conditions could explain the large differences in observed LAI and biomass between the two years (Garrigues et al., 2015).

### 7.1.2 The effect of including a secondary crop on the same field at Avignon on the energy and carbon fluxes

The representation of crops, either using the crop model (Avi-sequential) or using grasses (Avi-grass) has a similar effect on the surface fluxes, showing that the leaf level photosynthesis, stomatal conductance, water stress and leaf-to-canopy scaling within JULES with or without the crop model is approximated correctly when compared with observations. This code is used by the wider vegetation in JULES as well as the crop model. Fluxes of carbon and heat are generally captured well in JULES with all the simulations generally following the observations through the timeseries. There are regular differences in the fluxes during the sorghum season of the Avi-single simulation, which only includes wheat. The Avi-single simulation cannot simulate the correct land cover during the sorghum season, which leads to $LE$ that is too low and $H$ that is too high during the sorghum season. In the Avi-sequential simulation, the representation of the land-surface at Avignon is closer to the the observed land-cover because bare soil, wheat or sorghum can be reproduced within the simulation when they have been observed. This analysis shows that the main heat and carbon fluxes are changed by the implementation of sequential cropping, with the main differences occurring during the period where a crop is observed, but is not usually represented in the simulation.

Sequential cropping would previously have represented the Avignon site using one of two methods: The first, simulating only one crop (here represented by India-single), with the rest of the year represented by bare or almost bare soil. The second option is the approach used by ISIMIP (Warszawski et al., 2013, 2014); this uses a fraction of each gridbox to simulate each crop. In the first single crop option, the water, carbon and energy fluxes are incorrect for the season that is not explicitly simulated, e.g. the sorghum season in the single crop case shown here. In the second option, two crops are modelled in the same gridbox but on a small area of the gridbox, thereby allowing a yield for each crop to be obtained for each gridbox; this can be post-processed to give larger scale yields. The ISIMIP approach is effective if yields are the primary focus, but it is not applicable for understanding the use of water resources across different seasons. The real benefit of the sequential cropping capability is that it is able to represent the observed land-cover in a single simulation, which generally improves the simulation of water, energy and carbon fluxes, as is shown for Avignon compared with observations. The simulation of individual crops is very similar between the sequential and single crop methods because they use the same code, however, including sequential crops enables a continuous simulation for multiple years and seasons, which allows a coherent usage of resources between the different seasons. This means the demand for resources can be estimated across years and seasons, which includes memory in the model of the conditions during the previous season and the resources used by a previous crop.

This site at Avignon is a valuable resource that will help develop and test future specific parameterizations for these crops and others that are also grown at this site. It is hoped that the suite that runs JULES at Avignon with and without sequential crops could become one of the 'golden' sites that is referred to in Williams et al. (2018) and thereby aid future development of JULES and other land surface and crop models to include a sequential cropping capability.

## 7.2 India discussion

The India single gridbox and regional simulations are designed to provide similar representations of the rice–wheat crop rotation for the Uttar Pradesh and Bihar region. These two North Indian states are amongst the main producers of these crops using this rotation. The aim of the single gridbox simulations is to conduct a similar analysis to that provided for Avignon in Sects. 6.1 and 7.1. The aim of the regional simulations is to show that this method can be used at larger scales. Both types of India simulations are for the period 1991–2007. India-sequential is run in a regional simulation for the two states and as a single gridbox run for four gridboxes across these two states. We assume that irrigation only occurs during the wheat season with no irrigation during the rice season, because rice is grown during the wettest part of the year. However, some irrigation may occur during monsoon breaks; therefore it would be useful to develop JULES to recognise a break in monsoon rainfall and trigger irrigation of rice if the monsoon break is accompanied by a drop in soil moisture. Also, temperatures in JULES cannot damage any of the crops being modelled either by being too high, too low or not low enough. Future work using JULES-crop would also benefit from developments to enable the model to simulate when and where crops suffer from heat stress or problems with soil nutrients, pests and diseases and for these to be able to have an impact on crop yields. The simulations shown here consider a small number of rotations, crops and regions. However; different varieties and types of crops; the timings of sowing and harvesting; together with many possible irrigation options can have a large impact on the model results. This is an important consideration for future work and should be investigated fully when applying this method to new areas. The India point gridbox

simulations provide some interesting points for discussion in the following sections; Crop growth and development in these simulations is discussed in Sect. 7.2.1 and the fluxes are discussed in Sect. 7.2.2.

### 7.2.1 The effect of including a secondary crop on the same field on the growth and development of the primary crop for the India locations

India-sequential and India-single are run for the same four points as a comparison between the two crop methods. Wheat is the common crop to both of the simulations and produces an almost identical wheat crop in the single and sequential crop simulations in terms of LAI, canopy height and yield. The India single gridbox simulations show that the implementation of sequential cropping does not change the crop development or growth for the primary crop, in this case wheat.

Across the two sets of single gridbox simulations there are important differences between the four points. The observed yields for both rice and wheat are higher in the west of Uttar Pradesh and reduce as you go east across the region to Bihar. WestUP has the least available soil moisture, lowest rainfall and higher temperatures than the other locations, yet the observed yields and therefore the actual productivity are higher than for example, EastBi. The observed yields at EastBi are the lowest of the four locations, where the cooler wetter conditions should be more conducive to achieving higher yields; these are neither observed nor modelled. A combination of factors may lead to the models underestimating the WestUP wheat yields (compared to EastBi) and being much closer to observed yields in Bihar. One explanation is likely to be the differing management practices between the two states of Uttar Pradesh and Bihar. Uttar Pradesh is characterized by high agricultural productivity with effective irrigation systems (Kumar et al., 2005) and early adoption of new management practices (Erenstein and Laxmi, 2008). Bihar on the other hand has lower agricultural productivity, farms tend to be smaller and more fragmented, irrigation systems are less effective (Laik et al., 2014) and adoption of new technology is also slower due to the lack of available machinery (Erenstein and Laxmi, 2008). Yield gap parameters are included in many crop models in order to account for the impact of differing nutrient levels, pests, diseases and non-optimal management (Challinor et al., 2004), thus explaining the difference between potential and actual yield under the same environment (Fischer, 2015). This is not included in these simulations.

An alternative explanation for the difference in model yields from west to east could also be that at the western locations, the humidity is lower (higher VPD) and the temperatures are higher; these conditions may provide another contributory factor for the model underestimating the yields there. The humidity in the simulations could be lower than in reality for two reasons: first we are running JULES in standalone mode. This means that the land-surface and therefore the crop is unable to influence the atmosphere through evaporation because the humidity is prescribed by the driving data at each timestep. Second the driving data is from an RCM that does not include irrigation (Mathison et al., 2015) so the humidity in the driving data is not modified by evaporation due to irrigation. We are therefore missing the part of the water cycle that allows evaporation from the surface to affect the humidity. This region is intensively irrigated (Biemans et al., 2013) which means that there is a significant contribution from the evaporation due to irrigation and the recycling of water into precipitation (Harding et al., 2013; Tuinenburg et al., 2014) that cannot be accounted for here. Tuinenburg et al. (2014) estimate that as much as 35 % of the evaporation moisture from the Ganges basin is recycling within the river basin. We hypothesize that the VPD may be too high in our forcing data and this could be affecting the model yields at this location (Ocheltree et al., 2014).

The yields in the model are also affected by the other choices made in setting up the model. For example, the stage at which leaf senescence begins is given by a user defined parameter in JULES (sen_dvi_io). In these simulations this is set to be when the DVI is equal to 1.5. At this stage the carbon from the leaves starts to be remobilized to the harvest pool (Fig. B.10); which consists of both the reproductive parts of the plants and the yellow leaves (Williams et al., 2017). During this senescence period, the plants continue to respire but as the leaves are lost, photosynthesis reduces. This results in a decline in NPP, which begins early in the season. In addition the allometric coefficients that control the partitioning of carbon to the different parts of the crop in JULES are currently those from Osborne et al. (2015); it is possible that the results could be improved for South Asia if these were tuned to more appropriate values for the crops there.

### 7.2.2 The effect of including a secondary crop on the same field on the energy, carbon fluxes and soil conditions for the India locations

The wheat season is common to both India-sequential and India-single simulations, during this season the carbon fluxes are indistinguishable from each other. The main differences between the carbon fluxes in these simulations occur during the rice season, which is represented in India-sequential but is not being modelled in India-single. The rice season is not represented in India-single, which means India-single has an NPP and GPP of zero for this period. The largest differences between the heat fluxes in India-sequential and India-single also occur during the rice season. The effects of including sequential crops on the sensible and latent heat fluxes are consistent with the differences shown for the Avignon simulations, with a higher $H$ and lower $LE$ in the India-single than in India-sequential.

In the irregular cropping rotation at Avignon there is sometimes a considerable length of time between crops, which means there is an extended period of bare soil. It is therefore unclear if the soil moisture is affected by sequential cropping or by evaporation from bare soil at Avignon. However, India represents a regular cropping system without these long periods of bare soil, which means changes in soil moisture are more likely to be due to the implementation of sequential cropping. In the India-single simulation the available soil moisture is less variable than the India-sequential simulation, this is not limited to an individual crop season but is evident throughout the year. In general, this increased variability is more pronounced for the more arid western locations, with the wetter eastern locations being less affected. Sequential cropping is therefore modifying the soil moisture availability across all the crop seasons.

## 8    Conclusions

In this paper we describe and demonstrate a new development for JULES enabling more than one crop to be simulated at a given location during a particular growing season, thereby including a sequential cropping capability. This is an important development, allowing more accurate representation of land use and surface coverage in regions where two or more crops are grown in rotation and another step towards being able to include JULES-crop in earth system and climate models. This development does not modify the crop model, therefore the crop growth and development remain the same between single and sequential crop simulations and for regular or irregular crop rotations. However, for these types of crop rotations, the

inclusion of sequential crops in JULES does modify the fluxes of carbon and heat, particularly changing the contribution from the different components of the latent heat flux. For regular rotations, sequential crops also affects the availability of soil moisture throughout the simulation. Including the correct land-use and surface coverage in models means that the simulations can produce more realistic fluxes of carbon, water and energy; these are important for understanding the impacts of climate

change. The continuous simulation of all crops throughout the year also provides a more complete picture of the total demand for water resources which is important for climate impacts assessments. There are relatively few models that are able to simulate sequential cropping, but there is a growing need as more regions of the world adopt this cropping system as a viable way of adapting to climate change (Hudson, 2009). We demonstrate the method and evaluate its impact for a site in Avignon; this a site that has grown crops in rotation for several years and therefore has a lengthy and detailed observation record. We use this

site to simulate a winter wheat–sorghum rotation in JULES approximated using spring wheat and maize. We apply this same method to four locations that use a regular sequential cropping system for the rice–wheat rotation in the northern Indian states of Uttar Pradesh and Bihar, in order to inform its implementation for a regional simulation of South Asia.

We show that JULES can simulate two realistic crops in a growing season both at Avignon and across Uttar Pradesh and Bihar. At Avignon, the maxima of LAI, canopy height and biomass occur at approximately the correct times of the year

and the observed GPP and energy ($H$ and $LE$) fluxes are reproduced, correlating well with observations (r values of above 0.7). However, the magnitude of the biomass for wheat is underestimated and LAI is overestimated compared with Avignon observations. In general, there are only small differences between using the crop model and using grasses to represent the crops at this site, indicating that JULES-crop can reproduce the LAI and canopy height well enough to compare well with the observed surface fluxes. The representation of crops at Avignon could be improved by including crop specific parameterizations

of winter wheat and sorghum in the model, although sorghum would probably require two different sets of parameters for a significant improvement because the two varieties grown at the site are so different.

The sequential cropping system is used widely in the Tropics, especially regions such as Pakistan, India and Bangladesh. We run a regional simulation of JULES with sequential cropping for the Indian states of Uttar Pradesh and Bihar and also for four locations within these states; these are two of the main producers of rice and wheat in India and use of the rice–spring

wheat rotation is prevalent in this region. This region is highly variable, both in terms of temperatures (ranging from 7 to 52 ° C) and rainfall (between 0 and 15 $\mathrm{mm\ day^{-1}}$) with these locations showing a cooling moistening trend from west to east making conditions for growing crops very different across a relatively limited area. JULES produces both a rice and wheat crop across the region and for each of the four gridbox simulations, with yields for the locations in the cooler, wetter east of the region closer to observed yields than those in the warmer drier west. We propose two possible reasons for this difference,

although in reality both could be contributing factors. One explanation for the differences in observed yields between WestUP and EastBi is the differing management practices between the two states of Uttar Pradesh and Bihar. The western locations are typically more effective at adopting new technology and therefore have higher yields than the eastern locations. This difference from west to east may therefore be reduced by a yield gap parameter. Alternatively ensuring that irrigation is represented in the forcing climate data used to drive JULES may reduce the differences between the observed and model yields at WestUP.

The lack of irrigation in the forcing data, could reduce evaporation from the surface. Tuinenburg et al. (2014) highlight that this makes a considerable contribution to the overall moisture budget for South Asia.

The work presented here has shown that sequential cropping is an important addition to JULES, providing a closer representation of the land surface where crops are grown in rotation. These changes are part of the JULES code from version 5.7 although the analysis shown here use an earlier version of JULES (vn5.2). This analysis has provided valuable information for using this sequential cropping method for future larger crop simulations in the form of an example regional simulation. Model intercomparison projects such as AgMIP (Rivington and Koo, 2010; Rosenzweig et al., 2013, 2014) and ISIMIP (Warszawski et al., 2013, 2014) have hugely benefited the crop and land-surface modelling communities by accelerating development and understanding of land surface models. On the basis that this cropping system is likely to be a feature of the future land-surface, not just in the tropics but globally as an adaptation to climate change, we encourage other modelling communities to develop their models to include a sequential cropping capability so that future model intercomparisons can include this and find ways to improve it further.

*Code and data availability.* The JULES model code used in this paper is available from the Met Office Science Repository Service on registering: https://code.metoffice.gov.uk/trac.

The version of the model used in this analysis is an enhanced JULESvn5.2, this branch is available from this link: https://code.metoffice.gov.uk/trac/jules/browser/main/branches/dev/camillamathison/vn5.2_croprotate_irrigtiles.

The developments contained in this branch are now implemented into the trunk of JULES from version 5.7. The regional climate model datasets used will hopefully be available via the Centre for Environmental Data Analysis (CEDA) catalogue. It is hoped that the Avignon rose suite will also be made available in order to aid future model development.

# 9 Appendix A: Avignon comparison

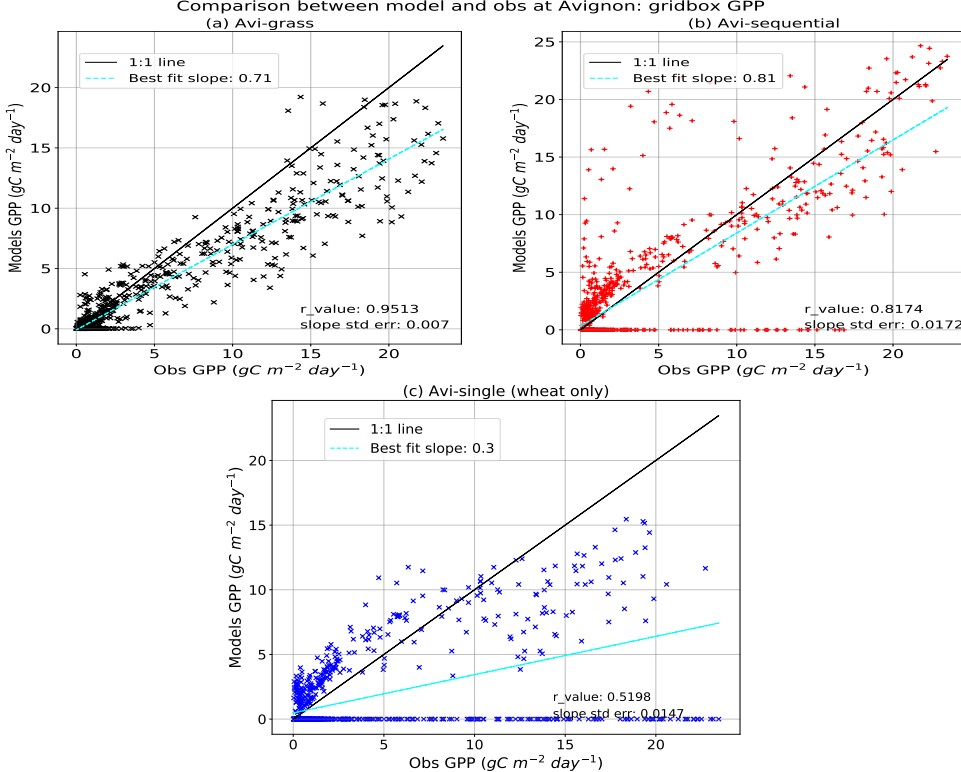

**Figure A.1.** Comparison of Observed GPP at the Avignon site against the modelled GPP between 2005 and 2008 for Avi-grass (a) and Avi-sequential (b) and Avi-single (c).

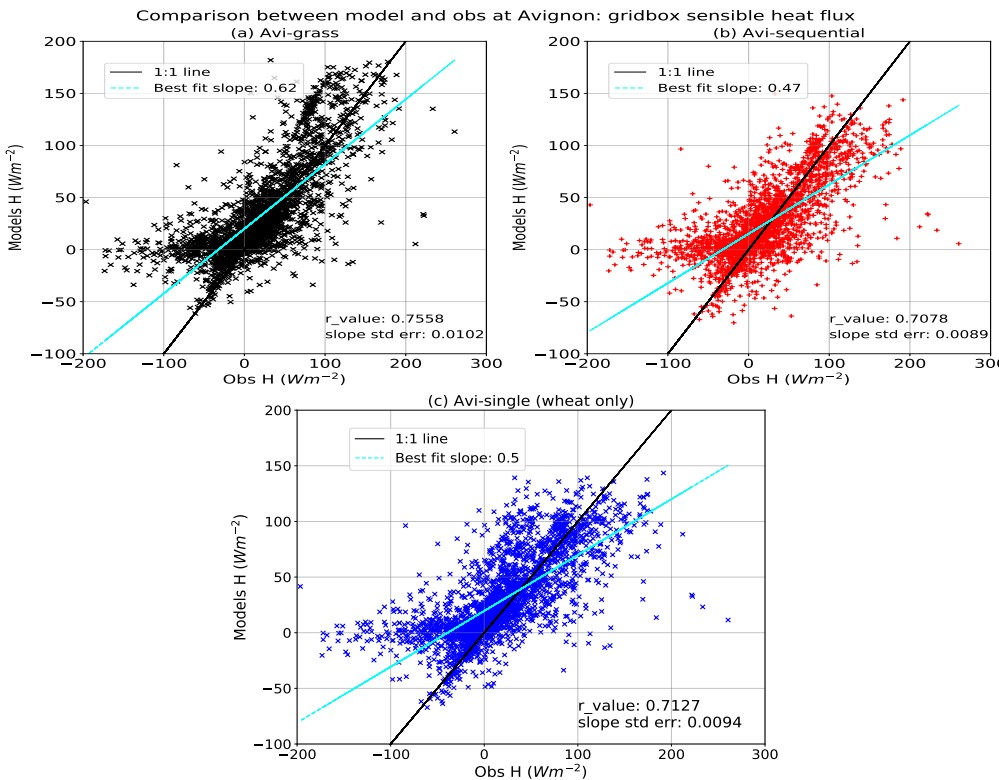

**Figure A.2.** Comparison of observed $H$ at the Avignon site against the modelled $H$ between 2005 and 2013 for Avi-grass (a) and Avi-sequential (b) and Avi-single (c).

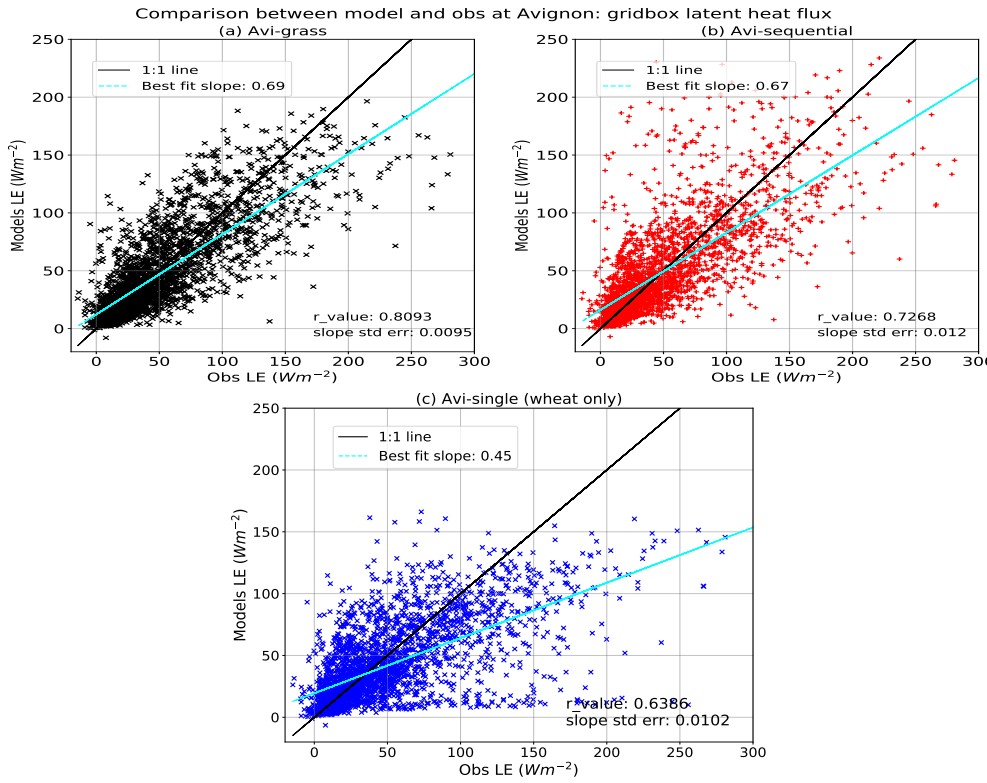

**Figure A.3.** Comparison of observed $LE$ at the Avignon site against the modelled $LE$ between 2005 and 2013 for Avi-grass (a) and Avi-sequential (b) and Avi-single (c).

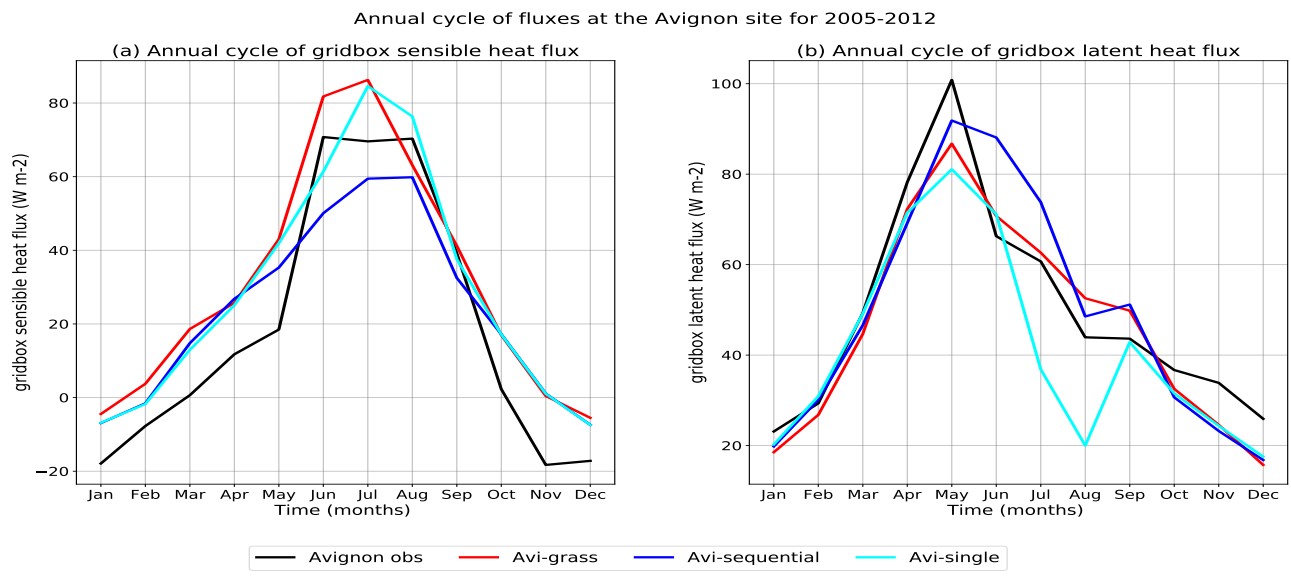

**Figure A.4.** Annual cycle of $H$ (a) and $LE$ (b) compared with observations (black line) at the Avignon site between 2005 and 2013. Annual cycles for the simulations are also shown: Avi-grass (red line), Avi-sequential (blue line) and Avi-single (cyan line).

# 10 Appendix B: India single gridbox comparison

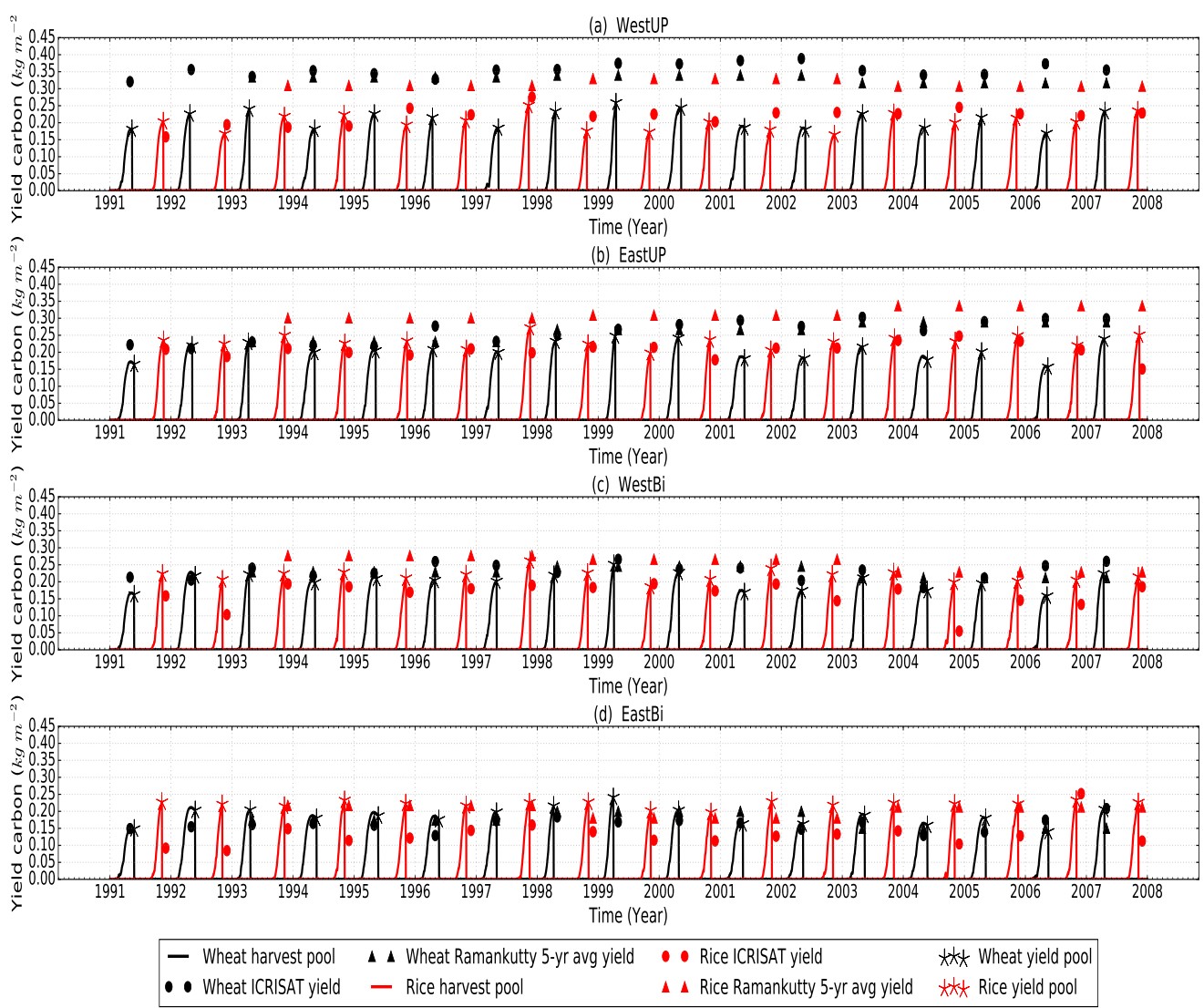

**Figure B.1.** Timeseries of crop harvest pool (solid lines) with the JULES yield from the sequential crop run at the time it is output by the model (asterisks) for rice (red) and wheat (black) at each of the India locations shown in Fig. 3. Also shown are two sets of observations; annual yields from ICRISAT (2015) shown by the filled circles and 5 year averages from Ray et al. (2012a) shown by the filled triangles (following the same colours with rice shown in red and wheat in black)

Sequential cropping: Total biomass for both rice and wheat

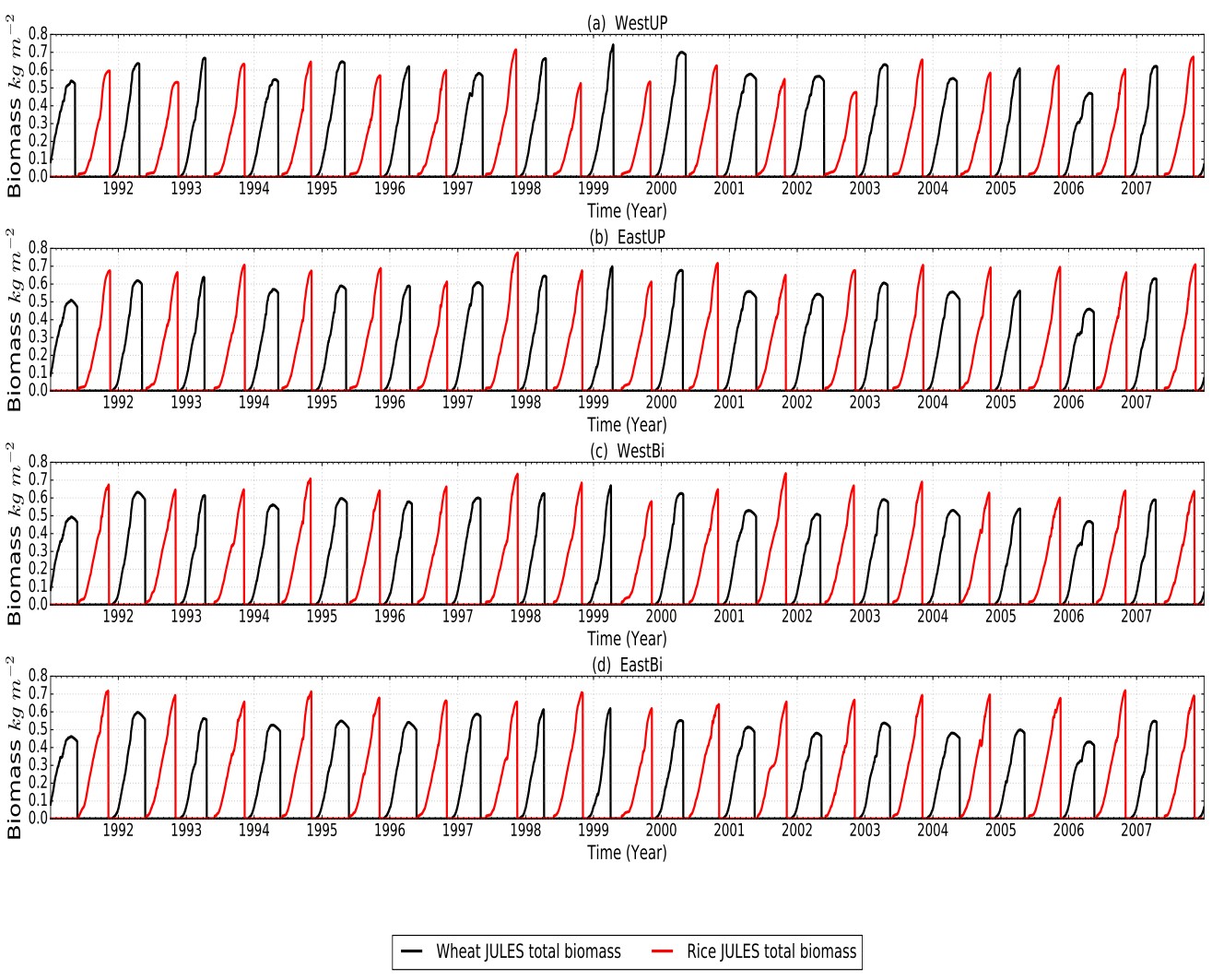

**Figure B.2.** Timeseries of total biomass for rice (red) and wheat (black) at each of the India locations shown in Fig. 3.

India gridboxes: Canopy height (Canht)

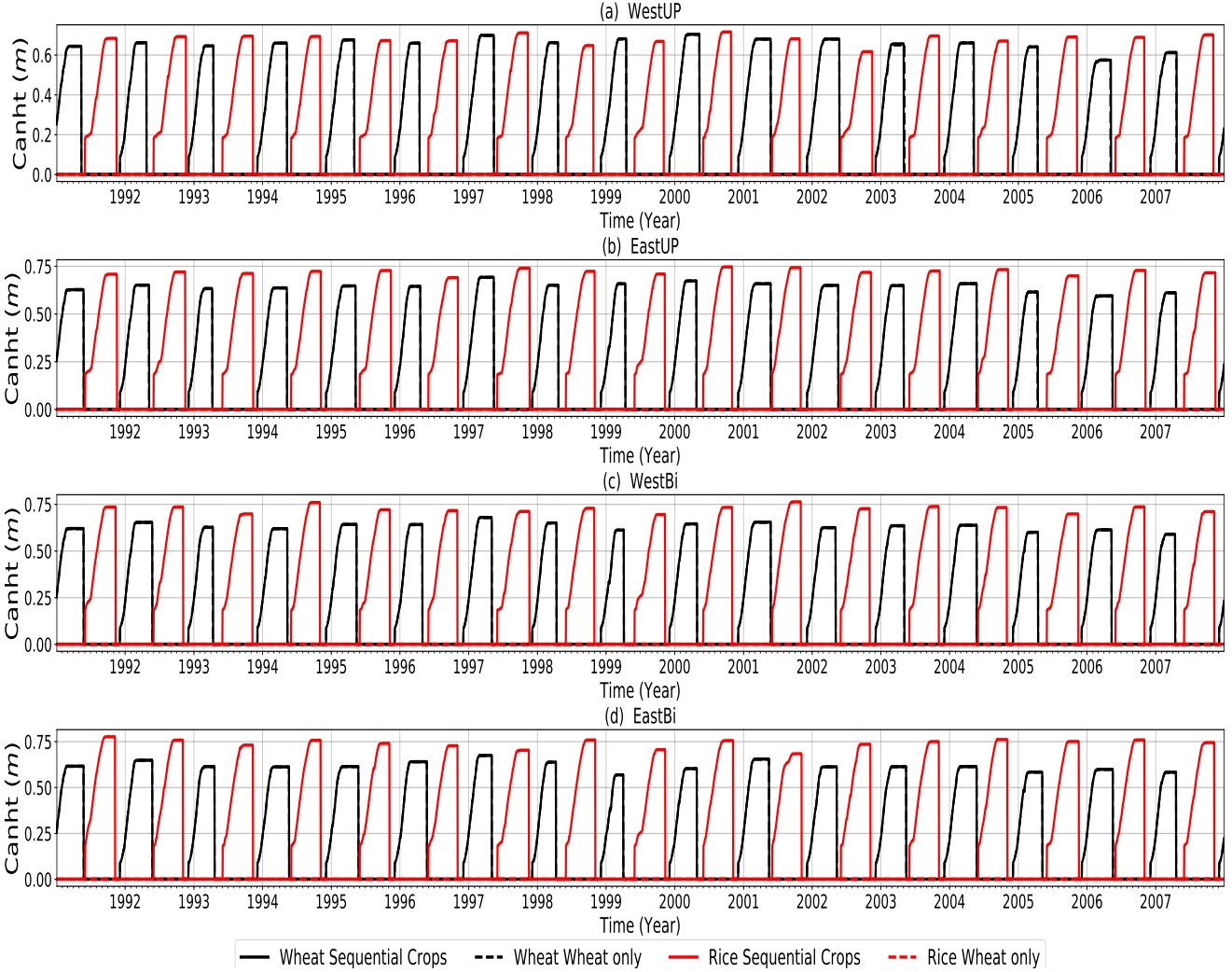

**Figure B.3.** Timeseries of canopy height for rice (red) and wheat (black) at each of the India locations shown in Fig. 3. This plot shows India-sequential (solid) and India-single (dashed) although the two simulations are indistinguishable during the wheat season because the wheat in these simulations are so similar.

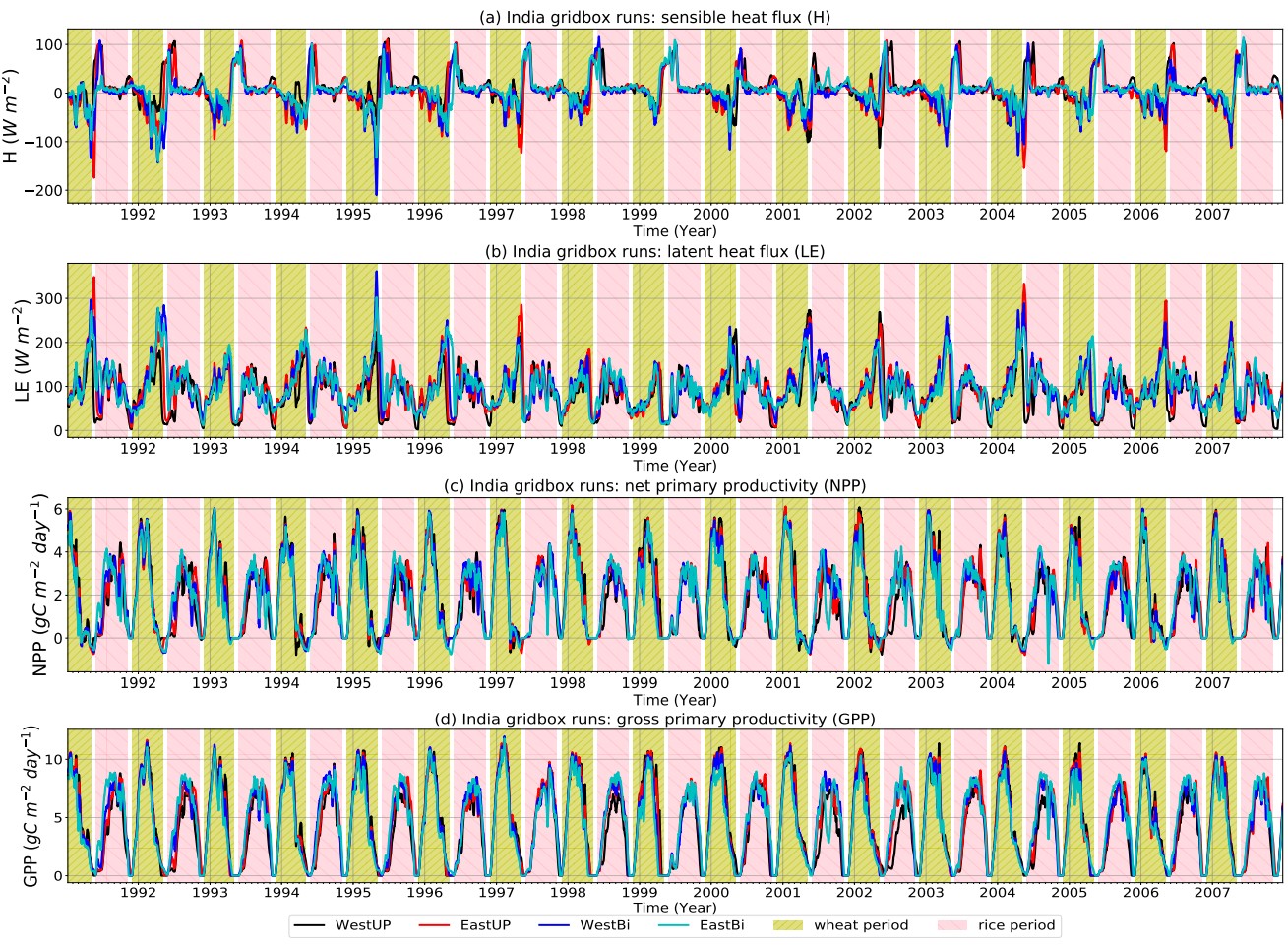

**Figure B.4.** Timeseries of $H$ (a), $LE$ (b), gridbox NPP (c) and gridbox GPP (d) at each of the India locations shown in Fig. 3. Each location is represented by a solid line of a different colour: WestUP - black, EastUP - red, WestBi - blue and EastBi - cyan

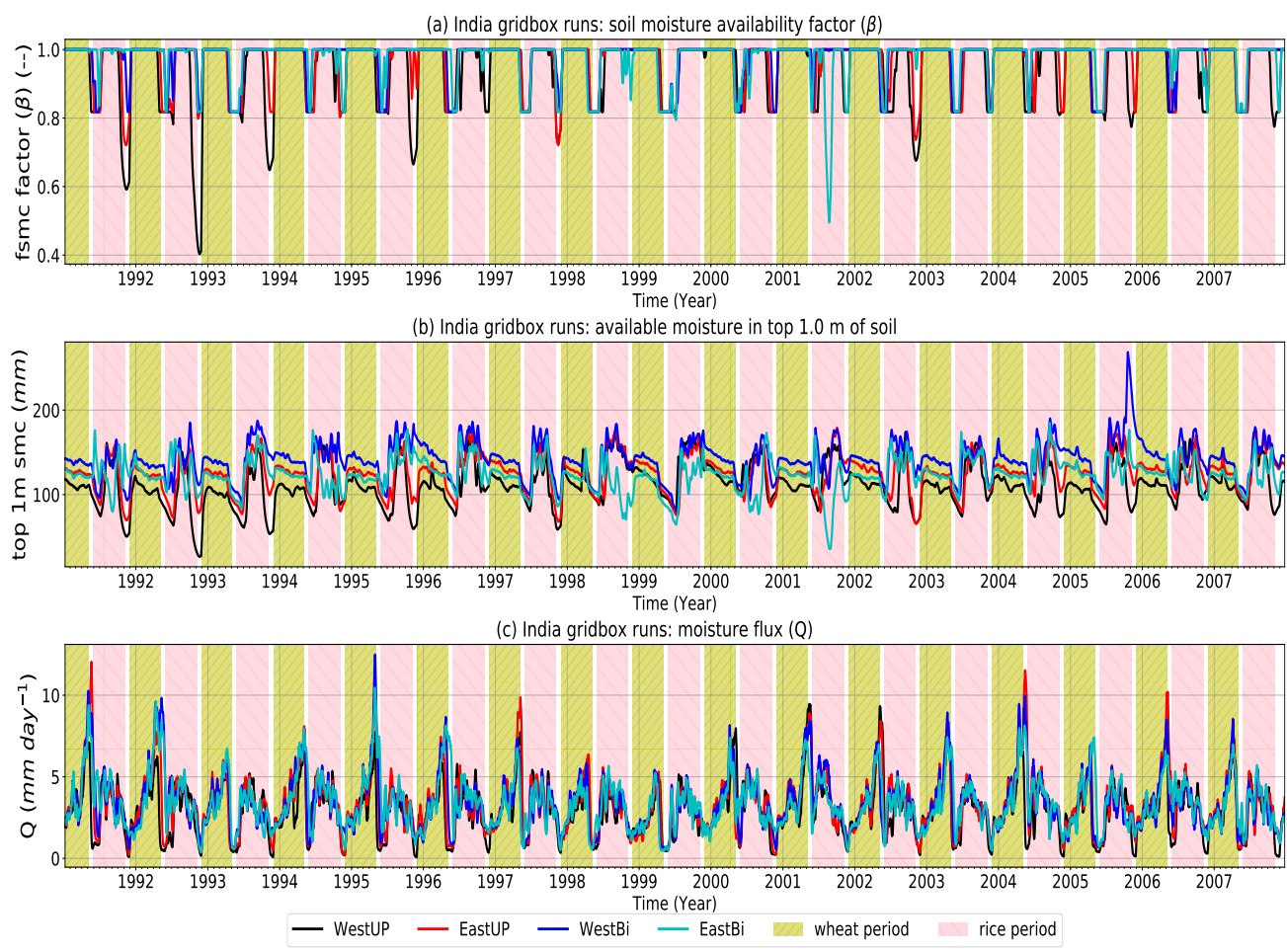

**Figure B.5.** Timeseries of $\beta$ (a), available soil moisture in the top 1.0 m of soil (b) and moisture flux (c). Each location is represented by a solid line of a different colour: WestUP - black, EastUP - red, WestBi - blue and EastBi - cyan

.

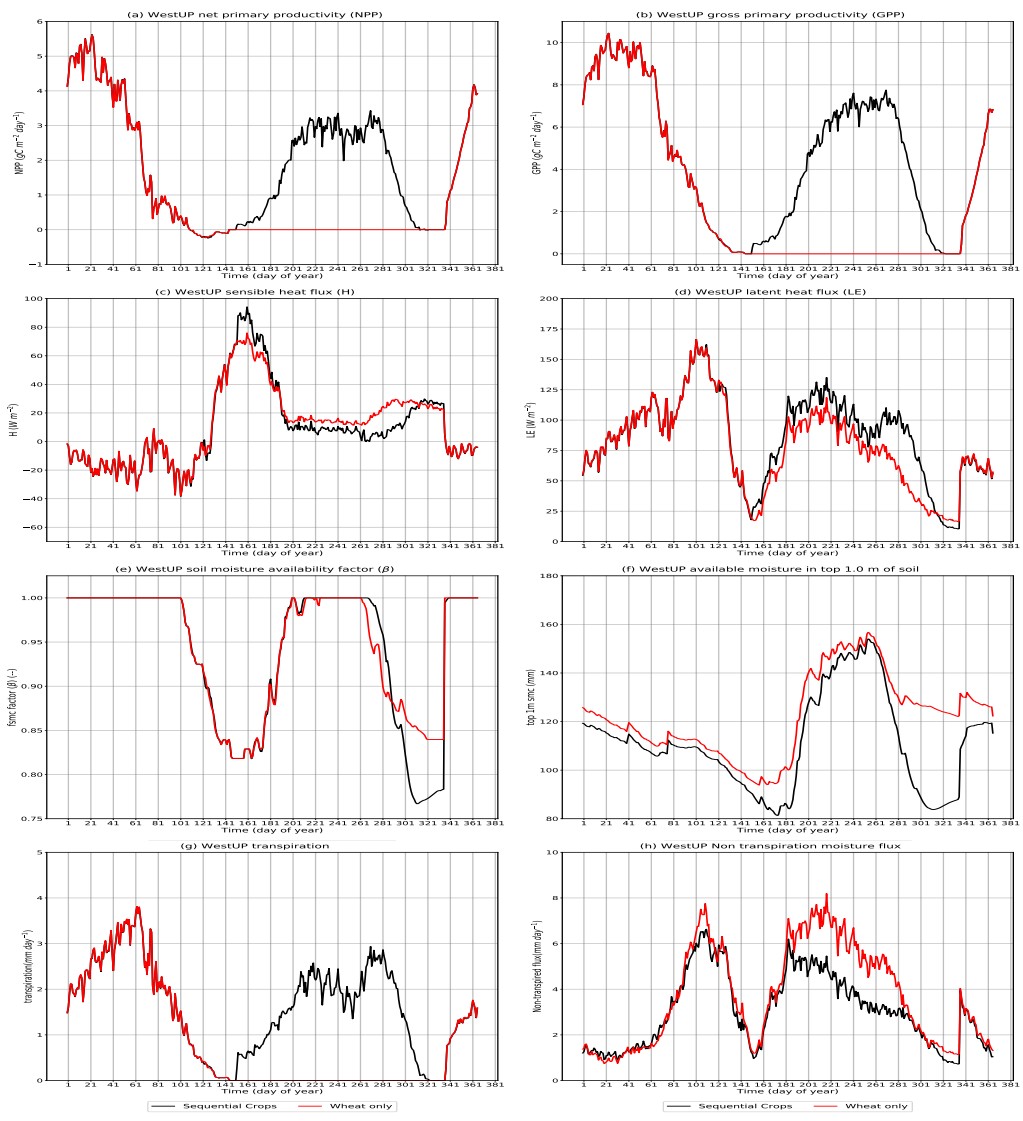

**Figure B.6.** Annual climatology of fluxes (in day of year) for WestUP for India-sequential (black) and India-single (red): Carbon fluxes: NPP (a) and GPP (b). Heat fluxes: sensible heat $H$ (c) and latent heat $LE$ (d). Soil moisture variables: $\beta$ (e) and soil moisture availability in the top 1 m of soil (f). Moisture fluxes: evapotranspiration (g) and non-evapotranspiration moisture fluxes (h).

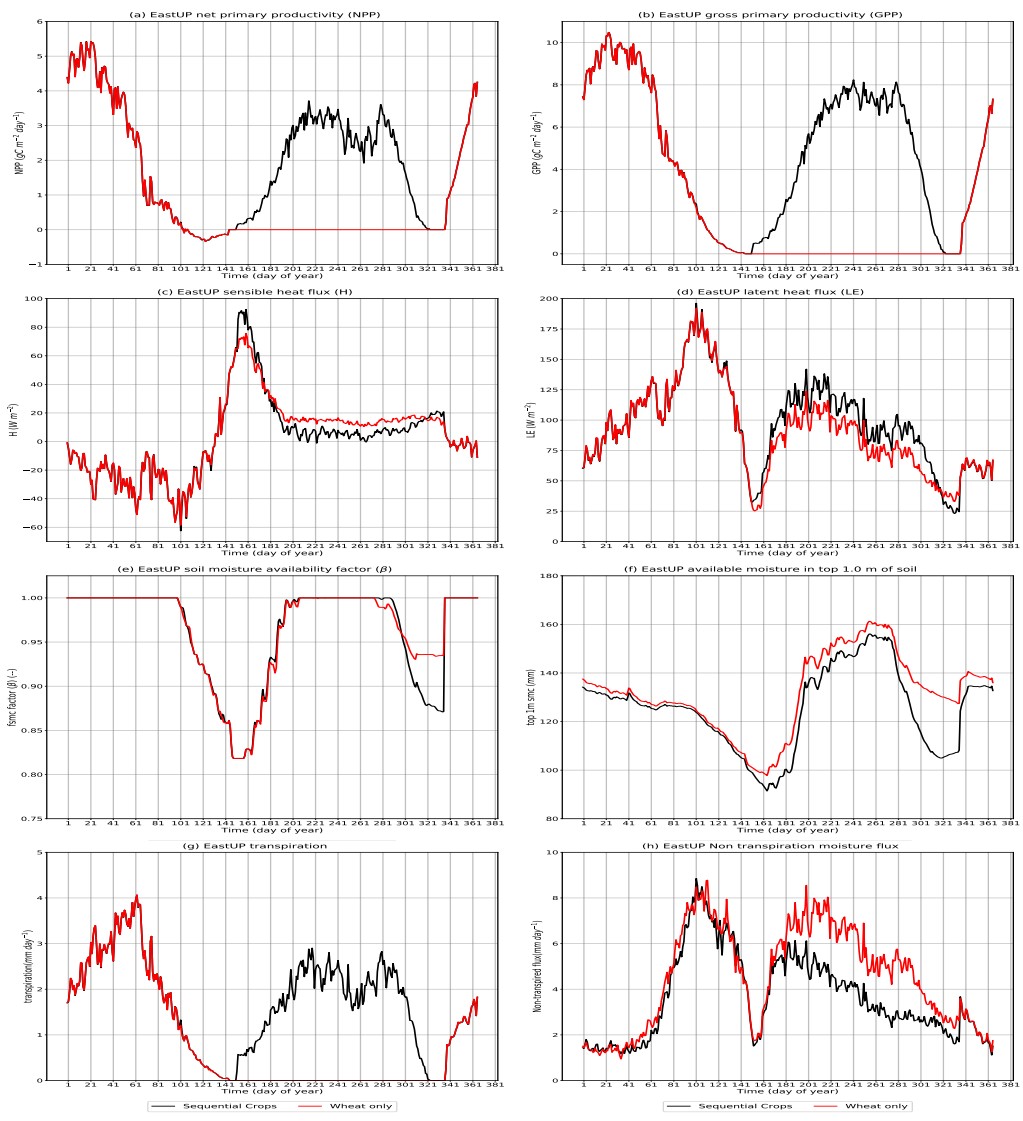

**Figure B.7.** Annual climatology of fluxes (in day of year) for EastUP for India-sequential (black) and India-single (red): Carbon fluxes: NPP (a) and GPP (b). Heat fluxes: sensible heat $H$ (c) and latent heat $LE$ (d). Soil moisture variables: $\beta$ (e) and soil moisture availability in the top 1 m of soil (f). Moisture fluxes: evapotranspiration (g) and non-evapotranspiration moisture fluxes (h).

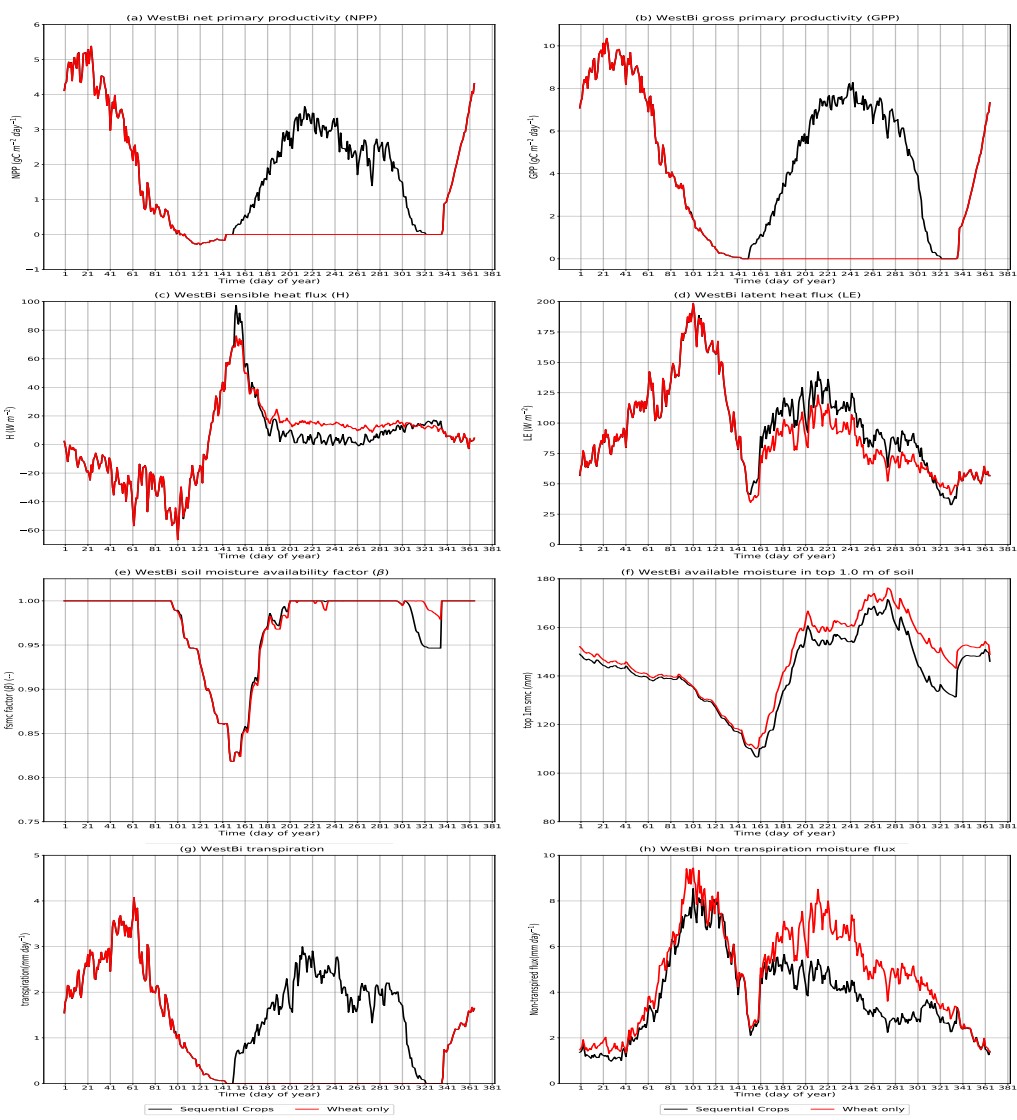

**Figure B.8.** Annual climatology of fluxes (in day of year) for WestBi for India-sequential (black) and India-single (red): Carbon fluxes: NPP (a) and GPP (b). Heat fluxes: sensible heat $H$ (c) and latent heat $LE$ (d). Soil moisture variables: $\beta$ (e) and soil moisture availability in the top 1 m of soil (f). Moisture fluxes: evapotranspiration (g) and non-evapotranspiration moisture fluxes (h).

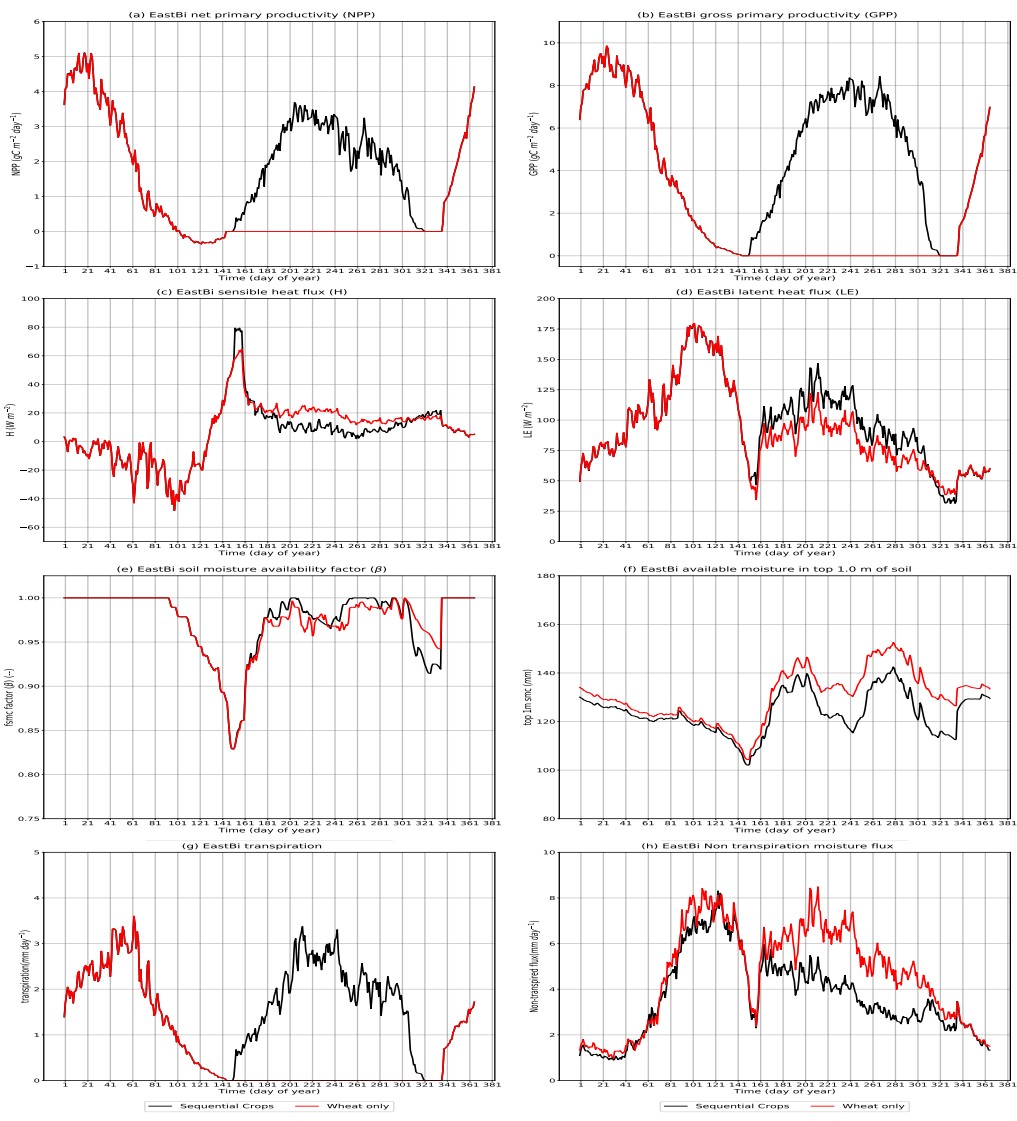

**Figure B.9.** Annual climatology of fluxes (in day of year) for EastBi for India-sequential (black) and India-single (red): Carbon fluxes: NPP (a) and GPP (b). Heat fluxes: sensible heat $H$ (c) and latent heat $LE$ (d). Soil moisture variables: $\beta$ (e) and soil moisture availability in the top 1 m of soil (f). Moisture fluxes: evapotranspiration (g) and non-evapotranspiration moisture fluxes (h).

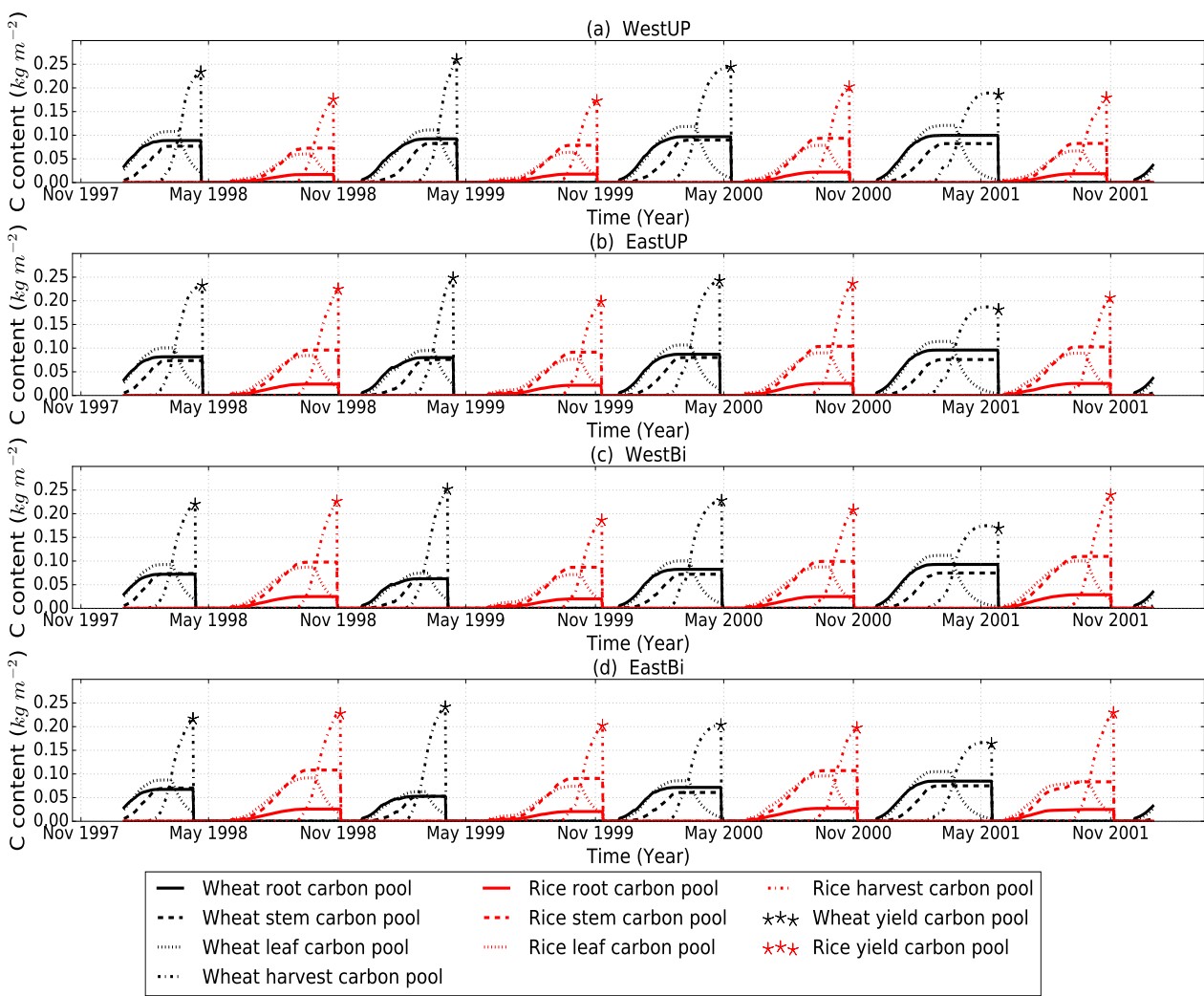

**Figure B.10.** Timeseries of each crop carbon pool: leaf (solid lines), root (dashed), stem (dotted) and harvest (dash-dot) with the JULES yield at the time it is output by the model (asterisks) for rice (red) and wheat (black) at each of the India locations shown in Fig. 3 for a subset of years of the simulation between 1998 and 2001.

# 11 Appendix C: India regional simulation

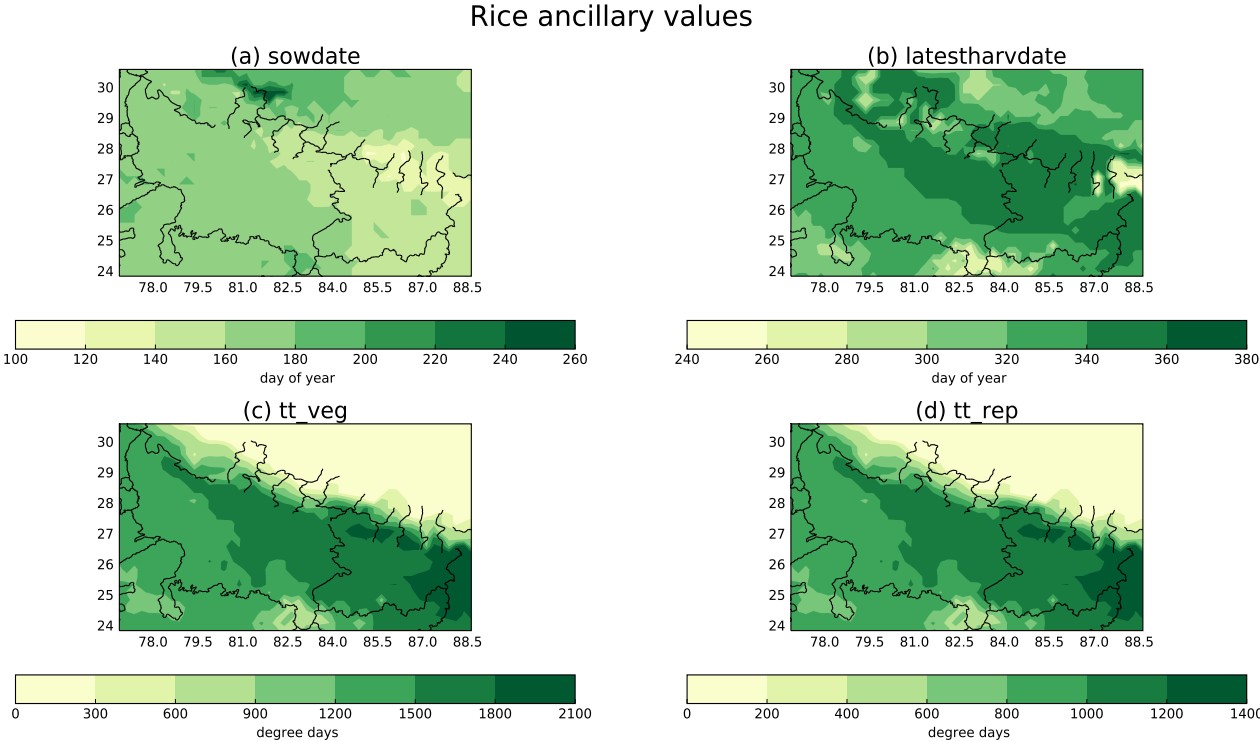

**Figure C.1.** The values used in the regional JULES ancillary for rice. Sowing date (a) and latest possible harvest date (b), both in units of day of year. Thermal time for the vegetative stage (c) and thermal time for the reproductive stage (d) both in units of degree days

.

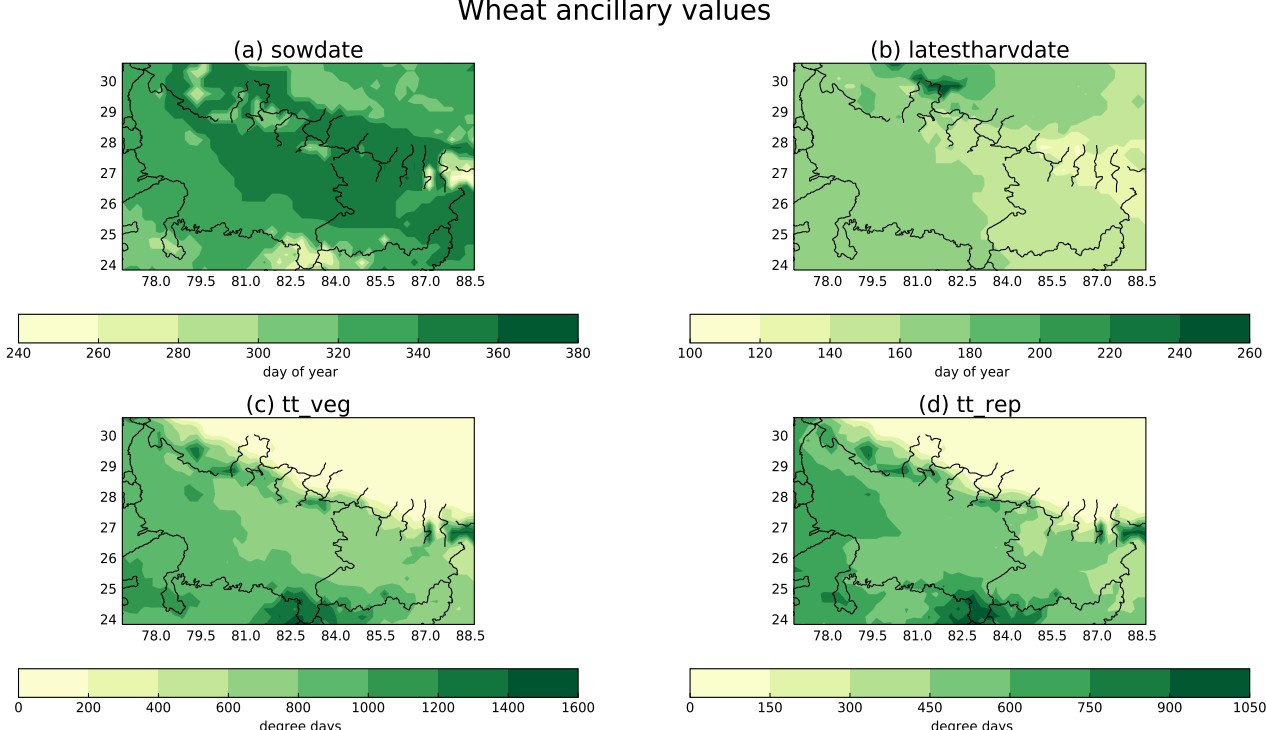

**Figure C.2.** The values used in the regional JULES ancillary for wheat. Sowing date (a) and latest possible harvest date (b), both in units of day of year. Thermal time for the vegetative stage (c) and thermal time for the reproductive stage (d) both in units of degree days

.

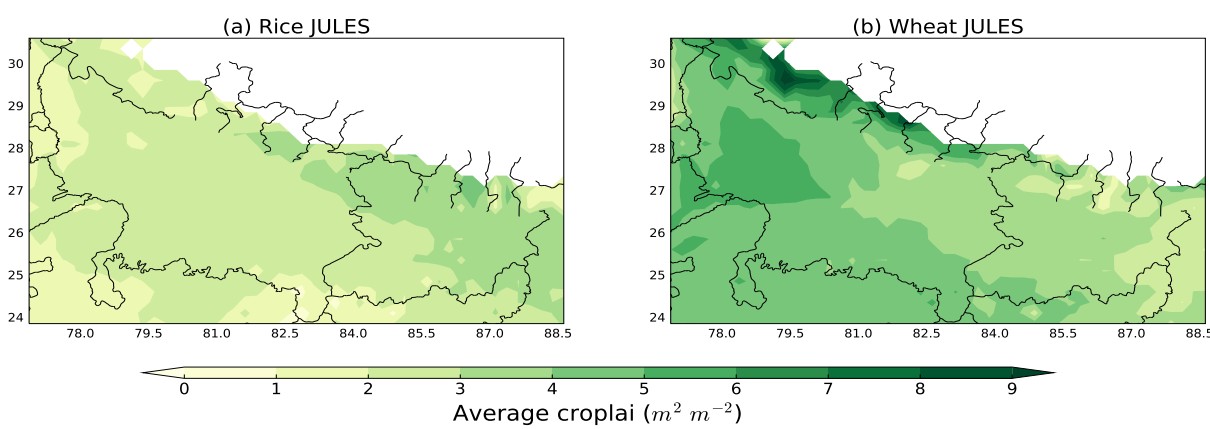

**Figure C.3.** Average of the maximum annual crop LAI for rice and wheat for the period 1991-2007 across Uttar Pradesh and Bihar.

## Crop canopy height averaged for period 1991-2007

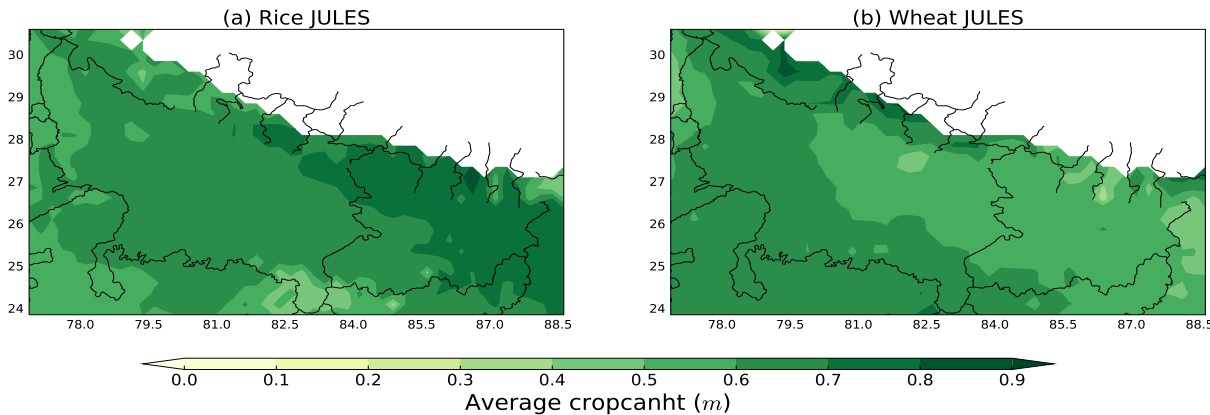

**Figure C.4.** Average of the maximum annual crop canopy height for rice and wheat for the period 1991-2007 across Uttar Pradesh and Bihar.

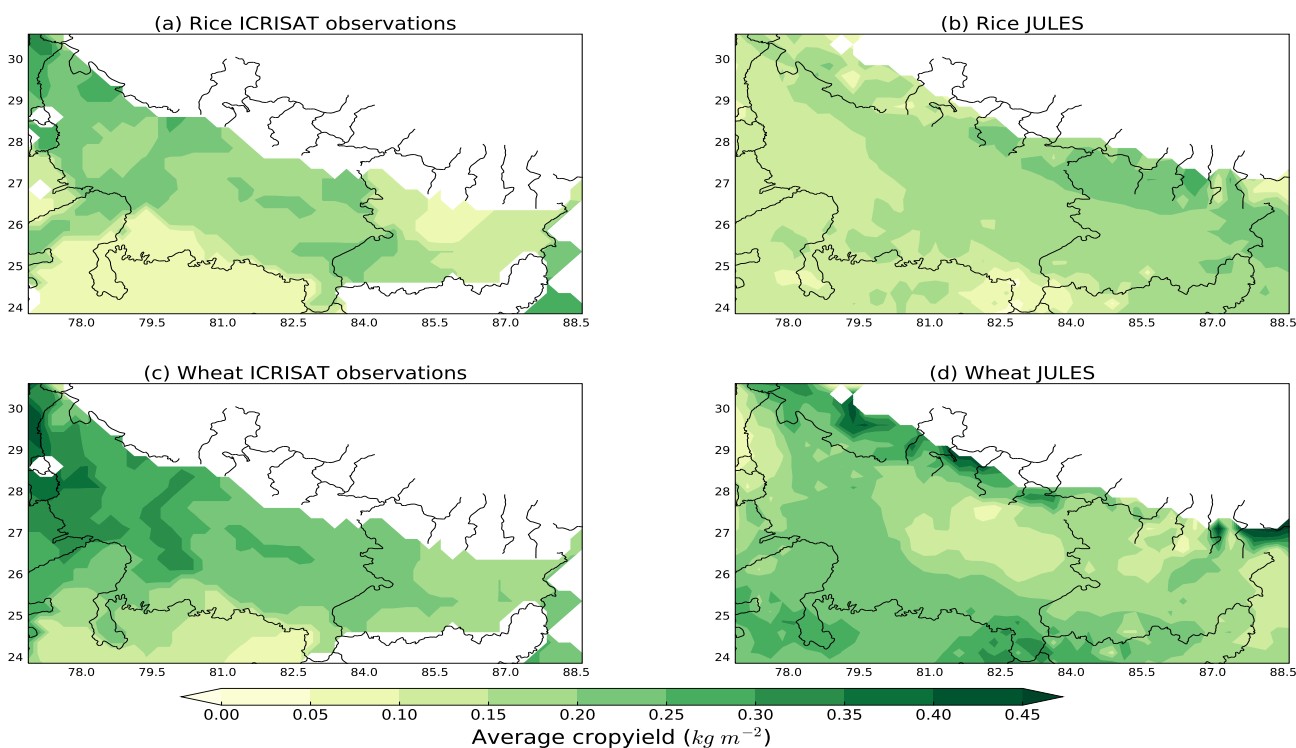

**Figure C.5.** A comparison of observed ICRISAT (2015) rice yields (a) with JULES rice yields (b) and observed ICRISAT (2015) wheat yields (c) with JULES wheat yields (d) for the period 1991-2007 across Uttar Pradesh and Bihar.

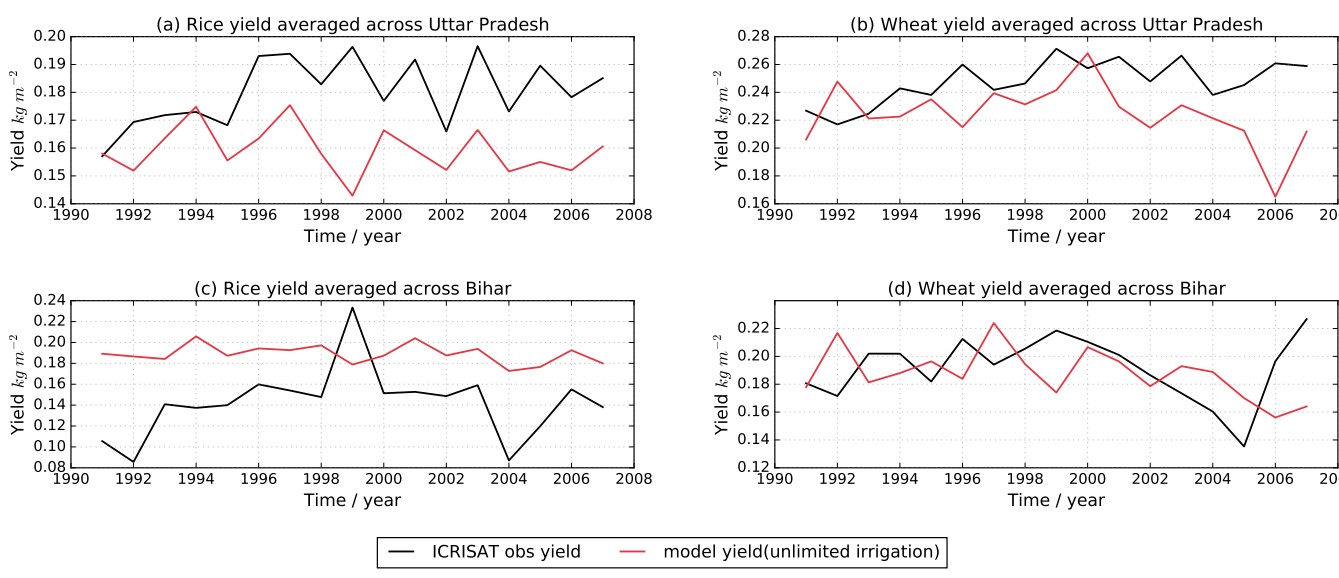

**Figure C.6.** Annual timeseries of the JULES yield for Uttar Pradesh and Bihar for rice and wheat compared with ICRISAT (2015) observations. Irrigation is applied to wheat with no limitation on water availability (see Sect. 4).

*Author contributions.* Andrew J Challinor, Pete Falloon and Andy Wiltshire provided general scientific guidance throughout the paper and helped prioritise the initial requirement for including sequential cropping in JULES, toward the aim of modelling integrated impacts for South Asia. Chetan Deva provided gridded observations of cropyield courtesy of ICRISAT. Sébastien Garrigues and Sophie Moulin provided Avignon data. Karina Williams provided code for generating the crop ancillary files and an initial suite for Avignon without crops.

5  *Competing interests.* The authors declare that they have no conflict of interest.

*Acknowledgements.* The research leading to these results has received funding from the European Union Seventh Framework Programme FP7/2007–2013 under grant agreement no. 603864. Camilla Mathison, Pete Falloon, Andy Wiltshire and Karina Williams were supported by the Met Office Hadley Centre Climate Programme funded by BEIS and Defra. Thanks also to Kate Halladay for helping me understand the soil moisture and irrigation implications. The authors thank the INRAE EMMAH laboratory at Avignon (France) in charge of the "flux
10  and remote sensing site" for the supply of the data. The authors thank the reviewers and the editor for their comments and suggestions.

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

| Flag | JULES notation | Avignon settings | India settings | Effect of switch |
|---|---|---|---|---|
| Canopy radiation scheme | can_rad_mod | 6 | 6 | Selects the canopy radiation scheme. |
| Irrigation demand | l_irrig_dmd | F | T | Switches on irrigation demand. |
| Irrigation scheme | irr_crop | - | 2 | Irrigation occurs when the DVI of the crop is greater than 0. |
| Physiology | l_trait_phys | F | F | Switches on trait based physiology when true. |
| Sowing | l_prescsow | T | T | Selects prescribed sowing. |
| Plant maintenance respiration | l_scale_resp_pm | F | F | Switch to scale respiration by water stress factor. If false this is leaf respiration only but if true includes all plant maintenance respiration. |
| Crop rotation | l_croprotate | T | T | A new switch to use the sequential cropping capability. |
| Irrigation on tiles | frac_irrig_all_tiles | - | F | Switch to allow irrigation on all or specific tiles |
| Irrigation on specific tiles | set_irrfrac_on_irrtiles | - | T | A new switch to set irrigation to only occur on a specific tile. |
| Specify irrigated tile(s) | irrigtiles | - | 6 | Setting to set the value(s) of the specific tile(s) to be irrigated. |
| Number of tiles irrigated | nirrtile | - | 1 | Setting to set how many tile(s) to be irrigated. |
| Set a constant irrigation fraction | const_irrfrac_irrtiles | - | 1.0 | A new setting to set the value(s) of the irrigation fraction for specific tile(s) to be irrigated in the absence of a file of irrigation fractions. |

**Table 1.** JULES flags used that are new or different from those in Osborne et al. (2015)

| Parameter | JULES notation | Description (units) | Winter wheat | Sorghum | Spring wheat | Rice |
|---|---|---|---|---|---|---|
| $T_{low}$ | t_low_io | Lower temperature for photosynthesis ($^\circ$ C). | 5 | 18 | 5 | 15 |
| $T_{upp}$ | t_upp_io | Upper temperature for photosynthesis ($^\circ$ C). | 30 | 53 | 30 | 40 |
| $n_{eff}$ | neff_io | Scale factor relating $V_{cmax}$ with leaf nitrogen concentration. | 0.8e-3 | 0.75e-3 | 0.8e-3 | 0.95e-3 |
| $n_l(0)$ | nl0_io | Top leaf nitrogen concentration (kg N/kg C). | 0.073 | 0.07 | 0.073 | 0.073 |
| fsmc method | fsmc_mod_io | When equal to 0 we assume an exponential root distribution with depth. | | | 0 | 0 |
| | | When equal to 1, the soil moisture availability factor, fsmc, is calculated using average properties for the root zone. | 1 | 1 | | |
| $d_r$ | rootd_ft_io | If fsmc_mod_io = 0 $d_r$ is the e-folding depth (m). | | | 0.5 | 0.5 |
| | | If fsmc_mod_io = 1 $d_r$ is the total depth of the root zone (m). | 1.5 | 1.5 | | |
| p0 | fsmc_p0_io | Parameter governing the threshold at which the plant starts to experience water stress due to lack of water in the soil. | 0.5 | 0.5 | 0.5 | 0.5 |
| $\mu_{rl}$ | nr_nl_io | Ratio of root nitrogen concentration to leaf nitrogen concentration. | 0.39 | 0.39 | 0.39 | 0.39 |
| $\mu_{sl}$ | ns_nl_io | Ratio of stem nitrogen concentration to leaf nitrogen concentration. | 0.43 | 0.43 | 0.43 | 0.43 |
| $Q_{10,leaf}$ | q10_leaf_io | $Q_{10}$ factor in the $V_{cmax}$ calculation. | 1.0 | 1.0 | 1.0 | 1.0 |

**Table 2.** JULES plant functional type (PFT) parameters and values modified for use in this study. We include only the values that have been changed or are new in JULES since Osborne et al. (2015)

| Parameter | JULES notation | Description (units) | Winter wheat | Sorghum | Spring wheat | Rice |
|---|---|---|---|---|---|---|
| $T_b$ | t_bse_io | Base temperature ($^\circ$ K). | 273.15 | 284.15 | 273.15 | 278.15 |
| $T_m$ | t_max_io | Max temperature ($^\circ$ K). | 303.15 | 317.15 | 308.15 | 315.15 |
| $T_o$ | t_opt_io | Optimum temperature ($^\circ$ K). | 293.15 | 305.15 | 293.15 | 303.15 |
| $TT_{emr}$ | tt_emr_io | Thermal time between sowing and emergence ($^\circ$ Cd). | 35 | 80 | 35 | 60 |
| $TT_{veg}$ | tt_veg_io | Thermal time between emergence and flowering ($^\circ$ Cd). | Table 4 | Table 4 | Table 5 | Table 5 |
| $TT_{rep}$ | tt_rep_io | Thermal time between flowering and maturity ($^\circ$ Cd). | Table 4 | Table 4 | Table 5 | Table 5 |
| $T_{mort}$ | t_mort_io | Soil temperature ($2^{nd}$ level) at which to kill crop if DVI>1 ($^\circ$ K). | 273.15 | 281.15 | 273.15 | 281.15 |
| $f_{yield}$ | yield_frac_io | Fraction of the harvest carbon pool converted to yield carbon. | 1.0 | 1.0 | 1.0 | 1.0 |
| $DVI_{init}$ | initial_c_dvi_io | DVI at which the crop carbon is set to $C_{init}$. | 0.0 | 0.0 | 0.0 | 0.0 |
| $DVI_{sen}$ | sen_dvi_io | DVI at which leaf senescence begins. | 1.5 | 1.5 | 1.5 | 1.5 |
| $C_{init}$ | initial_carbon_io | Carbon in crop at emergence in kgC/m$^2$. | 0.01 | 0.01 | 0.01 | 0.01 |

**Table 3.** JULES crop parameters used in this study. The sorghum cardinal temperatures are from Nicklin (2012) with the other parameters those used for Maize in Osborne et al. (2015). We include only the values that have been changed or added since Osborne et al. (2015). Table 3 of Osborne et al. (2015) provides the original PFT parameters and Table 4 of Osborne et al. (2015) provides the original crop parameters).

| Year | Crop | Sowing date | Harvest date | Emergence-flowering | Flowering-maturity | Sowing DOY |
|---|---|---|---|---|---|---|
| 2005 | Winter wheat | 27 Oct 2005 | | 1301.3 | 867.5 | 300 |
| 2006 | | | 27 Jun 2006 | | | |
| 2007 | Sorghum | 10 May 2007 | 16 Oct 2007 | 647.6 | 791.5 | 130 |
| 2007 | Winter wheat | 13 Nov 2007 | | 1401.0 | 934.0 | 317 |
| 2008 | | | 1 Jul 2008 | | | |
| 2009 | Sorghum | 25 Jun 2009 | 22 Sep 2009 | 462.5 | 565.3 | 176 |
| 2009 | Winter wheat | 19 Nov 2009 | | 1308.6 | 872.4 | 323 |
| 2010 | | | 13 Jul 2010 | | | |
| 2011 | Sorghum | 22 Apr 2011 | 22 Sep 2011 | 679.5 | 830.5 | 112 |
| 2011 | Winter wheat | 19 Oct 2011 | | 1559.6 | 1039.7 | 292 |
| 2012 | | | 25 Jun 2012 | | | |

**Table 4.** Thermal times in degree days used in this study for the Avignon site, these are based on the observed sowing and harvest dates from Garrigues et al. (2015).

| Location | Crop | Sowing DOY | Emergence-flowering | Flowering-maturity |
|----------|------|-----------|---------------------|--------------------|
| WestUP | Spring wheat | 335 | 1007.6 | 671.1 |
| | Rice | 150 | 1759.4 | 1181.3 |
| EastUP | Spring wheat | 335 | 993.55 | 662.5 |
| | Rice | 150 | 1865.5 | 1243.5 |
| WestBi | Spring wheat | 335 | 991.54 | 661.6 |
| | Rice | 150 | 1907.55 | 1271.7 |
| EastBi | Spring wheat | 335 | 1019.21 | 679.1 |
| | Rice | 150 | 1976.96 | 1300.64 |

**Table 5.** The sowing day of year (Sowing DOY) and thermal times in degree days used in this study for the locations in Uttar Pradesh and Bihar, India (see Fig. 3 for a map of the locations), the values given here are based on the observed sowing and harvest dates from Bodh et al. (2015)

.

| Variable | Simulation type | RMSE | Bias | r value |
|----------|-----------------|------|------|---------|
| GPP ($gCm^{-2}day^{-1}$) | grass | 2.0 | -1.0 | 0.95 |
| | sequential | 3.0 | 0.0 | 0.82 |
| | single | 5.0 | -2.0 | 0.52 |
| $H$ ($Wm^{-2}$) | grass | 37.0 | 13.0 | 0.76 |
| | sequential | 38.0 | 6.0 | 0.71 |
| | single | 39.0 | 11.0 | 0.71 |
| $LE$ ($Wm^{-2}$) | grass | 28.0 | -3.0 | 0.81 |
| | sequential | 33.0 | 0.0 | 0.73 |
| | single | 37.0 | -8.0 | 0.64 |

**Table 6.** Table of statistics comparing the Avignon simulations with observations for each type of run: Avi-single (single), Avi-sequential (sequential) and Avi-grass (without the crop model).