# Peer review of "Developing a sequential cropping capability in the JULESvn5.2 land–surface model"

_Geoscientific Model Development, 2019_

## Referee Comment (RC1) · Anonymous Referee #1 · 30 May 2019

The manuscript by Mathison et al., describes a model development in the JULESvn5.2 land-surface model to include sequential cropping. While the JULES model already includes agriculture, multiple growing seasons of crops, which are common in many tropical regions, is not included. In fact, multiple cropping is generally not included in land surface models. The approach uses a site in Avingon, with intense observations, to demonstrate the model capability and then four sites in India (across a gradient of temperature and precipitation) as test cases. The specific crops grown in the sites aren't represented in JULES, so proxies are used. The model is evaluated against several variables, including carbon pools, LAI, GPP, NEE, and heat fluxes, but the most important feature is capturing the peak of the growing season of the different crop types. Although the results vary, the model does reproduce consistent crop seasons.

Overall, I find the manuscript to be interesting and a good first step in the next generation of crop modeling. However, there are several areas where additional clarification is needed. The manuscript would benefit from editing (punctuation and grammar in particular).

My main concern with this manuscript is the focus in the results on comparing with observations. While I think that is valuable, most of the results and discussion center on parameterization of the crops in JULES causing under or over-estimated yields (or other variables), and not necessarily the results from the addition of sequential cropping. This includes soil moisture, VPD, carbon partitioning, temperature. Since this discussion never loops back to the sequential cropping relationship, it should be removed because it adds length to the paper without providing valuable discussion.

There are also a significant amount of discussion in the results section. I provide some examples in my General Comments. These discussions in the results section belong elsewhere (if at all) in the manuscript (see previous comment).

Finally, I think what should be addressed is how the authors plan to scale this work up to larger simulations. The method requires external data on sequential cropping that is generally not available, especially at large scales. I agree that incorporating sequential cropping into agricultural models is important, but the method requires data on planting, harvest, and the location and types of crops that are grown sequentially. What is the next step to provide this?

General Comments:

P 4, L 22-24: I believe there is a lot of discussion in section 6. Either the discussion should go in section 7 or relabel the sections to reflect that. Also, section 5 is not listed here and should be for consistency.

P 5, L 6: Can this be clarified to indicate air or leaf temperature?

P 5, L 23: Please include the reference for the dataset(s) that were used for the sowing

and harvest dates when observations weren't available?

P 6, L 8-9: Is forcing a harvest before maturity is reached realistic? Did any of the model simulations need to use this constraint any year?

P 6-7, Section 3: The DVI discussion might be better in section 2 where the model description is provided.

P 7, L 6-7: What do you mean when you state the effective temperature differs "between models"?

P 7-8, Section 4: The authors should expand on how these specific variables are important for the crop model and how they influence the variables that are being compared with observations (e.g., GPP, latent heat flux, sensible heat flux).

P 8, Section 4.1: The authors mention that the Avignon site is in France in the introduction. Location information (including approximate latitude and longitude) would be good in this section.

P 10, Section 5: It would be useful to have some climate information for all the sites. While some information is provided for the India sites in Figure 2, Section 6.2 and Figure 6, similar information should be included for the Avignon site.

P 11, Section 6: There is a lot of discussion in this section, which should just be results.

P 11, L 5: I do not see AviJUL-grass results in Figure 3.

P 11, L 10-27: I do not think it is very surprising that the simulations do not capture the observed biomass and LAI. The crops in JULES are corn and spring wheat whereas the crops at the Avignon site are sorghum and winter wheat. The fact that the model did well for the 2009 sorghum season, which was an anomaly for that crop at the observed site, indicates the crop parameters need to be tuned for a true comparison. I'm not suggesting the authors need tune the model for this manuscript, however, the different crops could play a large role in the model performance, which should have a place in

the discussion.

P 11 L 28- P12: The discussion for p0 comes completely by surprise. The authors don't explain how p0 affects GPP and what the different values represent. This part feels like a discussion and doesn't belong in the results section, considering these additional water sensitivity simulations weren't discussed previously. The additional simulations with modified p0 do not add value to the results and I suggest they be removed (along with Figure A5). Then suddenly, the paragraph changes gears, jumping to GPP. The GPP results includes a grass simulation comparison, which I find unnecessary. The focus should be comparing the crops simulations and observations to be consistent with the rest of the section 6.1. The paragraph needs some editing and is written sloppily. For example, in L 15, "...early the decline is very close to the observations" does not make sense. Finally, delete the last paragraph on page 12.

P 14 L4-5 – P 15: This paragraph is confusing. It is not clear how soil moisture availability factor (beta) is calculated for the model or observations, nor is it clear what the impact has on simulations, other than an early decline in GPP. What is not discussed is that part of the problem could be related to how irrigation is handled in the model. It was stated in the beginning of the manuscript that the irrigation for sorghum was included in the rainfall data, but it was not specified how. If the timing of irrigation is off, that could explain some of the water stress for sorghum that is seen in the model but not in the observations (figure 5). Regardless, I think this is taking away from the main point of the results and I do not feel it is necessary. If it is kept, it should be moved to section 7.

P 15, L6: Do the "four levels" refer to four soil layers in JULES?

P 16-17: "the aim of this simulation is to demonstrate the method rather than provide a perfect representation of either of these crops." Yes – I agree with this statement. This should be the focus of section 6.1 – how this method improves the crop simulations. All the discussion about soil moisture and water stress muddle this point. Highlight the

positives of the approach and mention the tuning discussion for future work.

P 19, L 17: the decline in NPP for wheat is hard to see in the figure.

P 20, L 20-33: The VPD discussion here belongs in section 7, it is not results, but discussion. Furthermore, it is not known what the effect of VPD on is on the model yields. The discussion on the plant response to VPD is difficult to follow and clearly does not belong here.

P 21: It is unclear in this discussion how much of this is related to poor wheat parameterization in JULES or how much is related to the sequential cropping component.

Technical Comments:

Define all acronyms (for example, GPP is never defined; DVI is not defined until Appendix B)

P 7, L 4: switch the ";" to a ":"

P 19, L35: the sentence beginning "Therefore suggesting it is not water stress..." is a fragment.

P 20, L 11-12: "in the simulations" is used twice.

P 20, L 12 and L14: Use first and second rather than "firstly" and "secondly"

P 27, L2-3: The sentence beginning "In general the model produces..." is unclear.

Figures need consistent labeling. Put units on the y-axis. Some figures have the y-axis units in the title, others don't have any at all. Also, in Figures 3,4,5,A1,A2,A3,A4,A5 the label "modelled soil moisture and LAI" is confusing. Shouldn't it be AviJUL_sqcrop?

Figure A1: What four simulations are you referring to in the caption?

---

## Referee Comment (RC2) · Anonymous Referee #2 · 18 Jun 2019

**General comments**

The manuscript submitted by Mathison et al. presents the development of a new functionality within the JULES land-surface model enabling at accounting for several crops in one growing season (ie sequential cropping capability). This is certainly a functionality that will contribute at better representing the phenology and the water and energy fluxes and other related variables in areas of the World where multicropping is commonly applied. The motivations behind this development are rather well adressed in the Introduction section. However, the manuscript lacks of key significant objectives and the rest of the manuscript strongly suffers of this absence.

Currently, the manuscript presents the results of this new version at two locations: 1) a super-site in France (Avignon) where long-term measurements of canopy height, lai,

above ground biomass, carbon (GPP) and energy (latent and sensible heat) fluxes are available. Simulated variables are compared to in-situ data for this set of fluxes and biometric variables. In addition, GPP, LE and H fluxes simulated with the version including the sequential cropping functionality are compared to those obtained with a version in which lai and the canopy height are prescribed from in-situ observations; 2) a set of four Indian "points" where the model is ran with sequential cropping of wheat and rice. Modelled annual yields over these four points are compared to two observation-based yield estimates and the model evolution for an extended set of variables (lai, carbon pools, LE, H, NPP, GPP, soil moisture, dvi, canopy height, maintenance respiration, plant respiration) is shown for these four points.

As presented in the manuscript, the objective of the paper appears to be the development of the sequential cropping functionality and the results presented in the manuscript are mainly a set of model outputs for a suite of variables showing that the objective has been achieved (by producing two cycles for LAI, canopy height, gpp, ... within a year). The development of the sequential cropping is a functionality helping at addressing scientific questions. In this respect, although being a Technical and Development Paper, the manuscript really needs more challenging scientific objectives. Currently, due to the lack of appropriate scientific evaluation of the sequential cropping functionality (and of its added value), I would not support the publication of the manuscript in its present form.

A general question to address could be "what's the added value of representing the sequential cropping on a given variable or process?". The variables and/or the scale to focus on should be chosen adequately, as it is sure that looking at LAI at site-level can only lead to the conclusion that the version with the sequential cropping performs better than a version with only one crop per year, for a site where sequential cropping is applied. Addressing that type of question would need the provision of simulations with a model configuration without the sequential cropping (ie with only one crop growing within a year), to be used as a reference. The authors mention page 29 line 15

that they plan using this sequential cropping method for regional simulations ("these regional simulations will be the focus of work that follows this paper"). I would strongly encourage the authors to include these regional simulations in the present study. Performing such regional simulations for present-day conditions, with and without the sequential cropping (ie with only one of the two crops grown within a year) would enable to investigate the impact of the sequential cropping on the vegetation intensity and the soil moisture at regional scale and to directly compare these modelled variables with observation-based data provided by remote-sensed products (LAI (or fPAR) and soil moisture from satellite sensors such as MODIS or GRACE for instance). This could be performed on the studied region of North India.

Still aiming at assessing the added value of representing the sequential cropping, another key objective could be to quantify the effect of considering a crop on a given crop period (instead of bare soil) on the consecutive crop period in terms of yield or soil moisture content (typically what is the lag effect of a crop development ?). For instance, on the four Indian points, do you improve the model predictability in terms of yield during the periods where wheat is grown when considering rice/wheat rotations (sequential cropping) compared to simulations with only wheat periods (and bare soil during the periods where rice is grown). The same for rice yield assuming bare soil instead of wheat development. You may use the same simulation set-up for looking at the impact on the model predictability for soil moisture at regional scale (see above).

Specific comments

- The manuscript presents the development of a sequential cropping capability in the JULES model. The authors define the sequential cropping as the cultivation of two or more crops on the same field in a given year (page 2 line 20). Later, the results of this new version are shown on a site in France (Avignon) where sorghum and wheat are grown. Page 7 line 1, it is written that sorghum is grown in summer and winter wheat in winter. I don't think this is correct. Winter wheat is sown in winter but grows over the spring up to early summer, while sorghum is sown in spring and grows up to late

summer (see figure 3). In this respect, Avignon is not a site where sequential cropping is applied but rather a site with the rotation of two crops (sorghum and wheat) over two consecutive years. This should be clarified in the manuscript. Although I have no doubt that Avignon is a super site where a huge set of measurements are performed, I don't clearly see the gain of applying JULES on this site when evaluating the sequential cropping functionality as it is not strictly speaking a site where sequential cropping is practiced. The motivations for using that site mentioned by the authors are "to illustrate that the new sequential cropping functionality in JULES can simulate more than one crop within a year and reproduce the correct growing seasons for each crop" (Page 8 line 9). In the results section it is also mentioned that "the aim of presenting this simulation is to demonstrate the method rather than provide a perfect representation of either of these crops" (Page 16 line 9). These objectives could be achieved by performing model simulations elsewhere than in Avignon, in regions where sequential cropping is commonly applied (like the region of India you focus on in the manuscript for instance). Also, about the model simulations performed for the Avignon site, I don't clearly understand the need/interest of the AviJUL-grass simulation which is driven by observations for LAI and the canopy height.

- Some figures would need some improvements. Especially, avoid the repetition of a same information in any sub-panel of a same figure. This is the case on Figure 3, Figure 4, Figure 5 about the location (Avignon). You can simply mention once at the top of the figure that it is for Avignon (or only in the Figure legend). The same with "India Points" for Figures 6, 10, 11, B2, B3, ... There are also redundancies between the information on the top of some subpanels and the information on the y-axis legend : on Figure 3 (total above ground biomass, LAI, canopy height), on Figure 4, Figure 5, Figure 6, ... Please specify one information at only a single location in one panel. On the other hand, there is information missing about variable units on Figure 4. Units of GPP, Latent heat flux and sensible heat flux are not specified at any place in the figure and not in the figure legend. The same on Figure 5 for "available moisture".

Technical comments

Page 3 line 11: Maybe explain what are the kharif and rabi seasons as they are quite specific terms.

Page 5 line 20: Define DVI before to use it.

Page 7 line 7: "between models". Which models do you refer to ?

Page 7 line 19: "The simulations are divided ...". Please, rephrase: The description of the simulations is divided ....

Page 7 line 19: "Section 4.1 applies the method...". Please, rephrase: Section 4.1 presents how the method is applied..."

Page 7 line 23: Please define PFT before to use it.

Page 7 lines 26 to 30: Provide units to the variables and parameters used (vcmax, neff, nl, mu_rl, mu_sl) and a more physiological meaning to them.

Table 2: Could you clarify the value of 1 for Q10. Does it mean that Vcmax is insensitive to temperature ?

Page 8 line 16: Could you clarify the use of a spring wheat parametrization to represent the C3 winter wheat crop at Avignon. Especially regarding what is mentioned later for India Simulations Page 8 line 33 (the wheat varieties grown in this region are spring wheat, this is an important distinction as spring wheat does not require a vernalization period which is important for winter wheat varieties. Does the wheat variety sown in Avignon in winter need a vernalization period or not ?

Page 9 line 7: Map (b) in Figure 2

Page 10 line 23: include measurements of soil moisture

Page 10 line 31: Citations need parenthesis.

Page 11 line 24: The sentence "For 2008 and 2012..." has the same meaning than the

previous sentence.

From Page 11 line 28 to Page 12 line 1: Should be put in the Methods section

Page 13 line 4 (and table 6): Provide units to the values of RMSE and Bias

From Page 16 line 1 to Page 17 line 6: This paragraph should be moved to the Discussion or Conclusion sections.

Figure 7: When comparing harvested biomass from JULES to the two observation-based estimates, I think that it would be more suitable to present the model/data comparisons with scatter plots. It will better highlight the model capacity at simulating observed interannual variability than using time-series.

---

## Referee Comment (RC3) · Anonymous Referee #3 · 18 Jun 2019

Mathison et al. presented the work enabling the JULES model to simulate sequential cropping system. While it is important for land surface models to account this process and the model performance looks reasonable, the model developments presented and the tests run are not yet informative to the community. I have several major concerns on the manuscript.

As a manuscript for GMD, model developments should be carefully and clearly presented. However, the section described the model development only took about half page with a simple flowchart, which is also not very informative. Even the settings of the simulations are described in more detail. In addition, all the equations presented are developments made by previous studies. All these make the reviewer think the manuscript is more like a model application rather than a model development.

Although the authors presented a lot of figures, many of them are not central to questions in which readers may be interested. The authors show the model performance accounting sequential cropping, but how does it compare with the one not accounting sequential cropping? What will be the difference if simulating two seasons of crops as two tiles? Will the LAI be different? Will the yield? At least to this reviewer, the authors fail to prove the improvements brought to the land surface model.

The authors set a "deadline" to harvest a crop even if it is not mature, in order to facilitate the next season of crop. However, it is hard to imagine this is a reasonable manner to simulate farmers' behavior. Will farmers cut down their crop grown for several months just for growing a new season of crop? Large scale applications of sequential cropping may face a lot more challenges than the few tested sites here. It is not yet convincing that the model is ready for larger-scale application at its present form.

---

## Author Comment (AC1) · 25 Jul 2019

**Authors initial response to reviewers**

The authors would like to thank all three reviewers for their comments on the manuscript. I have responded to each of the comments in turn.

**1.0    Reviewer 1**

**1.1    Response to general comments from reviewer 1**

In response to the general comments from reviewer 1, I will separate out the results section into three sections: results, discussion and conclusions. I will also clarify the reasons for what is being discussed. I did not include the regional simulations in this paper as they have not been fully evaluated yet. However, I will include a clear explanation of how these point simulations have informed the regional simulations and include some prototype regional simulations to show how the method is extended from point to regional runs.  I will also include reference to an earlier paper that includes a method to derive sowing and harvest dates for this region from the Asian Summer Monsoon. This is a simple method that does not require a lot of observed data.  I will also refer to the ICRISAT crop area data set which will facilitate the calculation of where the crops are grown.

**1.2    Response to individual comments**

1.       P 4, L 22-24: I believe there is a lot of discussion in section 6. Either the discussion should go in section 7 or relabel the sections to reflect that. Also, section 5 is not listed here and should be for consistency.

This is addressed in the Section 1.1 above. We will amend the manuscript to have results, discussion and conclusion sections and thank you for pointing out that section 5 is missing from this list, this will also be corrected.

2.       P 5, L 6: Can this be clarified to indicate air or leaf temperature?

We will clarify this in the text, in JULES this is air temperature.

3.       P 5, L 23: Please include the reference for the dataset(s) that were used for the sowing and harvest dates when observations weren't available?

We will include this in the text. We refer to a method from a previous paper that estimates the sowing and harvest dates based on the Asian Summer Monsoon published in ESD (Mathison et al 2018).

4.       P 6, L 8-9: Is forcing a harvest before maturity is reached realistic? Did any of the model simulations need to use this constraint any year?

This is a safeguard built into the model in order that the model can move onto the next crop in a clear and clean way. This latest harvest date is usually set to a date well after the expected harvest date and therefore the crop is expected to be harvested by this time. If the model is working correctly this should not be needed, however if it is used then the user knows that the harvest has been triggered because the crop has not matured. This means the user knows when the model is not

working correctly and can investigate this further. The latestharvestdate was not needed in either the point or the regional simulations that will be presented in the new manuscript. However, this safeguard, is preferable to the simulation of a crop growing for an unrealistically long time and overlapping into next growing season, thereby disrupting the simulation of the second crop. The latestharvestdate is particularly useful for checking if the sequential cropping is being implemented correctly on a regional or global scale. JULES is run for a wide variety of environments and conditions, with some quite different to where the model has been developed and tuned. This variety of the uses of JULES mean that it important to make sure that the crops are running properly and being harvested as expected. I will include explanation of this in the manuscript and explain that this is ideally not used. When it is needed it allows any problems to be properly investigated.

5.  P 6-7, Section 3: The DVI discussion might be better in section 2 where the model description is provided

The paragraph on DVI in this section will be incorporated into the previous model description section.

6.  P 7, L 6-7: What do you mean when you state the effective temperature differs "between models"?

The effective temperature is the function that the model uses to relate temperature to the cardinal temperatures that define when a plant develops. Different models define their effective temperature function in different ways. This is described in Wang et al 2017. I will explain this more clearly in the text and include this reference.

7.  P 7-8, Section 4: The authors should expand on how these specific variables are important for the crop model and how they influence the variables that are being compared with observations (e.g., GPP, latent heat flux, sensible heat flux).

Explanation will be added to the text.

8.  P 8, Section 4.1: The authors mention that the Avignon site is in France in the introduction. Location information (including approximate latitude and longitude) would be good in this section.

This will be added to the text.

9.  P 10, Section 5: It would be useful to have some climate information for all the sites. While some information is provided for the India sites in Figure 2, Section 6.2 and Figure 6, similar information should be included for the Avignon site.

I will add some climate information for Avignon to this section.

10.  P 11, Section 6: There is a lot of discussion in this section, which should just be results.

See section 1.1 and reply to comment 1 in this section.

11.  P 11, L 5: I do not see AviJUL-grass results in Figure 3.

AviJUL-grass results are not shown in this Figure, primarily because LAI and canopy height are prescribed in these simulations and would therefore follow the observations exactly. I will explain this more clearly in the text.

12. P 11, L 10-27: I do not think it is very surprising that the simulations do not capture the observed biomass and LAI. The crops in JULES are corn and spring wheat whereas the crops at the Avignon site are sorghum and winter wheat. The fact that the model did well for the 2009 sorghum season, which was an anomaly for that crop at the observed site, indicates the crop parameters need to be tuned for a true comparison. I'm not suggesting the authors need tune the model for this manuscript, however, the different crops could play a large role in the model performance, which should have a place in the discussion.

This will be included in the redrafted discussion section

13. P 11 L 28- P12: The discussion for p0 comes completely by surprise. The authors don't explain how p0 affects GPP and what the different values represent. This part feels like a discussion and doesn't belong in the results section, considering these additional water sensitivity simulations weren't discussed previously. The additional simulations with modified p0 do not add value to the results and I suggest they be removed (along with Figure A5). Then suddenly, the paragraph changes gears, jumping to GPP. The GPP results includes a grass simulation comparison, which I find unnecessary. The focus should be comparing the crops simulations and observations to be consistent with the rest of the section 6.1. The paragraph needs some editing and is written sloppily. For example, in L 15, "…early the decline is very close to the observations" does not make sense. Finally, delete the last paragraph on page 12.

This is noted and will be addressed in the redrafted results and discussion sections

14. P 14 L4-5 – P 15: This paragraph is confusing. It is not clear how soil moisture avail-ability factor (beta) is calculated for the model or observations, nor is it clear what the impact has on simulations, other than an early decline in GPP. What is not discussed is that part of the problem could be related to how irrigation is handled in the model. It was stated in the beginning of the manuscript that the irrigation for sorghum was included in the rainfall data, but it was not specified how. If the timing of irrigation is off,that could explain some of the water stress for sorghum that is seen in the model but not in the observations (figure 5). Regardless, I think this is taking away from the main point of the results and I do not feel it is necessary. If it is kept, it should be moved to section 7.

This is noted and will be addressed in the redrafted results and discussion sections

15. P 15, L6: Do the "four levels" refer to four soil layers in JULES?

Yes, in these simulations there are four soil layers in JULES at 0.1, 0.25, 0.65 and 2.0 m so that the bottom layer is at 3.0 m. I will clarify this in the text.

16. P 16-17: "the aim of this simulation is to demonstrate the method rather than provide a perfect representation of either of these crops." Yes – I agree with this statement. This should be the focus of section 6.1 – how this method improves the crop simulations. All the discussion about soil moisture and water stress muddle this point. Highlight positives of the approach and mention the tuning discussion for future work.

This is noted and will be addressed in the redrafted results and discussion sections.

17. P 19, L 17: the decline in NPP for wheat is hard to see in the figure.

The NPP shown here is gridbox NPP which declines during the wheat season shown by the yellow blocks on Figure 10 (c), this happens about two thirds of the way through the season. I will clarify this in the text. If this does not help, plots per tile are also available (similar to Figure B1) which show NPP and GPP for each tile (rice and wheat) in two colours (red and black)) for each location on a separate plot. I will establish which of these options is clearer and amend the manuscript accordingly.

18.     P 20, L 20-33: The VPD discussion here belongs in section 7, it is not results, but discussion. Furthermore, it is not known what the effect of VPD on is on the model yields. The discussion on the plant response to VPD is difficult to follow and clearly does not belong here.

This is noted and will be addressed in the redrafted results and discussion sections

19.     P 21: It is unclear in this discussion how much of this is related to poor wheat parameterization in JULES or how much is related to the sequential cropping component.

This will be clarified in the manuscript. This discussion is suggesting that the way that carbon is allocated between carbon pools for wheat is worthy of investigation into the lower yields for wheat.

**1.3     Technical Comments:**

1.      Define all acronyms (for example, GPP is never defined; DVI is not defined until Appendix B)

This will be corrected in the manuscript

2.      P 7, L 4: switch the ";" to a ":"

This will be corrected in the manuscript

3.      P 19, L35: the sentence beginning "Therefore suggesting it is not water stress..." is a fragment.

This will be corrected in the manuscript

4.      P 20, L 11-12: "in the simulations" is used twice.

This will be corrected in the manuscript

5.      P 20, L 12 and L14: Use first and second rather than "firstly" and "secondly"

Noted. This will be corrected.

6.      P 27, L2-3: The sentence beginning "In general the model produces..." is unclear.

This will be addressed in the redrafted results and discussion sections

7.      Figures need consistent labeling. Put units on the y-axis. Some figures have the y-axis units in the title, others don't have any at all. Also, in Figures 3,4,5,A1,A2,A3,A4,A5 the label "modelled soil moisture and LAI" is confusing. Shouldn't it be AviJUL_sqcrop?

All figures will be checked for consistent labelling and units. The labels will be changed to AviJUL-sqcrop etc for clarity and consistency with the text

8.      Figure A1: What four simulations are you referring to in the caption?

This is a missed deletion. This is an older caption and will be updated in the manuscript.

**2.0    Reviewer 2**

**2.1    Response to general comments from reviewer 2**

I have summarized the comments from reviewer 2 to respond to them.

1.      The manuscript lacks key significant objectives and the rest of the manuscript strongly suffers of this absence.
2.      The manuscript really needs more challenging scientific objectives.
3.      What's the added value of representing the sequential cropping on a given variable or process?

The manuscript does have significant objectives, and these are now better articulated in the introduction. Sequential cropping provides clear added value for two reasons:
   i.      to improve simulations of water resources where this type of cropping system is in use. This cannot be done if the model only simulates one crop per year because this does not happen in the real world.
   ii.     including sequential cropping in models is also a more realistic representation of the land surface in terms of land cover and therefore fluxes. The climate affects both the water and crops in this standalone setup, while simultaneously allowing the interactions between water and crops across the year.
These interactions are important for understanding water resources. I will clarify these points in the text.

4.      I would strongly encourage the authors to include these regional simulations in the present study. Performing such regional simulations for present-day conditions, with and without the sequential cropping (ie with only one of the two crops grown within a year) would enable to investigate the impact of the sequential cropping on the vegetation intensity and the soil moisture at regional scale and to directly compare these modelled variables with observation-based data provided by remote-sensed products (LAI (or fPAR) and soil moisture from satellite sensors such as MODIS or GRACE for instance). This could be performed on the studied region of North India.

As suggested by the reviewer in future work, I intend to use the GRACE, MODIS and SMOS satellite data for evaluation of regional simulations but this is probably beyond the scope of this paper which is primarily to present the method and model developments. However, I will include some prototype results for the regional simulations that cover the same Uttar Pradesh and Bihar region as the point simulations. This should help demonstrate the application of the method for a region.
I also considered using MODIS LAI in this paper for checking the crop seasons, however MODIS doesn't include crop specific information, so the LAI observed might be for a different crop to the one being modelled in JULES.

**2.2       Response to specific comments from reviewer 2**

1.        The manuscript presents the development of a sequential cropping capability in the JULES model. The authors define the sequential cropping as the cultivation of two or more crops on the same field in a given year (page 2 line 20). Later, the results of this new version are shown on a site in France (Avignon) where sorghum and wheat are grown. Page 7 line 1, it is written that sorghum is grown in summer and winter wheat in winter. I don't think this is correct. Winter wheat is sown in winter but grows over the spring up to early summer, while sorghum is sown in spring and grows up to late summer (see figure 3). In this respect, Avignon is not a site where sequential cropping is applied but rather a site with the rotation of two crops (sorghum and wheat) over two consecutive years. This should be clarified in the manuscript. Although I have no doubt that Avignon is a super site where a huge set of measurements are performed, I don't clearly see the gain of applying JULES on this site when evaluating the sequential cropping functionality as it is not strictly speaking a site where sequential cropping is practiced. The motivations for using that site mentioned by the authors are "to illustrate that the new sequential cropping functionality in JULES can simulate more than one crop within a year and reproduce the correct growing seasons for each crop" (Page8 line 9).

I will clarify this detail regarding Avignon in the manuscript. The flexibility of our method allows it to be applied to both a strict sequential cropping system and one that is more irregular. However, this method is still operating in a similar way to a pure sequential cropping system at this site because in some years Sorghum is planted first and is immediately followed by wheat within a month or two. On this basis it is still an appropriate site for demonstrating this method, valid for both crop rotations and sequential cropping.

2.        In the results section it is also mentioned that "the aim of presenting this simulation is to demonstrate the method rather than provide a perfect representation of either of these crops" (Page 16 line 9). These objectives could be achieved by performing model simulations elsewhere than in Avignon, in regions where sequential cropping is commonly applied (like the region of India you focus on in the manuscript for instance). Also, about the model simulations performed for the Avignon site, I don't clearly understand the need/interest of the AviJUL-grass simulation which is driven by observations for LAI and the canopy height.

The purpose of including Avignon is because it provides a wealth of observations for evaluating Land surface models. Without observations of these fluxes there is no way of knowing if the model is correctly representing the fluxes and the coverage of the land surface. The purpose of including a simulation that does not use the crop model but approximates crops using grasses is to show how the model performs with the correct LAI and height i.e. it is a clean test of the representation of leaf photosynthesis, stomatal conductance, water stress and leaf-to-canopy scaling within the model (these parts of the code are shared by both natural vegetation and crops). There is also no equivalent site for South Asia.

3.        Some figures would need some improvements. Especially, avoid the repetition of a same information in any sub-panel of a same figure. This is the case on Figure 3, Figure 4, Figure5 about the location (Avignon). You can simply mention once at the top of the figure that it is for Avignon (or only in the Figure legend). The same with "India Points" for Figures 6, 10, 11, B2, B3, ... There are also redundancies between the information on the top of some subpanels and the information on the y-axis legend: on Figure 3 (total above ground biomass, LAI, canopy height), on Figure 4, Figure 5, Figure 6, Please specify one information at only a single location in one panel. On the other hand, there is information missing about variable units on Figure 4. Units of GPP, Latent heat flux and sensible heat flux are not specified at any place in the figure and not in the figure legend. The same on Figure 5 for "available moisture"

These comments have been noted and as outlined in the responses to reviewer 1, all figures will be reviewed and checked for consistency, units and titles.

**2.3    Technical comments**

1.       Page 3 line 11: Maybe explain what are the kharif and rabi seasons as they are quite specific terms.

The kharif season is the monsoon season and the rabi season is the dry season in South Asia. This will be clarified in the text

2.       Page 5 line 20: Define DVI before to use it.

This means Development Index which is first referred to in Equation 3 of Section 2although it is not explicitly defined here, so we will add this to the text.

3.       Page 7 line 7: "between models". Which models do you refer to ?

Different models define the effective temperature function in different ways. In Wang et al (2017) various temperature response functions are shown in Figure 1, In this paper JULES is most similar to type 4. Each have a different effect on the development of crops. In JULES the temperature response function defined by the effective temperature increases gradually to the optimum with a steeper decline toward the maximum. Other models have a flatter top to this function and others have no decline above the optimum. This will be clarified in the text.

4.       Page 7 line 19: "The simulations are divided ...". Please, rephrase: The description of the simulations is divided ....

This will be rephrased

5.       Page 7 line 19: "Section 4.1 applies the method...". Please, rephrase: Section 4.1 presents how the method is applied..."

This will re-phrased in the text

6.       Page 7 line 23: Please define PFT before to use it.

PFT means Plant functional type which is defined in the caption of Table 2, which was subsequently moved to the end of the manuscript during editing before submission. This definition will be added to the text of the manuscript as well.

7.       Page 7 lines 26 to 30: Provide units to the variables and parameters used (vcmax, neff,nl, mu_rl, mu_sl) and a more physiological meaning to them.

$V_{cmax}$ :    Maximum rate of carboxylation of Rubisco ($mol\ CO_2\ m^{-2}s^{-1}$)
$n_{eff}$ :    Scale factor in the $V_{cmax}$ calculation (mol $CO_2$ m$^{-2}$ s$^{-1}$ kg C (kg N) $^{-1}$)
$n_l(0)$:    Mass of Nitrogen per mass of Carbon at the top of the canopy kg N (kg C) $^{-1}$

These two parameters affect the respiration of the plant.

$\mu_{sl}$ :     Ratio of stem nitrogen concentration to leaf nitrogen concentration, i.e the mass ratio of nitrogen to carbon in the stem divided by the ratio of nitrogen to carbon in the leaves

$\mu_{rl}$ :     Ratio of root nitrogen concentration to leaf nitrogen concentration, i.e. the mass ratio of nitrogen to carbon in the roots divided by the ratio of nitrogen to carbon in the leaves

This will be added to the text

8.      Table 2: Could you clarify the value of 1 for Q10. Does it mean that Vcmax is insensitive to temperature?

$Q_{10\_leaf}$ : $Q_{10}$ value for carboxylation of Rubisco used in the $V_{cmax}$ calculation.

No, there is still a dependence on temperature ($T_{low}$ and $T_{upp}$). These relationships will be clearer by showing the equations in the manuscript. In response to the reviewer's points 7 and 8, I will include the equations and explanations for these parameters in the text in section 2. These were omitted from the initial submission to avoid further repetition of previous publications but including them will help explain the parameter choices.

9.      Page 8 line 16: Could you clarify the use of a spring wheat parametrization to represent the C3 winter wheat crop at Avignon. Especially regarding what is mentioned later for India Simulations Page 8 line 33 (the wheat varieties grown in this region are spring wheat, this is an important distinction as spring wheat does not require a vernalization period which is important for winter wheat varieties. Does the wheat variety sown in Avignon in winter need a vernalization period or not?

In the real world the fact that winter wheat requires a vernalization period and spring wheat does not is an important distinction between the two varieties. However, in JULES temperatures cannot damage any of the crops being modelled, either by being too high, too low or not low enough. When temperatures are outside the optimum range for development in JULES, development is slowed down but as soon as the temperatures return to within the optimum range the crop continues its development from where it left off.  The lack of vernalization does not therefore cause the yields to completely fail in JULES as they do in reality.  While this process is missing from JULES, it is acceptable to approximate winter wheat using spring wheat. I will clarify this in the text and describe the implications for the results.

10.     Page 9 line 7: Map (b) in Figure 2

Fig will be added prior to the number of the figure in this sentence.

11.     Page 10 line 23: include measurements of soil moisture

This sentence will be corrected.

12.     Page 10 line 31: Citations need parenthesis.

These citations are part of the text and therefore do not need parentheses. However, I will add punctuation as this should clarify that these references are part of the text.

13.     Page 11 line 24: The sentence "For 2008 and 2012..." has the same meaning than the previous sentence.

The clearer sentence will be retained and the other deleted.

14.     From Page 11 line 28 to Page 12 line 1: Should be put in the Methods section

As mentioned in response to reviewer 1, Section 6 and Section 7 will be redrafted into separate results, discussion and conclusions sections. This paragraph will be moved to the methods section or most appropriate section as part of this redrafting.

15.     Page 13 line 4 (and table 6): Provide units to the values of RMSE and Bias

The manuscript will be updated to include this information.

16.     From Page 16 line 1 to Page 17 line 6: This paragraph should be moved to the Discussion or Conclusion sections.

As mentioned in response to reviewer 1, Section 6 and Section 7 will be redrafted into separate results, discussion and conclusions sections.

17.     Figure 7: When comparing harvested biomass from JULES to the two observation-based estimates, I think that it would be more suitable to present the model/data comparisons with scatter plots. It will better highlight the model capacity at simulating observed interannual variability than using time-series.

I am not sure there are enough observations of biomass for the scatter plot to work. However, I will investigate alternative ways of displaying this information for the updated manuscript

**3.0     Reviewer 3**

**3.1     Response to general comments from reviewer 3**

1.     The section described the model development only took about half page with a simple flowchart, which is also not very informative. Even the settings of the simulations are described in more detail. In addition, all the equations presented are developments made by previous studies. All these make the reviewer think the manuscript is more like a model application rather than a model development.

JULES needed substantial code changes for the implementation of the new double cropping scheme. The inclusion of double cropping within JULES is a necessary development for applications which require simulation of the annual cycles of water and carbon fluxes in many regions of the world, including India. The addition of this functionality can therefore can be placed in the category of model development. The relevant equations from existing parts of JULES were reproduced here for completeness with the flowchart clearly showing the way that the crop rotation is done within JULES. There was no need for anymore text regarding model development.

2.     Although the authors presented a lot of figures, many of them are not central to questions in which readers may be interested. The authors show the model performance accounting sequential cropping, but how does it compare with the one not accounting sequential cropping? What will be the difference if simulating two seasons of crops as two tiles? Will the LAI be different? Will the yield? At least to this reviewer, the authors fail to prove the improvements brought to the land surface model.

One of the main reasons for including sequential cropping is to improve simulations of water resources across a single monsoon season. The very fact that the land surface is more representative of reality with two crops growing in sequence representing the real coverage of the land will produce more realistic fluxes and interactions with the atmosphere. This is important for the simulation of water resources and the interactions between water and agriculture. ISIMIP adopt the approach that assumes a small amount of each crop in each of the gridboxes and scale up. Even using this approach, it is difficult to see the whole effect this would have on the fluxes. Including the double cropping is to look in a wholistic way at the land-surface coverage and the effects this has on the fluxes for a region.

3.      The authors set a "deadline" to harvest a crop even if it is not mature, in order to facilitate the next season of crop. However, it is hard to imagine this is a reasonable manner to simulate farmers' behaviour. Will farmers cut down their crop grown for several months just for growing a new season of crop?

This is a model safeguard and ideally would not be used, please see response to reviewer 1 in section 1.2 point 4.

4.      Large scale applications of sequential cropping may face a lot more challenges than the few tested sites here. It is not yet convincing that the model is ready for larger-scale application at its present form.

Regional simulations have been completed so the model, is to some extent, ready for large-scale application. In the previous response to reviewer 2 (section 2.1, point 4), I will be including recently completed prototype regional simulations as a demonstration of applying this method to a region. However, the reviewer raises an important point regarding the many possible choices for the regional crop simulations, regarding for example, the rotations, timings and crops. I will make the assumptions clear in the text and explain any limitations.

---

## Author Response (AR1)

**1 AUTHORS RESPONSE TO THE REVIEWS**

Thank you to all the reviewers for their comments. These have now been incorporated into a thoughtfully and extensively modified manuscript. An earlier form of this paper has formed part of my PhD thesis and I have also incorporated the changes suggested by my examiners. In my initial response to the reviews, on the open discussion, I responded to the reviewer comments with my intended changes to the manuscript. I have not repeated these here, as this document is still available on the discussion, should it be needed. Instead, Section 2 is a list of the major changes that have now been made to the manuscript; these include regional simulations for Uttar Pradesh and Bihar and considerable restructuring of the results, discussion and conclusions sections. All the figures have also been standardized. I have not listed all the small changes, e.g. grammar or punctuation changes, as these are covered in the marked-up version of the manuscript provided in Section 3. The marked-up version of the manuscript in Section 3, shows the considerable changes that have now been made in response to the comments.

**2 A LIST OF ALL THE CHANGES TO THE MANUSCRIPT**

The changes are listed by section and line number, where appropriate.

Abstract: The abstract includes mention of the regional run and some tidying up of the language

- 1. Introduction:
  - Reordered the section to discuss the South Asia economy prior to the intercropping and sequential cropping.
  - Clarified the objectives and the reasons for including sequential cropping
  - Clarified the type of cropping system used in Avignon and India.
  - Clarified why the Avignon site is used and the reason for including the AviJUL-grass simulation
  - Explained what the kharif and rabi season are.
- 2. Model description
  - Added more information to the model description to aid the explanation of the parameters that had been modified for use in this study.
  - Included more equations to aid the explanation.
  - Removed description of the photoperiod (RPE) as this is not used and is therefore unnecessary.
  - Moved the DVI description to this section from Section 3.
  - Included reference to Wang et al 2017 to explain regarding the different effective temperature definitions and which one JULES is most like.
  - Explained more fully why we have implemented the safeguard of the 'latestharvestdate'.
  - Included description of GPP and NPP
- 3. Method for sequential cropping in JULES
  - Clarified the starting point for the method.
  - Added season information to the Avignon crops
  - Clarified that the 'latestharvestdate' safeguard is set but is not used in this study because it is not required unless something has gone wrong and the user needs to be alerted.
- 4. Model simulations

- Clarified the crop seasons for growing crops at Avignon and the type of rotation in use.
- Added information on the parameters that were modified and the impact they have on the crops.
- Included difference between winter wheat and spring wheat (vernalization) and why this is not a problem for JULES because this aspect of wheat is not modelled explicitly.
- Included explanation of the setting of parameter to reduce soil moisture stress (P0)
- Added clarification regarding the use of the Avignon site.
  - 4.1 Avignon site simulation
    - Clarified why spring wheat is used to model Avignon winter wheat and maize used for both sorghum types
    - Clarified why AviJUL-grass is used
  - 4.2 India simulations
    - Small changes to explain set up of regional simulations in addition to the point simulations.
- 5. Observations
  - Mostly small changes to this section
- 6. Results
  - 6.1 Avignon site results
  - Added information on Avignon climate and included a figure with the temperature and rainfall (including irrigation); this is now Fig 3.
  - Modified Fig 4 so the units are only on the y axis and labelling is consistent. Included the uncertainty in the observations on this figure. Modified the caption to reflect this. Each of the panels now has a legend because the labels were different for each panel.
  - Soil moisture is now discussed earlier in the setting up of the simulations. By modifying PO soil moisture stress is no longer relevant to this paper.
  - Moved some content into a new discussion section.
  - Removed Fig. A.5 from Appendix A as it is no longer discussed.
  - 6.2 India point results
  - Introduced a new section for the India point results
  - Replaced Fig. 7 with a scatter plot and put the timeseries in the Appendix (Fig. B.1)
  - Moved some content into a new discussion section.
  - Modified Fig. 10 and Fig. 11 to be annual climatology which is easier to read and moved the timeseries into the Appendix (Fig. B.5 and Fig. B.6)
  - Removed surplus Figures from the Appendix B.
  - 6.3 India regional results
  - Introduced a new section for the India regional results
  - This is new requested content, includes new plots and text.
- 7. Discussion

There is now a new section for discussion that is separate from the results and conclusions.

- 7.1 Avignon discussion
  - This is a new discussion section which includes new content and some that was in the results section of the previous draft.
- 7.2 India discussion
  - This is a new discussion section which includes new content and some that was in the results section of the previous draft.
  - This section also discusses the new regional India simulations.
- 8. Conclusions

- Redrafted now that there is a separate section for results and discussion as well.

**3 A MARKED-UP VERSION**

**Developing a sequential cropping capability in the JULESvn5.2 land–surface model**

Camilla Mathison1,2, Andrew J Challinor2, Chetan Deva2, Pete Falloon1, Sébastien Garrigues3,4, Sophie Moulin3,4, Karina Williams1,5, and Andy Wiltshire1,5

[revised manuscript text omitted]

---

## Referee Report (RR1)

[referee-annotated manuscript omitted]

---

## Author Response (AR2)

**Authors response to reviewers**

Many thanks to the three reviewers for reviewing the manuscript. Reviewer 1 is happy with the manuscript, with further changes requested from reviewers 2 and 3. This response is to each of reviewers 2 and 3, in turn.

**1.0 Response to reviewer 2:**

The comments from this reviewer are in blue and the authors responses are provided in black. Thank you for recognising the work that has gone into the previous submission. The major comment in this review was as follows

The absence of comparison of simulations with and without the sequential cropping. As mentioned in my first report, this is the only way to quantify the added value of representing the sequential cropping on a given variable or process. This would need the provision of simulations with a model configuration without the sequential cropping (ie with only one crop growing within a year), to be used as a reference. This remark has also been done by one of the two other referees of the original manuscript ("The authors show the model performance accounting sequential cropping, but how does it compare with the one not accounting sequential cropping? What will be the difference if simulating two seasons of crops as two tiles? Will the LAI be different? Will the yield? At least to this reviewer, the authors fail to prove the improvements brought to the land surface model").

The authors replied to my comment by mentioning that "including sequential cropping in models is [...] a more realistic representation of the land surface in terms of land cover and therefore fluxes. And to the other referee by mentioning that "The very fact that the land surface is more representative of reality with two crops growing in sequence representing the real coverage of the land will produce more realistic fluxes and interactions with the atmosphere." That is certainly true that the sequential cropping is more representative of reality (for regions where it is a common practice) but the "translation" into more realistic fluxes is rather an assumption which is stated in several places in the revised manuscript, for instance page 29 line 27. I think it is needed to move from that assumption to a quantitative assessment of the impacts of the sequential cropping on the yield, lai, soil moisture, ... This could be done as a sensitivity study over the 4 Indian points and the Indian regions simulations. To my opinion, the lack of this quantitative assessment is critical and detrimental to the present study.

We have conducted a quantitative assessment of the impacts of the sequential cropping on the crop, as suggested. Single crop simulations have been added to the point simulation at Avignon and the single gridbox simulations for the four India locations. We now show the sequential crops and single crop simulations both alongside each other and on the same plot to show what difference the sequential cropping makes to the simulation. For Avignon, we show the impact of sequential crops on the crop development and growth (see Fig. 5) and fluxes of heat and carbon (see Fig. 6 and in Appendix A, Figs A.1, A.2, A.3 and A.4). The Avignon site is used to compare against observations for the impact of sequential crops. For India we compare the same crop development and growth variables as for Avignon for the sequential and single crop simulations (see LAI in Fig. 9 and canopy height in B.3), but we only have yield observations for comparison against observations (see Fig. 8). The fluxes are shown for single and sequential crop simulations side by side in Fig 10 (carbon) and Fig 11 (energy). In Appendix B, Figs B.6, B7, B.8 and B.9 show these fluxes for each of the four locations separately, thereby showing a clear comparison between sequential and single crops.

The effect of sequential crops on soil moisture is more difficult to separate from the effect of bare soil evaporation at Avignon due to the irregular cropping system used there. However, the regular cropping system at the India locations is less likely to be affected by long periods of bare soil, so for these locations we also look at the effect of sequential cropping on soil moisture (see Fig 12 and Appendix B, Figs B.6, B.7, B.8, B.9, e and f). For the India single gridbox simulations, we also show transpiration (g) and non-transpiration (h) fluxes at each location, to look more closely at the impact of sequential crops on the evapotranspiration and non-evapotranspiration components of latent heat flux.

In the introduction we include a subsection (Sect. 1.2) clarifying the motivating factors in this study, to have a crop-modelling application suitable for use in atmospheric models thus precluding any post-processing solution that merges model runs. We have restructured the results and discussion to clearly reflect the clarified objectives of the analysis.

**1.1 Technical Comments**

Page 3 line 15: Give the meaning of LPJ-ml here, not line 17. "Lund-Potsdam-Jena managed Land model (LPJml) is one of ..."

**Corrected**

Eq 1: I'm surprised of the 2 temperature functions involved in the computation of Vcmax. What do they refer to ? Vcmax25 is not defined. I would assume it is the Vcmax value at 25°C but with the function [1+exp(0.3(Tc-Tupp)][1+exp(0.3(Tlow-Tc)], Vcmax does not equal Vcmax25 at 25°C. This is surprising.

**Eq. 2: if Q1O equals 1 for all crops you consider in your study, it is maybe simpler to remove fT from equation 1 and to remove eq. 2.**

Yes, we agree that it this notation is counterintuitive (since in general  $V_{cmax}$  at 25°C does not equal the parameter  $V_{cmax25}$ ). Therefore, we will combine equation 1 and 3, so that the misleadingly named parameter  $V_{cmax25}$  does not need to be defined. We would prefer to keep equation 2 since  $Q_{10leaf}$  is a free parameter in JULES and is not usually set to 1 in JULES runs (for example, in the paper that presents the JULES-crop model, Osborne et al, it is set to 2 for all crops).

**Page 9 line 3 to 11: this information about parameterization should be moved page 5 where equation 3 is described (at least the information that there is no nitrogen cycle in your model).**

The information regarding the nitrogen cycle and use with the crop model has been moved to the Model description section.

**Page 10 line 21: "the JULES", remove "the"**

Corrected

Page 13 line 17: Need parenthesis for the citations of Monfreda et al. (2008) and Ramankutty et al. (2008).

**Parentheses added**

Page 27 line 15: "very wet (500 mm of rain)". Page 13 it is mentionned that annual precipitation is 680 mm. Could you clarify ?

Added clarification. The quoted "very wet (500 mm of rain)" refers to the amount of rain that fell during one wheat season. This is a large proportion of the 687mm expected annually.

Page 28 line 6 to 8: Delete "In the following section we apply this same method to a range of locations that use the sequential cropping system in the north of India in order to implement this method for a regional tropical simulation."

Deleted

Page 28 line 27: Delete "in these simulations"

Deleted

Figure 10: X-axis is time in "day of year" not month

Corrected in Figure 10.

Figure 11: X-axis is time in "day of year" not month

Corrected in Figure 11.

Table 2: Add units to the parameters.

The units are provided in the description of each of the parameters in the third column of Table 2.

**2.0 Response to Reviewer 3**

The comments from this reviewer are in blue and the authors responses are provided in black. Thank you for recognising the work that has gone into the previous submission. The major comment in this review was as follows:

I have voiced my concern in the previous assessment that "the authors fail to prove the improvements brought to the land surface model." Some detailed suggestions were also made for authors to compare simulations with and without sequential cropping. The authors responded that "ISIMIP adopt the approach that assumes a small amount of each crop in each of the gridboxes and scale up", which is not ideal. I agree with it. Given this consensus, why do the authors not compare a simulation with ISIMIP approach and sequential cropping approach? This could clearly demonstrate whether sequential cropping makes some differences.

The prototype regional simulation provided has rather poor performance in many aspects. The authors may argue that it can come from several different sources not related with sequential cropping. However, for a study developing sequential cropping module, it is the authors' responsibility to demonstrate whether and to what extend the development has improved the simulations. To be more clear, I am not saying developing sequential cropping is not useful, but the authors have to prove it.

The readers cannot be convinced by current results and/or presentations.

In this revised manuscript we have attempted to address the reviewers main concern, to show the benefits of implementing sequential crops. We do this by showing the difference between the India gridbox and Avignon point simulations using the two methods available in JULES - single crops and sequential crops (see response to reviewer 1 for specific figures showing the effect of sequential cropping). The inclusion of sequential cropping is an important step in being able to couple the crop model to Earth System and Climate models for application in adaptation and mitigation studies, which is one of the primary motivating factors in this study. Given the choice would have to be only a single crop, we argue that the ability to capture sequential cropping is in itself a significant improvement.

We do not compare with the ISIMIP approach because this is a postprocessing method to estimate yields. The focus of this study is towards a full representation of the crop. Postprocessing methods

are therefore out of scope. Instead, the quantitative assessment of the impacts of the sequential cropping on the crop (see Reviewer 1) address the main issue identified here: that the benefit of sequential cropping is not demonstrated.

We have now more clearly defined what the objectives of this analysis are to try to make this clearer (see Sect.1.2) and restructured the results and discussion to reflect these objectives. Through comparison of single and sequential crop simulations, in this revised manuscript, we show what difference sequential crops makes to the crop development and growth and the fluxes of energy and carbon. As mentioned in our response to reviewer 1, for the regular India rotation we also consider soil moisture.

Many thanks for taking the time to review this manuscript

Camilla Mathison (on behalf of the authors)

**Developing a sequential cropping capability in the JULESvn5.2 land–surface model**

Camilla Mathison1,2, Andrew J Challinor2, Chetan Deva2, Pete Falloon1, Sébastien Garrigues3,4, Sophie Moulin3,4, Karina Williams1,5, and Andy Wiltshire1,5

[revised manuscript text omitted]
_{\rm cmax} = \frac{n_{eff} n_l(0) f_{\rm T}(T_{\rm c})}{[1 + e^{0.3(T_{\rm c} - T_{\rm upp})}][1 + e^{0.3(T_{\rm low} - T_{\rm c})}]}$$
(1)

$$f_{\rm T}(T_{\rm c}) = Q_{10_l eaf}^{0.1(T_c - 25)} \tag{2}$$

where  $f_{\rm T}$  is the standard  $Q_{10}$  temperature dependence (given in Eq. 2) and  $T_{\rm c}$  is the canopy temperature.  $n_{eff}$  represents the scale factor in the  $V_{cmax}$  calculation (in units of mol CO2 m-2 s-1 kgC(kgN)-1) and  $n_l(0)$  the top leaf nitrogen concentration (in units of kgN (kgC)-1). More details regarding the calculation of  $V_{cmax}$  are provided in Clark et al. (2011) and Williams

et al. (2017)1.  $V_{cmax}$  is an important component in two limiting factors for photosynthesis; the Rubisco-limited rate and the rate of transport of photosynthetic products; Equation 1 shows the relationship between  $V_{cmax}$  and temperature. GPP Gross Primary Productivity (GPP) is used to describe the total productivity of a plant; this defines the gross carbon assimilation in a given time. Net Primary Productivity (NPP) is GPP minus plant respiration; NPP is used in the crop partitioning code and subsequently in the calculation of the yield in JULES. The nitrogen cycle in JULES cannot yet be used with the crop model,

so in this study the same assumption is made as in Williams et al. (2017), that crops are not nitrogen limited

5

 $V_{\rm cmax} = \frac{V_{\rm cmax25} f_{\rm T}(T_{\rm c})}{[1 + e^{0.3(T_{\rm c} - T_{\rm upp})}][1 + e^{0.3(T_{\rm low} - T_{\rm c})}]}$

[revised manuscript text omitted]

---

## Author Response (AR3)

**Authors initial response to reviews (August 2020)**

The authors would like to thank both reviewers for their comments on the manuscript. I have responded to each of the comments in turn.

**1.0 Reviewer 3**

**1.1 Response to general comments from reviewer 3**

Thank you for recognising the work that has gone into the latest draft addressing the previous review comments. In the following response, the review comments are quoted in blue and the author responses are in black.

1. Figure 5 is a great addition to the manuscript, but the single cropping lines and sequential cropping lines are often not clearly distinguishable.

Thank you, Figure 5 has been in each version of the paper, although you are correct in that it has had more information added to it over the course of review process. In this latest version, the single crop simulation has been added. I raise the difficulty in distinguishing the sequential and single crop simulations in this figure on Page 16 of this version of the manuscript. It is mainly due to the same parameterization being used to generate the wheat crop in both simulations, this means that both the sequential crop simulations and the single crop simulations produce a crop that are very similar making them difficult to distinguish from each other. To make this figure clearer, I will increase the density of the dots in the dotted single crop lines to make these more visible and make the overall figure larger.

2. The authors has not performed regional simulations with single cropping setting, which is a bit hard for me to understand why. I wonder how much differences the sequential cropping settings may induce to the regional simulations comparing with the ISIMIP approach. Could the authors at least discuss over it?

Thank you for your comment. Comparison of sequential JULES yields with ISIMIP yields would be an interesting comparison and one that I would like to investigate in future work. We would need to develop an ancillary based on the crop fractions used by ISIMIP in their postprocessing step which though worthwhile is unfortunately beyond the scope of this paper. The regional simulation was included here as a demonstration of applying this method at a larger scale, with the primary motivation being to include crops in larger coupled simulations where the surface fluxes can have a large impact on the regional climate. We show in this study that this method of including sequential crops has little impact on the yields simulated in JULES mainly affecting the surface fluxes, which was why single crop regional simulations were not included. We will discuss it in the discussion as part of the redraft requested by reviewer 4.

**2.0 Reviewer 4**

**2.1 Response to general comments from reviewer 4**

Thank you for agreeing to review this manuscript and for your constructive comments. The main comment is on the structure of the paper. I have responded to each comment in this document (see the annotated pdf). I have referred to each comment, quoting it in blue and giving its location within the current version of the manuscript, with author responses in black.
In this section I will address the general comments from this reviewer which are for a general restructuring of the paper and to shorten the results:

1. I suggest to restructure the sections into the following:
          1. Introduction
          2. The JULES-crop model
               2.1. Model description
               2.2. Implementing sequential cropping (including rational for sequential cropping)
          3. Model simulations
               3.1 Avignon, France simulations
               3.2 Uttar Pradesh and Bihar, India simulations
          4. Model evaluation
               Describe hypothesis in current 1.2 and observations to compare against.
               Where observations are input to model explain in new section 3.
          5. Results
               5.2 Crop growth and development
                    5.2.1 Avignon
                    5.2.2 India
               5.3 Energy and carbon fluxes
                    5.3.1 Avignon
                    5.3.2 India
          6. Discussion
          7. Conclusion

I will adopt the structure proposed by Reviewer 4, with some slight modifications. Section 1 will include a subsection *1.1 Modelling sequential cropping in land surface models*. Section 2 will include the proposed subsections, with 2.2 separated as follows:
       2.2.1 JULES-crop: rationale for implementing sequential cropping in JULES-Crop
       2.2.2 The sequential cropping method and the modifications made to JULES-crop
Section 3 will be as recommended above. Section 4 will have a section for each region which will describe the climate and the observations for evaluating the model.
Section 5 will be as recommended but with 5.1 showing the evaluation of the model and an additional section 5.4 for soil moisture.

2. Results: I suggest shortening the results section and using fewer figures: one figure per section as suggested in my comment on the overall structure.

Figures 4 and 7 will be moved to the appendix and the text redrafted to shorten it. The number of figures will be reduced.

3. Discussion: Too long, shorten the discussion section and focus on the main points only.

I will make the modifications to the discussion requested in the comments provided in Section 2.2 (12 and 13) and shorten this section.

4. Conclusions: Shorten the conclusion section to only the most relevant conclusions and recommendations for model development. The current section is a mix of summary and discussion with some interesting conclusions here and there.

I will make the modifications to the conclusions requested in the comments provided in Section 2.2 (14 and 15) and shorten this section.

**2.2 Response to specific comments from reviewer 4**

1. Title: why not simplify to "Implementation of sequential cropping into JULESvn5.2 land-surface model"?

The co-authors and I have no objection to changing the title, provided the editor is ok with this change

2. Page 1 Line 6-11: Comment includes removal of text shown by strike through and replacement text from the reviewer comment is in red:

In this paper we implement sequential cropping in a branch of the Joint UK Land Environment Simulator (JULES) and demonstrate its use at a site in France and India.  We evaluate Jules with sequential cropping at Avignon, France providing over 15-years of continuous flux observations. . We apply JULES with sequential cropping  to a regional and 4-single-gridbox simulations for the North Indian states of Uttar Pradesh and Bihar to simulate the regular rice–wheat rotation.

We will implement these suggestions and amend this text to clarify the runs done with JULES (see comment 4) and therefore lines 6-11 of the abstract will be modified to read as follows:

*In this paper we implement sequential cropping in a branch of the Joint UK Land Environment Simulator (JULES) and demonstrate its use at a site in France and India. We evaluate JULES with sequential cropping at Avignon, France providing over 15-years of continuous flux observations (a point simulation). We apply JULES with sequential cropping to a regional 25 km resolution gridded simulation for the North Indian states of Uttar Pradesh and Bihar and 4-single gridbox simulations across these states, where each simulation is a 25 km gridbox, to simulate the rice–wheat rotation.*

See comment 9 for further changes to the Abstract.

3. Page 1 Line 13: The meaning of irregular and regular crop rotation is unclear to me.

We refer to the rice-wheat rotation in India as a regular crop rotation because the two crops are planted at approximately the same time of the year, every year. Avignon is a typical Mediterranean crop succession, characterized by a succession of winter and summer crops and in between periods of bare soil. When a summer crop follows a winter crop, the period of bare soil can last up to 9 months. This is typical for Mediterranean region and it is not represented in regional climate models. Therefore, the crops at Avignon vary more, with long fallow periods between crops for some of the years. A previous reviewer suggested that Avignon was therefore not sequential cropping and this description was included to make it clear that though the two rotations were different they were still defined as sequential crop rotations. I will clarify what is meant by regular and irregular crops in the

manuscript. See comment 2 for the proposed changes to the abstract and comment 9 for proposed changes in the manuscript.

4. Page 1 Line 17: Unclear what this is. Please clarify the runs you do with Jules better in the abstract

Avignon is a site simulation; this means it is for one point where we use the surface parameters from the site characteristics and meteorological information from the site itself. In the regional and single gridbox simulations the model is implemented in its standard configuration using standard soil parameters and meteorological information from a regional climate model. The regional run is mentioned earlier in the abstract (on line 9), this is a gridded simulation for the states of Uttar Pradesh and Bihar. The India single gridbox simulations also cover the same area as the regional run, so we can see if running for a larger area changes the conclusions drawn from the 4-single gridbox simulations. I will clarify this in the Abstract, see the proposed text in the response to comment 2.

5. Page 3 Section 1.1 Additional text in the title of this section: 1.1 Modelling sequential cropping in global land use and vegetation models.

I think this comment is asking that the title should more accurately reflect the contents of this section. On this basis we will modify the title of this section to be *1.1 Modelling sequential cropping in land surface models.*

6. Page 3 line 14: Please cite also updated model description: Schaphoff, S., von Bloh, W., Rammig, A., Thonicke, K., Biemans, H., Forkel, M., Gerten, D., Heinke, J., Jägermeyr, J., Knauer, J., Langerwisch, F., Lucht, W., Müller, C., Rolinski, S., Waha, K., 2018. LPJmL4 – a dynamic global vegetation model with managed land – Part 1: Model description. Geosci. Model Dev. 11, 1343–1375. https://doi.org/10.5194/gmd-11-1343-201

Thank you, I will include this reference.

7. Page 3 line 15: I am not aware of this LPJmL application and the paper is about rice production in India with no reference to LPJmL.

Thank you. After several reviews there has been lots of juggling around of the introductory text, I think this error must have been introduced as part of that process. I will correct this in the manuscript.

8. Page 3 line 33 to Page 4 line 7: Suggest to move to next section or the model description section.

This will be moved to the model description section.

9. Page 5 line 13: Please explain what an irregular rotation is.

Please see our response to point 3. On reflection, the detail of the crop rotations is not needed in the abstract, therefore we will amend the abstract to remove the types of rotations at Avignon and India. We will change the Abstract text between lines 12-16 of page 1 to be as follows:

*During the secondary crop growing period, the carbon and energy fluxes for Avignon (point simulation) and India (single gridboxes) are modified; they are largely unchanged for the primary crop growing period. For India (single gridboxes), the inclusion of a secondary crop using this sequential cropping method affects the available soil moisture in the top 1.0 m throughout the year, with larger fluctuations in sequential crops compared with single crop simulations even outside the secondary crop growing period.*

In the main manuscript we will ensure that we define clearly what we mean by a regular and irregular crop rotation with the following text included on page 5 at around line 12:

*We define the sorghum-wheat rotation at Avignon as an irregular crop rotation due to the occurrence of occasional long fallow periods. There are no long fallow periods in the rice-wheat rotation for India, so we refer to this as a regular crop rotation.*

10. Page 15 line 11-18: Move to new section Model evaluation

This will be moved as part of the general restructuring in Section 2.1

11. Figure 4, page 16: Is this a result or a model input?

This is a model input. One of the previous reviews requested information on the climate of Avignon using a similar representation to Figure 7 showing similar information for the India gridbox simulations. I will move Figure 4 and 7 to the relevant section of the Appendix.

12. Page 29 line 23-27: Remove the introductory part of the Discussion section.

This will be removed.

13. Page 29 line 29-31: Move this text to the description of the model simulations.

This will be moved to Section describing the model simulations.

14. Page 34: line 8-12: Suggestion to remove text with comment that this text is not a conclusion.

This text will be removed.

[revised manuscript text omitted]

**Figure 7.** Annual climatology of fluxes elimatologies (in day of year) of NPP for WestUP for India-sequential (blacka) and India-single (redb): Carbon fluxes: NPP (a) , and of GPP for India-sequential (b). Heat fluxes: sensible heat $H$ (c) and latent heat $LE$ India-single (d). Soil moisture variables: $\beta$ (e) and soil moisture availability Each of the India locations shown in the top 1 m Fig. 3 is represented by a solid line of soil (f). Moisture fluxesa different colour: evapotranspiration (g) WestUP - black, EastUP - red, WestBi - blue and non-evapotranspiration moisture fluxes (h)EastBi - cyan.

[revised manuscript text omitted]